# RH-dependent organic aerosol thermodynamics via an efficient reduced-complexity model

Kyle Gorkowski[1], Thomas C. Preston[1,2], and Andreas Zuend[1]

[1]Department of Atmospheric and Oceanic Sciences, McGill University, Montreal, Quebec, Canada
[2]Department of Chemistry, McGill University, Montreal, Quebec, Canada

**Correspondence:** Kyle Gorkowski (kyle.gorkowski@mcgill.ca), Andreas Zuend (andreas.zuend@mcgill.ca)

**Abstract.** Water plays an essential role in aerosol chemistry, gas–particle partitioning, and particle viscosity, but it is typically omitted in thermodynamic models describing the mixing within organic aerosol phases and the partitioning of semivolatile organics. In this study, we introduce the Binary Activity Thermodynamics (BAT) model, a water-sensitive, reduced-complexity model treating the non-ideal mixing of water and organics. The BAT model can process different levels of physicochemical mixture information enabling its application in the thermodynamic aerosol treatment within chemical transport models, the evaluation of humidity effects in environmental chamber studies, and the analysis of field observations. It is capable of using organic structure information including $O : C$, $H : C$, molar mass, and vapor pressure, which can be derived from identified compounds or estimated from bulk aerosol properties. A key feature of the BAT model is predicting the extent of liquid–liquid phase separation occurring within aqueous mixtures containing hydrophobic organics. This is crucial to simulating the abrupt change in water uptake behavior of moderately hygroscopic organics at high relative humidity, which is essential for capturing the correct behavior of organic aerosols serving as cloud condensation nuclei. For gas–particle partitioning predictions, we complement a Volatility Basis Set (VBS) approach with the BAT model to account for non-ideality and liquid–liquid equilibrium effects. To improve the computational efficiency of this approach, we trained two neural networks; the first for the prediction of aerosol water content at given relative humidity, and the second for the partitioning of semivolatile components. The integrated VBS + BAT model is benchmarked against high-fidelity molecular-level gas–particle equilibrium calculations based on the AIOMFAC model. Organic aerosol systems derived from $\alpha$-pinene or isoprene oxidation are used for comparison. Predicted organic mass concentrations agree within less than a $5\,\%$ error in the isoprene case, which is a significant improvement over a traditional VBS implementation. In the case of the $\alpha$-pinene system, the error is less than $2\,\%$ up to a relative humidity of $94\,\%$, with larger errors past that point. The goal of the BAT model is to represent the bulk $O : C$ and molar mass dependencies of a wide range of water–organic mixtures to a reasonable degree of accuracy. In this context, we discuss that the reduced-complexity effort may be poor at representing a specific binary water–organic mixture perfectly. However, the averaging effects of our reduced-complexity model become more representative when the mixture diversity increases in terms of organic functionality and number of components.

# 1   Introduction

In observational and modeling studies, non-ideal molecular interactions in liquid phases play an essential role in organic aerosol partitioning, cloud droplet activation, and atmospheric chemistry (Petters and Kreidenweis, 2007; Zuend et al., 2010; Pankow et al., 2015; Rastak et al., 2017; Ovadnevaite et al., 2017). Common thermodynamic mixing models and related
equilibrium frameworks are highly valuable for the computation of non-ideal mixing effects within liquid (aqueous) inorganic, organic, or mixed organic–inorganic phases through activity coefficient predictions. Models frequently used by the atmospheric aerosol community include the Aerosol Inorganic-Organic Mixtures Functional group Activity Coefficient (AIOMFAC) model (Zuend et al., 2008, 2011, 2010), the Universal Quasichemical Functional-group Activity Coefficients (UNIFAC) model (Fredenslund et al., 1975; Yan et al., 1999; Compernolle et al., 2009), the Model for Simulating Aerosol Interactions and Chemistry
(MOSAIC) (Zaveri et al., 2008), the improved thermodynamic equilibrium aerosol model (ISORROPIA II, "equilibrium" in Greek) (Nenes et al., 1998; Fountoukis and Nenes, 2007), and the Extended Aerosol Inorganics Model (E-AIM) (Clegg et al., 1992, 2001; Wexler, 2002; Clegg and Seinfeld, 2004, 2006). Each model comes with its specific advantages and limitations in chemical species and temperature range covered, as well as the trade-off in computational efficiency vs. accuracy. Such models, in combination with vapor pressure models, can predict the gas–aerosol partitioning of volatile and semivolatile inorganic
and/or organic species and thereby the expected aerosol composition and mass concentration for given environmental conditions and appropriate structural information about the chemical species involved. This makes detailed thermodynamic models very useful for model–measurement comparisons in the context of well-characterized laboratory experiments and modeling case studies of particulate matter (PM).

For inorganic salts, acids, and bases, it is possible to directly implement equilibrium thermodynamics models in "online"
large-scale Chemical Transport Models (CTMs). A typical implementation in CTMs is ISORROPIA II, which uses the availability of molecular-level information about the abundance of inorganic aerosol constituents or their gaseous precursors (e.g., ammonia and nitric acid) as model inputs (Nenes et al., 1998; Zhang et al., 2000; Fountoukis and Nenes, 2007; Zhang et al., 2012). More recently, MOSAIC has been used to account for the dynamic partitioning of semivolatile inorganic gases (Zaveri et al., 2008). In the case of organic aerosol and its volatile precursors, molecular-level chemical constituent information is gen-
erally lacking. Implementations of organic aerosol non-ideality, in current and past CTMs, approach the problem by choosing representative surrogate molecules for broad classes of organic compounds or by merely assigning a hygroscopicity parameter to characterize at least the water-affinity of the organic aerosol fraction (Pankow and Barsanti, 2009; Pankow et al., 2015; Pye et al., 2017; Zhang et al., 2012; Jathar et al., 2016; Kim et al., 2019). Aside from mechanistic and implementation challenges, the direct modeling of organic molecular structures would further have very few validation points as ambient measurements are
currently limited and constrained to a select set of identified organics (Tsigaridis et al., 2014; Lopez-Hilfiker et al., 2016; Sand et al., 2017). On top of that, atmospheric organic chemistry and aerosol formation remain an active area of ongoing research (Öström et al., 2017; Brege et al., 2018; Schum et al., 2018; McFiggans et al., 2019). However, research shows that including non-ideal water $\leftrightarrow$ organic interactions (here"$\leftrightarrow$" indicates an interaction) can have a substantial impact on organic aerosol particulate mass concentrations, water content, biphasic morphology, and cloud condensation nuclei (CCN) properties (Bua-

jarern et al., 2007; Zuend and Seinfeld, 2012; Song et al., 2013; You and Bertram, 2015; Gorkowski et al., 2016; Freedman, 2017; Gorkowski et al., 2017; Ovadnevaite et al., 2017; Rastak et al., 2017).

Rastak et al. (2017) showed the importance of aerosol water content in modeling and understanding both experimental findings as well as climate impacts via aerosol–radiation and aerosol–cloud–radiation interactions. In that study, non-ideal molecular interactions and liquid–liquid equilibrium were considered for reconciling aerosol simulations with laboratory measurements of organic aerosol hygroscopicity parameters below and above 100 % relative humidity. To explore the impact on climate, Rastak et al. (2017) assigned a fixed hygroscopicity parameter ($\kappa$) to the organic aerosol fraction, either 0.05 or 0.15, resulting in significant changes in the average top-of-the-atmosphere radiative fluxes in both the NorESM (-1.0 $\mathrm{W\,m^{-2}}$) and ECHAM6-HAM2 (-0.25 $\mathrm{W\,m^{-2}}$) climate model simulations. Therefore, the aerosol effects on climate are sensitive to aerosol water content and, by extension, the aerosol hygroscopicity representation in such large-scale models.

A practical model for non-ideal thermodynamics needs to handle varying levels of chemical input information while producing realistic predictions. The typical models for non-ideal aqueous organic thermodynamics applicable to a broad class of compounds, like AIOMFAC and UNIFAC, require relatively detailed molecular structure information as input. AIOMFAC is a chemical structure-based activity coefficient model that explicitly incorporates solution non-ideality among organics, water, and inorganic ions (https://aiomfac.lab.mcgill.ca; Zuend et al., 2008, 2011; Zuend and Seinfeld, 2012). In that model, as in UNIFAC, the computations involving organic compounds follow a group-contribution approach, which characterizes each organic molecule as a combination of present functional groups and their abundances within that molecule. In contrast, a thermodynamic model able to accept either detailed molecular structure information or far less detailed bulk chemical properties, e.g., molar masses and oxygen-to-carbon ratios (O : C) of organics, would offer more flexibility in environmental chemistry applications where molecular-level chemical structure information is often imperfect or lacking entirely. Only through a tight coupling of adequate models and measurements can we decipher observational evidence pointing at thermodynamic mixing effects, kinetic mass transfer limitations, or new chemical reaction pathways.

In this study, we introduce a newly developed, flexible thermodynamic mixing model and demonstrate its fidelity for activity coefficient calculations and coupled gas–particle partitioning predictions of aqueous organic aerosols. This non-ideal mixing model, called the Binary Activity Thermodynamics (BAT) model, accounts for water $\leftrightarrow$ organic interactions and thereby offers a method for determining the impact of water and the water content of organic phases at a given temperature and equilibrium relative humidity. The model was parameterized using a training database generated with the AIOMFAC model. The training database reliably constrains the BAT model coefficients across the full composition space of interest, as further discussed in Sect. 3. Such a systematic constraint would likely be unattainable if we were to use experimental data only. However, via the use of AIOMFAC, the BAT model is indirectly constrained by experimental data, since the adjustable parameters of the AIOMFAC model were optimized using experimental data (Zuend et al., 2011).

On its own, the BAT model can predict the non-ideal mixing in aqueous organic systems, including a computationally efficient and implicit treatment of the effects of liquid–liquid phase separation, which is important for scarcely water-soluble organic compounds. Moreover, the atmospheric chemistry and physics community will be particularly interested in our inte-

gration of the BAT model within an equilibrium gas–particle partitioning model. The partitioning model we use is a form of the non-ideal Volatility Basis Set (VBS) approach, which is introduced in Sect. 2.

## 2 Theory: Volatility Basis Set with Consideration of Non-ideality and Liquid–Liquid Equilibria

Our VBS describes the gas–liquid equilibria of organics and water using mass concentrations in the derivation, which allows for an easier interpretation of aerosol measurements. The partitioning components can also be lumped into logarithmically-spaced volatility bins forming a basis set, which is typically done in CTMs for computational efficiency. In this VBS derivation, with non-ideality and liquid–liquid equilibria considered, we bring together published information and outline more clearly important considerations and adaptations for a general multiphase case. The vapor–liquid equilibrium for a single liquid phase is derived from the modified Raoult's law (e.g. Pankow, 1994; Zuend et al., 2010). Subsequently, the general non-ideal VBS framework introduced here, accounts for the potential presence of multiple liquid phases in equilibrium. This VBS framework is independent of the activity coefficient model used – as long as compatible activity coefficient reference states are applied (conversions are possible among different choices). Thus, the fundamental equations do not change as activity coefficient models improve.

Derivation of a non-ideal VBS starts from Raoult's law with the inclusion of activity coefficients (Eq. 1). Non-ideal refers here to the mixing behavior in the liquid phase, while the gas phase is assumed to be an ideal gas mixture, which is a good approximation for air under atmospheric pressure (the use of fugacity coefficients would extend it to non-ideal gas mixtures).

The $j^{\text{th}}$ component in liquid phase $\pi$ has a pure-component liquid-state saturation vapor pressure $p_j^{\text{sat}}$ (a function of temperature only), a mole fraction $x_j^{\pi}$, and a composition- and temperature-dependent activity coefficient $\gamma_j^{(x),\pi}$. The $(x)$ superscript denotes it as a mole-fraction-based activity coefficient, and the $\pi$ superscript stands for liquid phase $\pi$. The component's equilibrium partial pressure (vapor pressure) over a bulk solution, $p_j$, is

$$p_j = p_j^{\text{sat}} x_j^{\pi} \gamma_j^{(x),\pi}. \tag{1}$$

On the general notation adhered to hereafter: the subscripts $j$ or $k$ index chemical species, while a subscript $\Sigma_k$ (or $\Sigma_j$) is a short-hand notation referring to the summed total covering all species. The superscripts indicate the corresponding phase: $g$ for gas, $\Sigma_{\pi}$ for all liquid phases, and $g + \Sigma_{\pi}$ for the combined total of the gas phase plus all liquid phases. Multiple liquid phases are indexed by $\pi$ and labeled by the superscripts $\alpha$, $\beta$, and so on until the $\Omega$ phase. Where applicable, a superscript in parentheses indicates the reference state (e.g., $(x)$ for a mole-fraction-based quantity).

The mass-concentration-based VBS framework is related to Eq. (1) by using the ideal gas law to convert vapor pressures and pure-component saturation vapor pressures into gas phase concentrations (i.e., $C_j^g$ and $C_j^{\text{sat}}$). This step yields Eq. (2), with the liquid phase composition expressed via component mole fractions as

$$C_j^g = C_j^{\text{sat}} x_j^{\pi} \gamma_j^{(x),\pi}. \tag{2}$$

The mole fractions ($x_j^\pi$) in that phase can be calculated from liquid phase concentrations $C_j^\pi$ if the molar masses ($M_j$) of all components are known (or reasonably estimable), resulting in

$$x_j^\pi = \frac{C_j^\pi}{M_j \sum_k \frac{C_k^\pi}{M_k}}.$$  (3)

The equilibrium gas-phase concentration of species $j$, expressed by mass concentrations, is obtained by combining Eqs. (2) and (3) into

$$C_j^g = C_j^{\text{sat}} \frac{C_j^\pi}{M_j \sum_k \frac{C_k^\pi}{M_k}} \gamma_j^{(x),\pi}.$$  (4)

In Eq. (4), we have essentially converted Raoult's law into a mass-concentration-based framework while accounting for non-ideality on a mole fraction basis.

## 2.1 Consideration of Multiple Liquid Phases

We have thus far considered the classical case referring to a single liquid phase, for which Eqs. (4) is sufficient to express the gas-phase concentration. If there are multiple liquid phases, like $\alpha$ and $\beta$, they too must be in thermodynamic equilibrium with each other as well as the common gas phase. Meaning the total liquid concentration ($C_j^{\Sigma\pi} = \sum_\pi C_j^\pi$) further separates into distinct liquid phases.

We define the fraction of species $j$ in each liquid phase (relative to total of $j$ in liquids) by $q_j^\pi$ (e.g., $C_j^\alpha = q_j^\alpha \times C_j^{\Sigma\pi}$). By this definition, the summation of $q_j^\pi$ for a single species over all phases is equal to one and the cumulative liquid-phase amounts of $j$ can be determined using any phase of choice, since

$$C_j^{\Sigma\pi} = \frac{C_j^\alpha}{q_j^\alpha} = \frac{C_j^\beta}{q_j^\beta} = ... = \frac{C_j^\Omega}{q_j^\Omega}.$$  (5)

With $C_j^g$ and multiple liquid phases defined, we can establish a relationship with the effective saturation concentration ($C_j^*$), also called the gas–particle partitioning coefficient or effective volatility. The initial definition of $C_j^*$ by Donahue et al. (2006) targeted mixtures of organic compounds only, but Zuend et al. (2010) pointed out its interpretation in a more general form. The effective saturation concentration of each species, including water and other inorganic constituents in liquid phase $\pi$, is defined by Eq. (6). The distribution of a species $j$ among multiple phases $\pi$ is accounted for in the effective saturation concentration by using Eq. (5). The summation over $k$ covers all species and is equal to the total mass concentration from all liquid phases, $C_{\Sigma_k}^{\Sigma\pi}$ ($= \sum_k \sum_\pi C_k^\pi$); this has also been denoted as $C_{\text{PM}}$ or $C_{\text{OA}}$ for organic aerosol systems in other studies. In this derivation, $C_{\Sigma_k}^{\Sigma\pi}$ is used as we include all liquid-phase species while excluding potential solid phases. Therefore,

$$C_j^{*,\pi} = \frac{C_j^g C_{\Sigma_k}^{\Sigma\pi}}{C_j^\pi} q_j^\pi.$$  (6)

The classical single-phase limit is obtained from Eq. (6) by setting $q_j^\pi = 1$ and simplifying $C_{\Sigma_k}^{\Sigma\pi}$ to $C_{\Sigma_k}$ or $C_{\text{PM}}$, which is valid in that case. Continuing the derivation, we then substitute Eq. (4) into Eq. (6) to arrive at a general expression of $C_j^{*,\pi}$ as

follows:

$$C_j^{*,\pi} = C_j^{\text{sat}} C_{\Sigma_k}^{\Sigma_\pi} \frac{\gamma_j^{(x),\pi} q_j^\pi}{M_j \sum_k \frac{C_k^\pi}{M_k}}.$$ (7)

The superscript $\pi$ in $C_j^{*,\pi}$ denotes that the $\pi^{\text{th}}$ liquid phase properties ($C_k^\pi$, $\gamma_j^{(x),\pi}$, and $q_j^\pi$) are used in this computation.

The fraction of $j$ partitioned to the condensed phase (i.e. the total of liquid phases), $\xi_j$, follows this general definition,

$$\xi_j = \left(1 + \frac{C_j^*}{C_{\Sigma_k}^{\Sigma_\pi}}\right)^{-1}.$$ (8)

Lastly, the total species concentration $C_j^{g+\Sigma_\pi}$ (i.e., the summed concentrations of $j$ in gas phase plus liquid phases) multiplied by $\xi_j$ yields the total concentration present in the liquid phase or phases (without specifying amounts in individual liquid phases),

$$C_j^{\Sigma_\pi} = C_j^{g+\Sigma_\pi} \xi_j.$$ (9)

The theoretical core for the equilibrium between multiple liquid phases and a single gas phase is built into Eqs. (7) and (8) – however, without information about how the phase fractions at equilibrium are determined in practice. Moreover, if the system is at thermodynamic equilibrium, then $C_j^*$ would be independent of which set of liquid phase properties are used in the calculation, i.e.,

$$C_j^* = C_j^{*,\alpha} = C_j^{*,\beta} = ... = C_j^{*,\Omega}.$$ (10)

In the following applications, we have only considered up to two liquid phases $\alpha$ and $\beta$, even though the theory derived in this section applies to any number of liquid phases. Our convention is to use phase $\alpha$ as the water-rich phase and phase $\beta$ as the water-poor (therefore organic-rich) phase. Since we use two phases, only $q_j^\alpha$ needs to be known as $1 - q_j^\alpha$ is equal to $q_j^\beta$ in the context here. Lastly, we emphasize again that any mole-fraction-based activity coefficient model can be used in applications of the vapor–liquid equilibrium theory derived in this section.

## 3   Binary Activity Thermodynamics (BAT)

The goal of the BAT model is to produce realistic results of non-ideal water–organic mixing behavior using minimal chemical information. Our target application is organic aerosol thermodynamics, but the BAT model may find applications in a variety of other fields. In any research problem constrained by limited chemical structure information about organic molecules interacting with water in solution, the BAT model can aid in elucidating those non-ideal interactions.

For organic aerosol, the missing thermodynamic effects which have a significant impact on simulations within CTMs or in the context of controlled laboratory studies, are the pseudo-binary interactions among water $\leftrightarrow$ organic, ion $\leftrightarrow$ organic, and organic $\leftrightarrow$ organic pairs of solution species. In complex solution systems, such pair-interactions occur among and in the

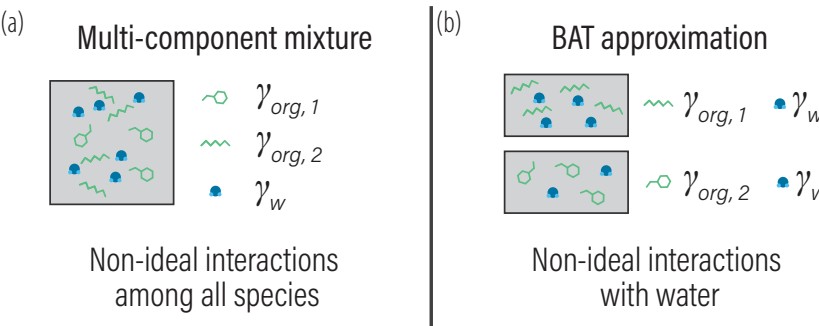

**Figure 1.** **(a)** Schematic of a realistic model for mixture component activity coefficients, accounting for non-ideal interactions between all species. **(b)** The BAT approximation separates the organic species and treats each organic species as a binary mixture with water. The water mole fraction in each binary mixture is then equilibrated with the gas phase water activity (i.e., RH). The water and organic masses in the equilibrated binary mixtures are then added together to represent the total liquid mixture.

presence of a multitude of organic and inorganic ion species. Thus, our initial foray in this work is in predicting activity co-efficients of binary mixtures of water and one organic component, which are then applied to estimate the non-ideal mixing in aqueous multi-organic solutions. At this point, we focus on aqueous organic aerosol phases in the absence of inorganic ions. Most observations and CTMs will report relative humidity (RH) rather than constraining the total water (vapor) mass concen-
5  tration. Under the typical low mass concentrations of organic particulate matter, the RH constraint allows for the prediction of the equilibrium water content and the organic activity coefficient associated with each binary organic–water mixture. This functional dependence of organic activity coefficients on water content can then be used in aerosol partitioning models to sim-ulate co-condensation of several organics with water (e.g., as RH increases) and to determine the possibility and extent of a liquid–liquid equilibrium. For example, this can be performed as a coupled activity–partitioning scheme within the non-ideal
10  VBS partitioning model outlined in Sect. 2.

Reducing the complexity of multicomponent mixing to a system characterized by binary interactions for numerical simplicity is where we elect to introduce a first approximation. In general, the activity coefficients used in Eq. (7) should account for the effects of non-ideal interactions among all species of a liquid phase. However, we approximate the activity coefficient of the $j^{\text{th}}$ organic in the liquid phase as being established solely due to interactions with its associated water fraction at given RH.
15  This concept is shown in Fig. 1 and is related to the idea of the Zdanovskii-Stokes-Robinson (ZSR) mixing rule, which is often applied to determine the water content of multicomponent aqueous (electrolyte) solutions (Zdanovskii, 1948; Stokes and Robinson, 1966; Clegg and Seinfeld, 2004). We are effectively saying that the only non-ideal interactions of a given organic species $j$ are those with its associated water amount and not with other organic species. For typical secondary organic aerosol (SOA) systems, we will show that the resulting error is less than 5 % for the majority of ambient RH.

## 3.1 BAT Activity Coefficient Model

With the scope of the BAT model outlined, we describe the theoretical thermodynamic framework for the binary activity coefficient calculations in the following. In a binary system, the only requirement for a thermodynamically sound activity coefficient model is satisfying the Duhem–Margules relation (Eq. 11), which implies conformance with the more familiar Gibbs–Duhem relation (Margules, 1895; McGlashan, 1963; Gokcen, 1996);

$$(1 - x_{org}) \frac{\mathrm{d}\ln(\gamma_w^{(x)})}{\mathrm{d}x_{org}} + x_{org} \frac{\mathrm{d}\ln(\gamma_{org}^{(x)})}{\mathrm{d}x_{org}} = 0, \qquad \text{(for } T, p \text{ constant).} \tag{11}$$

Here, $x_{org}$ is the mole fraction of the organic component, $T$ is the temperature, $p$ the total pressure, and $R$ the ideal gas constant. Note that the binary case implies $x_w = 1 - x_{org}$ for the mole fraction of water. Consistent with the Duhem–Margules relation, the molar excess Gibbs energy of mixing ($G^E$) is here defined as

$$G^E/RT = (1 - x_{org}) \ln(\gamma_w^{(x)}) + x_{org} \ln(\gamma_{org}^{(x)}). \tag{12}$$

This function describes the excess portion of the molar Gibbs energy of mixing, i.e., the contribution from non-ideal mixing behavior leading to deviations from the ideal molar Gibbs energy of mixing. The two mole-fraction-based activity coefficients are then related to $G^E$ via

$$\ln(\gamma_w^{(x)}) = (G^E/RT) - x_{org} \frac{\mathrm{d}(G^E/RT)}{\mathrm{d}x_{org}}; \tag{13}$$

$$\ln(\gamma_{org}^{(x)}) = (G^E/RT) + (1 - x_{org}) \frac{\mathrm{d}(G^E/RT)}{\mathrm{d}x_{org}}. \tag{14}$$

Equations (12 – 14) are generally valid for a wide range of functional forms of the composition dependence of $G^E$. The only thermodynamic constraint is that a $G^E$ function must also satisfy Eq. (11), which means $G^E$ must be zero for both $x_{org} = 0$ and $x_{org} = 1$. In addition $G^E$ must be capable of expressing maxima and minima within the mixed composition space ($0 < x_{org} < 1$) to correctly capture possible phase separation behavior. To accomplish this dependence, Redlich and Kister (1948) and McGlashan (1963) used a power series expansion in $x_{org}$ of the following form:

$$G^E/RT = x_{org}(1 - x_{org}) \left[ c_1' + c_2'(1 - 2x_{org}) + \ldots + c_n'(1 - 2x_{org})^{n-1} \right]. \tag{15}$$

Using Eq. (15) with Eqs. (13, 14), this power series with adjustable coefficients, $c_n'$ ($n = 1, 2, \ldots$), can be used to fit measured activity coefficient data for any binary system. By increasing the number of adjustable coefficients, any desired level of precision can be achieved – a powerful feature of such a model. In practice, fitting of four or fewer coefficients (not necessarily in sequence) usually leads to model–measurement agreement within experimental uncertainty (e.g. Clegg and Seinfeld, 2006; Zuend et al., 2011). Determining the coefficients for a binary mixture (e.g., malonic acid + water) using Eq. (15) will result in a set of coefficients only meaningful for that system (but unlikely applicable to similar other systems, say succinic acid + water). Therefore, in an attempt to design a more general organic activity coefficient model, we made two important changes.

First, we change the independent composition variable used in Eq. (15). Instead of mole fraction $x_{org}$, we introduce a scaled volume fraction ($\phi_{org}$) in the series expansion of $G^E/RT$, which can be expressed as a function of $x_{org}$ as follows:

$$\phi_{org} = x_{org} \left( x_{org} + (1 - x_{org}) \frac{\rho_{org}}{\rho_w} \frac{M_w}{M_{org}} \left[ s_1 (1 + \text{O} : \text{C})^{s_2} \right] \right)^{-1}. \tag{16}$$

The activity coefficients in Eqs. (13) and (14) remain on a mole-fraction-based scale, because the scaled volume fraction is accounted for in the derivative of the molar Gibbs excess energy with respect to $x_{org}$, by using

$$\frac{\mathrm{d}(G^E/RT)}{\mathrm{d}x_{org}} = \frac{\mathrm{d}(G^E/RT)}{\mathrm{d}\phi_{org}} \frac{\mathrm{d}\phi_{org}}{\mathrm{d}x_{org}}. \tag{17}$$

The exact equations and derivatives are listed in Sect. 2 of the Supplementary Information (SI).

In Eq. (16), $\rho_{org}$ and $\rho_w$ are the liquid-state densities of the organic component and water, respectively, while $s_1$ and $s_2$ are two scaling parameters determined during the model fitting to training data. Note that without the scaling factor in brackets [. . .], this equation would simply relate volume fractions to mole fractions. The densities of organic components are calculated using the relatively simple model by Girolami (1994) outlined in SI Sect. 4. This is advantageous for the reduced-complexity application of this work, because the Girolami (1994) model allows for an estimation of density based on molar mass, O : C, H : C, and N : C only – compatible with limited input information about the chemical structures of organics.

Second, we introduce a parameterization of the scalar $c_n'$ coefficients by means of multivariate functions, which are dependent on common characteristics of organic molecules. The notation change from $c_n'$ to $c_n$ denotes the use of the scaled volume fraction composition scale and the use of a parameterization for $c_n$. Here we use the elemental oxygen-to-carbon ratio (O : C) and molar mass ($M_{org}$) to characterize the organic compounds. We also explored the use of the elemental H : C ratio as an additional molecular property, but found that this descriptor did not noticeably improve the model at the attempted reduced-complexity level. The functional form for the parameterized coefficients based on organic properties is shown by Eq. (18), where $a_{n,1}$ to $a_{n,4}$ are the scalar fit parameters for the $n^{\text{th}}$ coefficient and $\exp(\dots)$ is the natural exponential function;

$$c_n = a_{n,1} \exp(a_{n,2} \times \text{O} : \text{C}) + a_{n,3} \exp\left( a_{n,4} \frac{M_w}{M_{org}} \right). \tag{18}$$

With these changes, we can state a different series expansion of the $G^E$ function using our scaled volume fraction formulation, including the parameterized coefficients $c_n$ (via Eq. 18),

$$G^E/RT = \phi_{org}(1 - \phi_{org}) \left[ c_1 + c_2(1 - 2\phi_{org}) + \dots c_n(1 - 2\phi_{org})^{n-1} \right]. \tag{19}$$

The introduced change of composition scale improves the flexibility of this model when optimized for a wide range of binary systems characterized by the same set of model parameters ($s_1, s_2; a_{n,1}, a_{n,2}, a_{n,3}$, etc., with $n = 1, 2 \dots$). The mole fraction scale works well for binary systems involving two components of similar molecular size and shape. However, this is rarely the case in aqueous organic mixtures with organic compounds of substantially higher molar mass than water. The volume fraction scale implicitly accounts to some extent for the size difference between organic and water molecules, which means that the coefficient functions $c_n$ do not need to correct for the molecular size- and composition-dependence as much as when

mole fraction were used. It is for a similar reason that local composition models like UNIFAC describe organic molecules as a combination of similar-sized segments (subgroups) occupying a regular lattice, which contributes to the so-called combinatorial activity in those models. The scaled volume fraction acknowledges that neither mole fraction nor volume fraction (nor mass fraction) perfectly accounts for the composition-dependence of activity coefficients when describing various binary systems.

Alternatively, a scaled mole fraction composition scale could have been used, but we chose to scale volume fractions as the scaling coefficient values constitute a smaller adjustment when used with this composition scale, meaning that a simpler scaling function was sufficient. Importantly, Eq. (19) remains consistent with all thermodynamic relations, including that $G^E$ becomes zero at both limits: $\phi_{org} = 0$ (when $x_{org} = 0$), $\phi_{org} = 1$ (when $x_{org} = 1$).

Equations (16 – 19) establish a thermodynamically sound activity coefficient model capable of describing various binary

organic–water systems with a common set of model parameters, as shown subsequently. Note, due to the normalization by $RT$, when optimizing our $G^E/RT$ model, we are implicitly accounting for a part of the temperature dependence of activity coefficients, notwithstanding the temperature-independent form of the $c_n$ function. Activity coefficients are weakly dependent on temperature so the error caused by a temperature deviation from 298 K will be relatively small for tropospheric conditions. With the equations for the BAT model derived, the fitted coefficients can subsequently be determined based on suitable

experimental or model-generated data sets.

### 3.2   BAT Model: Training Data and Parameter Optimization

The adjustable parameters of our BAT model were determined by numerical optimization using a database generated by the AIOMFAC model to cover a wide range of organic O : C ratios, molar masses, and mixture compositions at room temperature (298.15 K). The use of the AIOMFAC model as a benchmark allows for generating $x_{org}$, $\gamma_{org}^{(x)}$, and $\gamma_w^{(x)}$ data from highly

dilute to highly concentrated binary aqueous organic mixtures for each system considered, covering the full parameter space of interest. Since the AIOMFAC model includes a UNIFAC group contribution model for short-range molecular mixing, the data we generate in this work reflects the AIOMFAC flavor of a modified UNIFAC model (Zuend et al., 2011) as we do not cover interactions of organics with inorganic ions at this stage. In future, we plan to include ion $\leftrightarrow$ organic and organic $\leftrightarrow$ organic interactions, in which case AIOMFAC may serve again as a benchmark model to generate training data.

We generated a database of 37 known organic chemical structures and 123 artificial, yet possible chemical structures. There were an additional 16 organic chemicals used for a validation database (SI Table S6), and therefore not included in the fitting of the model. The artificial chemical structures start with a carbon chain backbone of variable length, to which a number of OH functional groups are attached. The chain lengths and the number of OH groups were varied such that a comprehensive population of the 2-dimensional O : C versus molar mass parameter space is achieved. The 37 known chemical structures

(mainly dicarboxylic acids) provide some diversity in the covered types of oxygen-bearing functional groups. For each structure there are an additional 40 data points at varying mole fractions, which means the training database has 6400 points and the validation database has 640 points.

Figure 2a shows the data used in the model parameter optimization. This training database was used to simultaneously fit the scalar $a$ and $s$ coefficients of the BAT model (Eqs. 16 – 19) using a constrained global optimization method (known as

GLOBAL) by Csendes (1988), which offers a Fortran implementation of the Boender-Rinnooy-Kan-Stougie-Timmer algorithm (Boender et al., 1982). Through trial-and-error optimization tests, we arrived at the functional forms of the eight power series coefficients ($a_{n,1-4}$; $n = 1, 2$) in Eq. (18) and the two volume fraction scaling coefficients ($s_1$, $s_2$) in Eq. (16). Only the first two terms (involving $c_1$ and $c_2$) in the power series expansion (Eq. 19) were found to be justified given the diversity of organic structures to be represented by a common parameterization. Moreover, we split the model parameterization into three different domains based on the limit of complete miscibility of organics with water and further separated by $O : C$, shown in Fig. 2a as blue, light green, and dark green regions. The blue domain includes components that have no miscibility limit with water. The light green domain starts at $\sim 30\%$ of the $O : C$ ratio reached at the miscibility limit and covers up to the blue domain. The dark green region covers the remaining lower $O : C$ space, which is populated by non-polar, poorly water-soluble organic compounds. In contrast, the blue domain represents relatively hydrophilic organic compounds, whereas the light green domain contains moderately hydrophobic molecules. These domains represent the three regions where each set of optimized parameters dominates. Parameter optimization for each sets of coefficients was carried out on a wider and overlapping $O : C$ range than shown in Fig. 2a. A sigmoidal function was introduced to provide a smooth transition when traversing from one of the domains to the next in the 2-D parameter space (e.g., when $O : C$ is increased gradually at a constant molar mass coordinate) – otherwise, spurious discontinuities would occur. The sigmoidal function provides a weighted mapping between the parameters from one domain to the next (over a short range in the boundary region). The optimal BAT model parameter sets and transition functions are tabulated in SI Sect. 2. An example of the sigmoidal transition function is shown in the SI, Fig. S1.

The limit of miscibility line in Fig. 2a marks the onset of a potential liquid–liquid phase separation in $O : C$ vs. molar mass space. In the domain below that line (at lower organic $O : C$), a miscibility gap is expected over a certain composition range (and corresponding water activity), while above that line there is none predicted. The miscibility limit was determined through an initial BAT fit using only the data in the $O : C$ range from 0.05 to 0.45, prior to the division of the 2-D space into the three domains (details in SI Sect. 2.2).

Generally, the BAT model showed good agreement to the training database with a root mean squared error (RMSE) in $a_w$ of 0.058 (5.8 % RH) and in organic activity ($a_{org}$) of 0.090. The validation database showed a similar agreement with a RMSE in $a_w$ of 0.066 and in $a_{org}$ of 0.096 (details in SI Sect. 5). The BAT model is valid for organic molecules within the following domain: $0 \leq O : C \leq 2$ and $75 \leq M_{org} \leq 500\,\mathrm{g\,mol^{-1}}$ with realistic behavior up to $750\,\mathrm{g\,mol^{-1}}$. Additional error analysis for the BAT model is shown in SI Sect 5. In panels (b) and (c) of Fig. 2, we show two examples of the BAT predictions, after domain-specific optimization, compared to the AIOMFAC-generated data. The BAT model tends to perform very well for the organics of the blue domain, as shown by the citric acid + water example. Citric acid is marked by a blue star in the coordinate space of Fig. 2a. The deviations of the BAT model prediction compared to AIOMFAC increases for hydrophobic compounds; an example is shown for 1-hexanol + water. Even though the model–model deviation increases, those discrepancies are typically amplified where one of the activities (i.e., the product of mole fraction times activity coefficient of a component) is greater than one, which refers to a non-equilibrium state. That is, over the related binary composition range, a miscibility gap would occur at equilibrium, consideration of which is further discussed in Sect. 4.

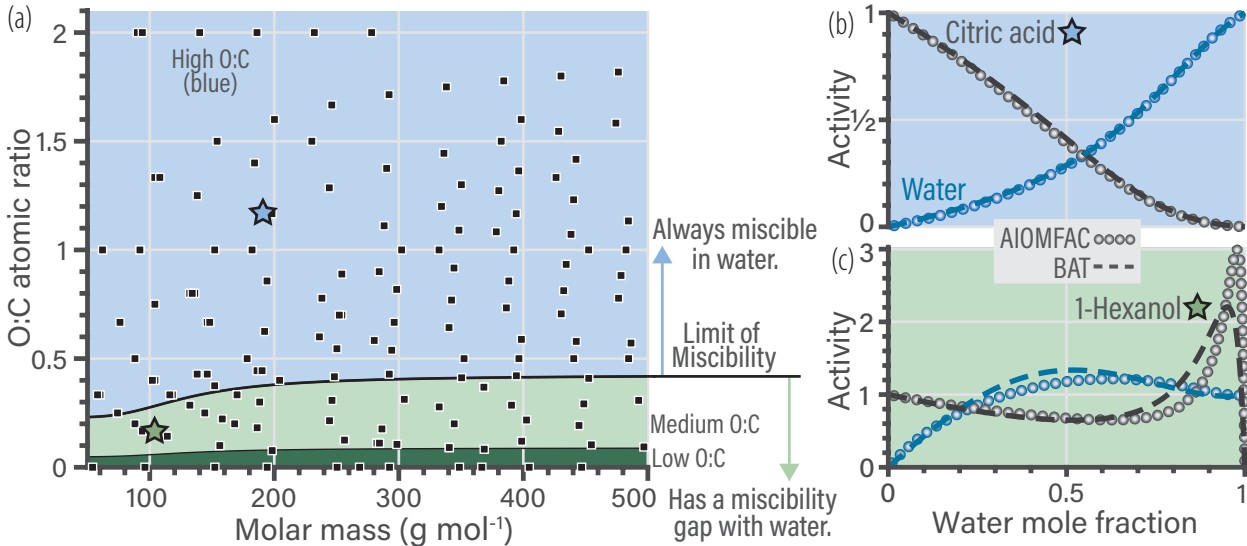

**Figure 2. (a)** The 160 molecular structures (black symbols) used in optimizing the BAT model for hydroxyl functional groups. The three colored regions indicate distinct sets of BAT model fit parameters: the blue region represents high-$O:C$, the light green medium-$O:C$ and the dark green low-$O:C$ organic molecules. The black line marks the determined miscibility limit, meaning that at lower $O:C$, organic molecules exhibit a miscibility gap with water (according to BAT) over a certain composition range, while at higher $O:C$ complete miscibility with water is predicted. **(b,c)** Comparison of the training data generated by AIOMFAC (open circles) with the BAT model (dashed lines) for two examples: **(b)** citric acid + water and **(c)** 1-hexanol + water. Predicted water activity is shown in blue and organic activity in black.

The non-ideal behavior of water–organic mixtures is now explored using the fitted BAT model. In Fig. 3, we can explore where non-ideal behavior is substantial enough to require consideration in aerosol thermodynamic modeling. Isolation of a single parameter in the BAT model can be more informative than using the more detailed AIOMFAC model. In the example of Fig. 3, we fix the molar mass at $200\ \mathrm{g\,mol}^{-1}$ and then scan the $O:C$ ratio from 0 to 1.2. The first items of note are

5    the contours and the associated color space showing water activity, which clearly indicates non-ideal mixing behavior. If the binary mixture were ideal, the white contour lines would be vertical lines referring to $a_w = x_w$. We show an example for $x_w = 0.2 = a_w(\text{ideal})$ in Fig. 3. In the rather hydrophobic region ($O:C \leq 0.4$), the equilibrium water uptake in terms of $x_w$ at given water activity (i.e., bulk equilibrium RH) is less than that of an ideal mixture. For example, an organic compound of $O:C \approx 0.19$ would require $a_w = 0.4$ to result in a mixture water content of $x_w = 0.2$, while an ideal mixture would achieve

10    this water content already for $a_w = 0.2$. Moving up towards higher $O:C$, there is a transition to rather hydrophilic behavior and the water uptake at given equilibrium RH is predicted to become higher than that of an ideal mixture ($a_w < x_w$). A narrow $O:C$ zone bridges the hydrophilic and hydrophobic domains, there the binary mixture would behave like an ideal mixture. In the parameter space displayed, the behavior of any specific binary water–organic system is non-ideal over nearly the whole $x_w$

range. As the mole fraction of water increases beyond 0.9, a binary mixture approaches ideal behavior for high-$O:C$ organic compounds ($O:C \gtrsim 0.8$).

The composition and $O:C$-dependence of liquid–liquid phase separation (LLPS) within binary water–organic systems is also evident from Fig. 3. In general, LLPS is expected to occur when the Gibbs energy of the whole system is minimized (globally) by splitting the system into two (or more) liquid phases of distinct compositions (Zuend et al., 2010, e.g.). In the case of binary aqueous systems, LLPS is indicated when an identical activity (either $a_w$ or $a_{org}$) is predicted for two different mole fractions of water, with the composition range in between defining the miscibility gap (Ganbavale et al., 2015). An example of this is occurring along the $a_w = 0.99$ contour line, denoted by a dashed line in Fig. 3. A clearer example of identifying this phase separation is also shown in Fig. S2 of the SI. In a binary mixture, LLPS is also clearly indicated anywhere a component activity is (predicted) to be greater than 1.0 when assuming a single liquid phase in the calculation (gray areas in Fig. 3). These gray areas mark initial compositions that would be unstable and quickly lead to separation into two phases of distinct water mole fractions; in the case of Fig. 3 with the final phase compositions given by the two intersection points of a line of constant $O:C$ (of compound in question) and the water activity contour at the edge of the phase separation area. Additional isopleths at different organic molar masses (75 to 2000 $\mathrm{g\,mol^{-1}}$) are shown in the SI Sect. 6. Based on BAT predictions, in comparison to the case shown in Fig. 3, this phase separation region moves to higher $O:C$ as the molar mass of the organics increases and to lower $O:C$ as molar mass decreases.

### 3.3 Molecular Functionality Translation

The BAT model described so far is tailored towards molecules dominated by hydroxyl functional groups in terms of oxygen-bearing groups. To increase the model's versatility, we will discuss our approach for incorporating other important oxygen-bearing functional groups into the BAT model framework. One option would involve generating another AIOMFAC training database focused on other functional groups with the subsequent fitting of new BAT model coefficients. This is possible, but for large functional groups the coverage in the $O:C$ *vs.* $M_{org}$ space would be sparse, leading to poorly constrained parameters. Due to that limitation, we went with a molecule functionality translation approach. This approach assumes that the $O:C$ ratio is proportional to a molecule's polarizability, which is then dependent on the type of oxygen-bearing functional group. If that assumption holds to good approximation, the effects of oxygen-bearing groups on activity coefficients can all be translated using a common polarizability scale based on the molecule's $O:C$ ratio. Similarly, if molar mass mainly provides information about the molecules effective volume, then a translation to a new volume scale (affecting the organic volume fraction) is needed as well. The density used in the BAT model is also modified since it is calculated from the $O:C$ and $M_{org}$ inputs.

Based on these assumptions, we use the hydroxyl functionality as a reference oxygen-bearing group and translate the specific properties of all other functionalized molecules to a hypothetical hydroxyl-equivalent molecule of modified $O:C$ and $M_{org}$. We introduce a two-coefficient sigmoidal function to perform this translation (see details in Sect. 2.4 of the SI). The coefficients of the translation function were fitted using AIOMFAC-generated data ($x_{org}$, $\gamma_{org}$, and $\gamma_w$) for each molecular functionality. For example, a common functionality formed via atmospheric chemistry is the hydroperoxide ($CH_nOOH$) group. If a molecule consisted of only hydroperoxide functional groups as oxygen-bearing groups, with an $O:C$ ratio of 1.0 and $M_{org}$

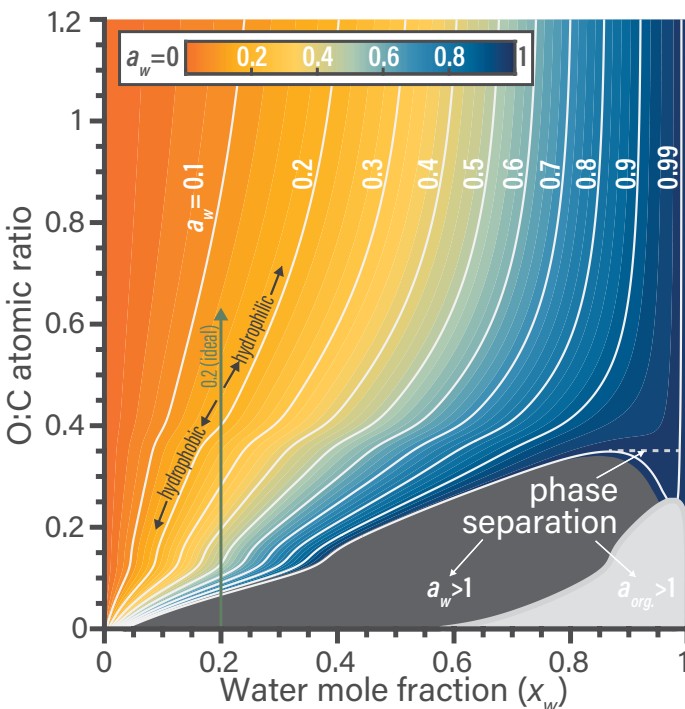

**Figure 3.** Predicted water activity contours generated by the BAT model for binary aqueous mixtures of generic organic compounds of constant molar mass of $200 \, \mathrm{g \, mol^{-1}}$ yet variable $\mathrm{O:C}$ at $T = 298.15 \, \mathrm{K}$. The contours link water mole fraction and the organic $\mathrm{O:C}$ to the resulting water activity in a binary water–organic mixture. The combined shaded regions in dark ($a_w > 1$) and light gray ($a_{org} > 1$) represent the minimum extent of liquid–liquid phase separation for a certain $\mathrm{O:C}$. The dashed tie-line shows an example of phase separation occurring over a limited range in composition along the $a_w = 0.99$ contour, as evident due to two possible $x_w$ values at the same $a_w$. The bumps in the contours at $\mathrm{O:C}$ of 0.12 and 0.4 stem from the transitions between the BAT model's low-, medium-, and high-$\mathrm{O:C}$ parameterization domains.

of $200 \, \mathrm{g \, mol^{-1}}$, the translated hydroxyl-equivalent molecule would have an $\mathrm{O:C}$ ratio of 0.51 and $M_{org}$ of $137 \, \mathrm{g \, mol^{-1}}$. Those two hydroxyl-equivalent molecular parameters are used as inputs for the (hydroxyl-based) BAT model to compute the activity coefficients of water and the actual organic molecule comprising the hydroperoxide functional groups. We reiterate that the BAT model is describing the whole molecule, and so these translations are not for the individual functional groups.

5    This method is different from the group contribution approach taken by UNIFAC and AIOMFAC, as here the whole molecule is assigned one effective functionality. For multifunctional molecules, a distinct multifunctional translation may be derived, like we did for the SOA oxidation products (see Fig. 4b). This can be done by using AIOMFAC to generate training data for multifunctional molecules that are representative of VOC oxidation products. The molecular translation coefficients are then fitted using the generated training database. If this fitting of the translation coefficients is not practical, then the predominant or

10   most representative oxygen-bearing functionality on the molecule should be chosen for an approximate molecule functionality

translation. Extensions to include organic nitrate and sulfate functionalities will be a topic of future development. In principle, additional molecular functionality translations for each combination of molecular functionalities could be developed, which would be practical if the number of permutations is small. If the number of combinatorial permutations of molecular functionalities is large, then that development direction would lead to increased complexity, which is not the goal of the BAT model.

We will explore different weighting and scaling methods of the translations coefficients based on $N:C$ and $S:C$ elemental ratios to retain the reduced-complexity approach. If accurate activity coefficient predictions of a known set of multifunctional molecules are desired and the molecular structures are known, then the use of AIOMFAC or a system specific model instead of BAT is recommended.

Figure 4b shows an example for the translation of a multifunctional hydroperoxide molecule (i.e., containing hydroxyl,
ketone and hydroperoxide functionalities). Such multifunctional hydroperoxide molecules are among the most difficult to represent well when using the functional group translation approach. We show two BAT model activity predictions, BAT (OH) directly used the molecules $O:C$ and $M_{org}$, whereas the BAT (translated) predictions use the translated molecule properties. In general, the translation gives the correct characteristics in terms of predicted water and organic activities but can have large errors. For the multifunctional hydroperoxide example, the BAT (translated) prediction is more hydrophobic than the
untranslated BAT (OH) prediction. The more hydrophobic behavior is consistent with the AIOMFAC predictions. The PEG-414 translation example (Fig. 4c) shows how close PEG is to a hydroxyl molecule, as the BAT (OH) activity curves agree with AIOMFAC. However, the BAT (translated) prediction does show improvement at $x_w > 0.85$. If there is ever a concern about the prediction accuracy for a given molecule, the BAT model output should be compared to experimental data (where available) and/or the AIOMFAC-web model (https://aiomfac.lab.mcgill.ca).

This translation approach can work in both directions, so we can also move the whole BAT model to a different functional group basis, e.g., resulting in carboxyl-based, ketone-based, ether-based, etc. parameterizations of the BAT model (here for the purpose of illustration). We use such translations to plot the limit-of-miscibility lines for all of the fitted functional group types considered (Fig. 4a). The dotted pink line is from the multifunctional hydroperoxide translation and the gold line is from the PEG translation, both have example translations shown in Fig. 4b and 4c rectively. The uncertainty range in the $O:C$
prediction of a limit of miscibility is also shown in Fig. 4a as a shaded gray region. These miscibility limit lines represent the same process (phase separation limit), but for different functional groups, so it is informative to compare their relative positions in Fig. 4a. The higher in $O:C$ the curve is, the more hydrophobic that functional group makes a molecule compared to hydroxyl groups, as it requires a higher $O:C$ to become completely miscible in water (at all proportions of mixing). The relatively large variability among the miscibility limits in terms of $O:C$ ratio emphasizes the importance of distinguishing
among different types of oxygen-bearing functional groups. In the case of ambient and laboratory-generated aerosol mixtures containing inorganic salts, the transition from LLPS to completely miscible (at any composition) spans a $O:C$ ratio range from 0.4 to 0.8 based on experimental data (Song et al., 2012; You et al., 2014; You and Bertram, 2015). That $O:C$ range is comparable to the difference between a hydroperoxide molecule with a molar mass of $100\ \mathrm{g\,mol^{-1}}$ *vs.* $400\ \mathrm{g\,mol^{-1}}$. The wide $O:C$ range can also be achieved by fixing molar mass at $400\ \mathrm{g\,mol^{-1}}$ and either having a hydroxyl or a hydroperoxide
functionalization. This similarity suggests that the types and abundances of oxygen-bearing functional groups are as important

as the salting-out effect by dissolved inorganic ions – at least concerning the miscibility with water. A future investigation on the limit of miscibility line for mixtures with and without dissolved inorganic ions may help elucidate that characteristic.

By the nature of this translation approach, each functional group case will have a similar curvature in the miscibility limit line, as it was propagated from the hydroxyl-based curve. After accounting for the RMSE of the different molecular translations the overall BAT model error in the water activity separation point was $< \pm 0.01$, the $a_w$ prediction error was $< \pm 0.09$, and the $a_{org}$ prediction error was $< \pm 0.15$ (see SI Sect. 5). Also, note that organic molecules with only ester functional groups are predicted to be the only ones having a miscibility gap up to $O : C$ of 1.0 according to the BAT model – and by extension AIOMFAC. Esters are among the poorly-constrained functional groups in AIOMFAC, whereas the hydroxyl functional group is among the well-constrained groups (Zuend et al., 2011). This is the case because the hydroxyl functional group benefits from a large amount of experimental data covering aqueous mixtures of alcohols, polyols, and sugars, enabling tight constraints for its interactions with water and other organic groups. This justifies the use of the hydroxyl group as a reference oxygen-bearing group during our initial fit of the BAT model's coefficients.

## 4 Coupled VBS + BAT Model

The non-ideal BAT model and the VBS approach can now be integrated into a coupled VBS + BAT model to simulate the gas–particle partitioning of organic aerosol systems. This integrated model will be benchmarked against high-fidelity AIOM-FAC gas–liquid equilibrium simulations with consideration of liquid–liquid phase separation. Conceptually, the VBS + BAT approach assumes that each organic is contributing its own water content to the total water content. We use the water mass fractions per organic compound predicted by the BAT model for a given water activity (equivalent to a given equilibrium RH for a bulk solution case) to sum up all the water contributions. This approach is closely related to the ZSR mixing rule for aqueous solutions. Aside from the organic mass concentrations (traditional VBS), the variable $C_{\Sigma_j}^{\Sigma_\pi}$ includes the cumulative water mass concentration from all particle phases, which in turn affects the $C_j^*$ values of all the organic species.

A conceptual flow chart of our VBS + BAT computational approach is shown in Fig. 6. The current version of the program is written in MathWorks $^\circledR$ MATLAB (R2018b) and is available for download (see code availability section).

### 4.1 Consideration of Liquid–Liquid Phase Partitioning

The first nontrivial change in the integrated VBS + BAT model is the consideration and treatment of a potential miscibility gap. In the case of a liquid–liquid equilibrium, the relative phase preferences are described by $q_j^\alpha$, the fractional liquid–liquid partitioning of a component to phase $\alpha$ ($q_j^\alpha \leq 1.0$ in the two-liquid-phases case). Liquid–liquid phase separation in a binary water–organic system at RH $< 100\%$ is reduced to a point and manifests itself by a jump discontinuity (Fig. 5a). The liquid phase is either a water-poor ($\beta$) or water-rich ($\alpha$) phase, with a sharp transition between these two possible states at a certain water activity ($q_j^\alpha = 1$ or 0). However, in the more general case of multicomponent aqueous organic mixtures, there is no discontinuity; rather, a smooth transition occurs in terms of individual component fractions partitioned to each phase depending on phase preference (related to polarity). Hence, the component fractions in phase alpha follow a smooth transition function

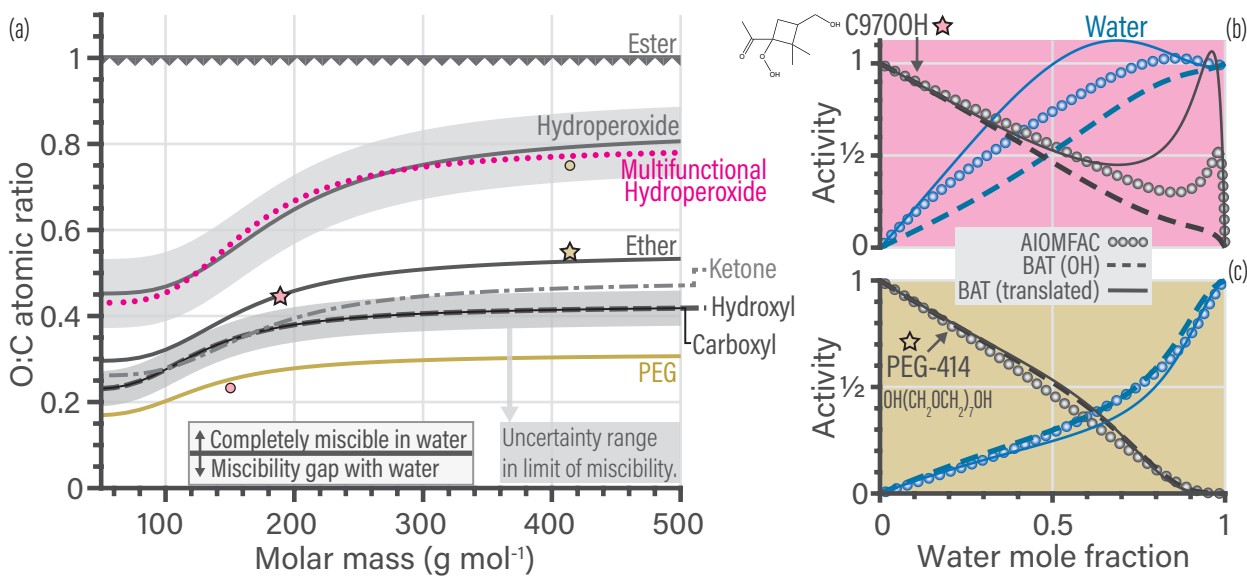

**Figure 4. (a)** Predicted limit-of-miscibility lines for different types of oxygen-bearing functional groups, generated from a translation of the hydroxyl-based BAT model (see text and SI Sect. 2.4). Above the line, the organic is completely miscible in water, and below the line, it has a miscibility gap with water. The uncertainty range (gray shaded region) for the hydroxyl-based limit is the average difference in O : C for the seven molecules that did not conform to the hydroxyl-based miscibility limit line. We propagated the uncertainty range from the hydroxyl-based to the hydroperoxide-based line. The dotted pink line is the translation used in (b), and the gold line is the translation used in (c). **(b,c)** Comparison of the data generated from AIOMFAC (circles), the BAT model with hydroxyl-based (OH) parameters (dashed lines) and the BAT model with translated input values for O : C and $M_{org}$ (solid lines). The change in the model input parameters are shown in (a) as stars for the original value (used in the BAT (OH) line) and circles indicating the translated values used as input for the BAT (translated) calculation. Water activity is in blue and organic activity in black. The comparison for a multifunctional hydroperoxide (C97OOH) is shown in **(b)**. The BAT (translated) curve represents a more hydrophobic organic than the BAT (OH) curve and it shows an $a_w > 1$ range, which is qualitatively consistent with AIOMFAC. The PEG-414 comparison is shown in **(c)** and highlights that the BAT (OH) model already captures most of the PEG behavior. The BAT (translated) curve for PEG-414 shows an improved treatment at $x_w > 0.85$. All cases for $T = 298.15$ K.

for $q_j^\alpha$ with changing RH. In AIOMFAC-based equilibrium calculations, the smooth transition results from the numerical minimization of a system's Gibbs energy, which depends on water content and therefore water activity. For our VBS + BAT model, regardless of binary or multicomponent cases, we represent the transition from a water-poor phase to a water-rich phase as a smooth transition occurring over a finite range in water activity. Instead of using a computationally expensive explicit numerical solution for the individual component's liquid–liquid partitioning, we approximate this transition behavior in a simplified, computationally efficient manner by prescribing a sigmoidal functional form for $q_j^\alpha$ of the organic components in the $a_w$ transition range. This functional form is not arbitrary; rather, it is a result of liquid–liquid equilibrium theory relating $q_j^\alpha$ to activity coefficient ratios in coexisting phases (Zuend and Seinfeld, 2013). Contrary to the organic species, the $q_w^\alpha$ value for water is a derived quantity and not prescribed, since the mass fraction of water contributions are accounted for on a per organic basis in each phase, resulting in a $q_w^\alpha$ value that depends on the liquid–liquid partitioning of all organics.

To approximate the location and $a_w$-width over which the liquid–liquid phase separation is prescribed to occur, we first determine a designated reference point, the so-called water activity separation point ($a_{w,\mathrm{sep}}$). When an organic is in a binary mixture with water, this point denotes the $a_w$ value at which the organic jumps from the water-poor to the water-rich phase ($\alpha$-phase) according to the BAT model prediction (refer to Fig. 5a). The $a_{w,\mathrm{sep}}$ is determined using the BAT model activities and associated Gibbs energy of mixing; see Sect. 3 of the SI for the specifics. Note, the BAT model does not directly output $a_{w,\mathrm{sep}}$, but $a_{w,\mathrm{sep}}$ is derived from the BAT model predicted activities. When there are multiple organic components, each has it's own defined $a_{w,\mathrm{sep}}$ derived from its mixing behavior with water in the binary case. Alternatively, in our model implementation, there is a program option to use a single $a_{w,\mathrm{sep}}$ for a multi-organic mixture, with the $a_{w,\mathrm{sep}}$ value based on average molecular properties of all organics. These average molecular properties are the means of $O:C$ and $M_{org}$ calculated from the liquid-phase species in a $\beta$-phase-only VBS + BAT equilibrium calculation (where $q_{org}^\alpha = 0$). This step allows us to estimate a single representative $a_{w,\mathrm{sep}}$ value for the multicomponent organic-rich phase, even though in reality each organic species may deviate from this average behavior. We then use the $a_{w,\mathrm{sep}}$ value as a reference point when approximating the liquid–liquid phase separation of multicomponent organic mixtures. Since both the behavior of average organic mixtures as well as individual organic compounds can be approximated by single $a_{w,\mathrm{sep}}$ values, the following broadening treatment for the liquid–liquid transition can be applied in both situations.

In our approximation, we set $q_{org}^\alpha = 0.99 \left(= q_{w,\mathrm{sep}}^\alpha\right)$ at the $a_{w,\mathrm{sep}}$ point. Then, for the curve broadening (of the step-like discontinuity), we use a sigmoidal function to approximate the $q_{org}^\alpha$ values representative of a multicomponent aqueous organic mixture (Fig. 5b). With the functional form and one point on the sigmoidal curve determined, we further need to constrain the width of the curve (or alternatively the slope at midpoint). We use the $a_w$ gap from $a_{w,\mathrm{sep}}$ to complete aqueous dilution, where $a_w \to 1$, to set a case-dependent transition function width ($\Delta a_{w,\mathrm{sep}} = 1 - a_{w,\mathrm{sep}}$). Choosing $\Delta a_{w,\mathrm{sep}}$ as the sigmoid half-width results in a gradual two-phase transition and allows the transition range to change for each organic mixture (or organic molecule in the binary case). For molecules that are more hydrophobic than the example represented in Fig. 5, the $a_{w,\mathrm{sep}}$ value would be closer to $1.0$, leading to a smaller $\Delta a_{w,\mathrm{sep}}$, which is consistent with the expected behavior predicted by independent AIOMFAC calculations. We place a minimum limit of $10^{-6}$ on $\Delta a_{w,\mathrm{sep}}$, so that $\Delta a_{w,\mathrm{sep}}$ retains a nonzero width. However, this limit remains a customizable model parameter. Based on these definitions, the sigmoid curve parameter ($s_c$) can

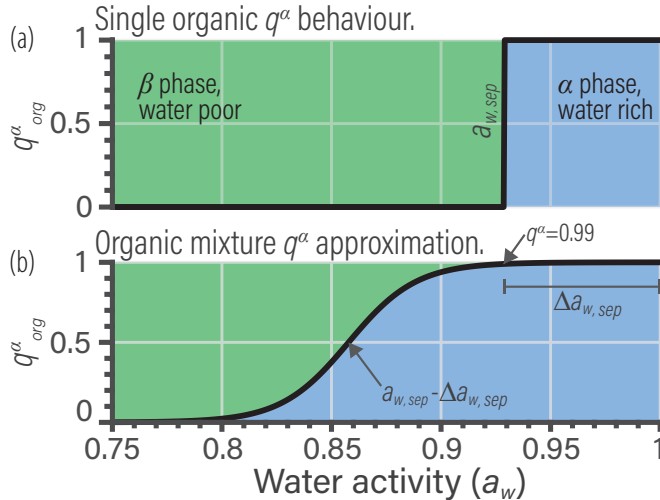

**Figure 5.** Example of the liquid–liquid phase partitioning for 1-hexanol as a function of water activity, expressed by $q_{org}^{\alpha}$. The $q_{org}^{\alpha}$ value is only relevant for mixtures that exhibit a miscibility gap. **(a)** The sharp transition present in a two-component water–organic mixture. **(b)** The broadening of the organic $q_{org}^{\alpha}$ to better represent behavior in multicomponent organic mixtures (Eq. 21).

be determined as

$$s_c = \ln\left(\frac{1}{1 - q_{w,\mathrm{sep}}^{\alpha}} - 1\right)\frac{1}{\Delta a_{w,\mathrm{sep}}}. \tag{20}$$

The value of $q_{org}^{\alpha}$ as a function of $a_w$ is then obtained as

$$q_{org}^{\alpha} = 1 - \frac{1}{1 + \exp\left[s_c(a_w - a_{w,\mathrm{sep}} + \Delta a_{w,\mathrm{sep}})\right]}. \tag{21}$$

5    Even with this approach, the liquid–liquid equilibrium partitioning can sometimes be unrealistic due to the binary mixture approximation. Unrealistic cases are identified by the VBS + BAT-predicted liquid organic aerosol mass dropping below that predicted for a corresponding single-phase simulation (only a single, organic-rich phase present). In such an unrealistic case, we use the average of the $\xi_j$ coefficients of the single-phase prediction and that from a two-phase simulation. Lastly, it is important to note that the $q^{\alpha}$ broadening treatment is only applied when the properties of any of the organic mixture species points to a possible miscibility gap at the water activity of interest. Otherwise, complete miscibility is assumed.

## 4.2   Deep Learning Neural Networks

Moving on from the phase separation treatment, we describe in the following a number of key computational features of our implementation. In designing the VBS + BAT model and its implementation, we aimed both for flexibility and minimal computational overhead. The main computational burden is associated with the non-ideal VBS solver when compared to a VBS solver assuming ideal mixing in the liquid phase, since the number of independent variables increases. The increase is because

the $C^*$ values cannot be approximated as constants, because they are dependent on the mole fractions of the organics and water as well as the activity coefficients. This means achieving convergence iteratively by varying only $C_{\Sigma_j}^{\Sigma_\pi}$ is not possible; instead iteration over the partitioning coefficients $\xi_j$ is necessary, i.e., solving a system of coupled algebraic equations numerically to a desired level of precision. A simple way to speed up convergence towards the equilibrium state is by improving the initial guess for the $\xi_j$ vector. Here we introduce a powerful application of deep learning Neural Networks (NN) for that purpose.

We employ a so-called deep belief network, which consists of multiple layers of artificial neurons (Liu et al., 2017). The neurons are arranged in a matrix and use a sigmoidal activation function which takes inputs from the neurons in the preceding layer, leading to a degree of activation of each neuron, which is then providing input to the next neuron layer. Artificial neural networks require large data sets of desired inputs and outputs to fit the activation function coefficients for each neuron. This allows the NN to "learn" the unspecified functional relationship between known inputs and outputs. In our case, large data sets can easily be generated with random VBS + BAT simulations, allowing for the training of the NN. We found useful applications for NNs for both an inversion of the BAT model and the coupled VBS + BAT model calculations, as noted in Fig. 6.

We use NNs with the BAT model to find the correct $x_{org,j}$ input, since in most applications $a_w$ is known but not $x_{org,j}$. For example, in CTM applications RH is a known quantity and, for bulk equilibrium simulations, the RH in the gas phase is equal to the $a_w$ in the liquid phase (when the Kelvin effect is negligible). The BAT model calculates $a_w$ for a given $x_{org,j}$, so a computationally more expensive approach would be to iterate over $x_{org,j}$ until the given RH in the gas phase and $a_w$ in the liquid phase match (using a solver for non-linear equations). The NN approach attempts to shortcut this costly iterative method by directly guessing $x_{org,j}$ for a given $a_w$. To fit the neuron activation functions, we generate a random data set of $O : C_j$, $M_{org,j}$, $x_{org,j}$, and $a_w$ using the BAT model. The data corresponding to systems with a miscibility gap are parsed into two separate categories to train a separate NN. We generated a database of $9.8 \times 10^6$ data points for miscible organics and $4.6 \times 10^5$ data points for phase separated systems. Each database was then split into training data (70 %), validation data (15 %), and test data (15 %), which was used to train the BAT-NN. Our NN inputs are $O : C_j$, $M_{org,j}$, and $a_w$ with $x_{org,j}$ as the target output. The NN is then generated and its parameters fitted using MATLAB's Neural Network Toolbox. The resulting BAT-NN inverts the BAT model quite well over the full $a_w$ space up to water activities of $\sim 0.95$, above which an iterative refinement is required for good agreement with the targeted $a_w$. For the $a_w < 0.95$ cases, the evaluation time for the BAT model is insignificant, only the iterative refinement of $x_{org}$ to match the given $a_w$ (for $a_w > 0.95$) causes the $0.58\,\mathrm{ms}$ computation time indicated in Fig. 6. The reported computation times were all determined by using a single core on an Intel Core i7-6500 U processor clocked at $2.50\,\mathrm{GHz}$.

Next, we attempted to reduce the computational cost of the VBS + BAT equilibrium solver. For this purpose, we employ a distinct artificial neural network to estimate the equilibrium gas–liquid partitioning coefficient ($\xi_j$) of each species. To facilitate using an NN, we first group the species into 11 decadal $C^{\mathrm{sat}}$ bins from $10^{-6}\,\mu\mathrm{g\,m^{-3}}$ to $10^4\,\mu\mathrm{g\,m^{-3}}$. We tested different NN input combinations and settled on using $C_j^{g+\Sigma_\pi}$, $O : C_j$, $M_{org,j}$, BAT-derived water mass fraction ($w_{w,j}$) and $a_w$ associated with organic component $j$. Using the VBS + BAT equilibrium solver, we generated a random database of 13,000 data points split into training data (70 %), validation data (15 %), and test data (15 %). This generated database was then used for the training of the NN. The NN output target is the vector of partitioning coefficients, which is subsequently used as the initial

guess for solving the coupled VBS + BAT system of non-linear equations. This two-step process (first NN, then numerical equilibrium solver) takes on average $12.8$ ms for a system with 11 species (the time required for the VBS + BAT equilibrium solver step scales approximately linearly with number of species).

The VBS-NN shows a smaller error for lower-$O:C$ ($< 0.5$) systems, but in all cases, it still needs some refinement by an iterative equation solver to achieve a target precision of less than $10^{-5}$ in $\xi_j$ error. With that said, the VBS-NN initial guess is successful in approximating the non-trivial equilibrium solution, which facilitates using an efficient, though less robust, gradient descent method. Our VBS + BAT equilibrium implementation in MATLAB uses the *fmincon* solver with the sequential quadratic programming algorithm for an average evaluation time of $\sim 10$ ms. Without the VBS-NN initial guess, a more robust interior-point algorithm must be used to find the non-trivial solution, resulting in an average evaluation time of $\sim 40$ ms.

The total evaluation time for a system comprised of 11 organic species plus water at a given $a_w$ is between 13 and 19 ms, depending on whether the iterative refinement loop within the BAT evaluation is active or skipped. This evaluation time is similar to that for a standard (ideal mixing) VBS, which on the same CPU results in an evaluation time of $7.2 - 15$ ms (either using the sequential quadratic programming or interior-point algorithm, respectively). Moreover, we expect an optimized Fortran implementation to further improve computational efficiency; thus, the penalty for a higher fidelity organic aerosol model may be even lower. With these implementation issues addressed, the integrated VBS + BAT model can be used to asses the impact of non-ideal mixing thermodynamics on predicted gas–aerosol partitioning and water content, both at low and high RH and for different levels of molecular-level input information.

## 5   Results: Comparison of VBS + BAT and AIOMFAC Predictions

The model comparison focuses on the predictions of bulk liquid aerosol mass concentration and how that metric changes when input data of lower chemical fidelity is used. AIOMFAC-based equilibrium gas–particle partitioning predictions are used as a benchmark. These calculations account for liquid–liquid phase separation and consider relatively high-fidelity input, as the AIOMFAC model uses functional group information for chemical structures and accounts for non-ideal interactions among all species. In contrast, the VBS + BAT approach only includes non-ideal water $\leftrightarrow$ organic interactions (implicitly assuming ideal organic $\leftrightarrow$ organic mixing) and rather limited molecular structure information ($O:C_j$ and $M_{org,j}$).

For our simulated aerosol systems, we use surrogate systems representing $\alpha$-pinene SOA and isoprene SOA products based on predictions from the Master Chemical Mechanism, as was detailed in Zuend and Seinfeld (2012) and Chen et al. (2011), respectively. The $\alpha$-pinene SOA system used here contains 10 organic species as surrogates of the SOA and the isoprene SOA system is comprised of 21 organic surrogate species; these are listed in Sect. 7 of the SI. Both systems have been compared to experimental data using AIOMFAC equilibrium calculations (Zuend and Seinfeld, 2012; Rastak et al., 2017). The pure compound liquid-state vapor pressures used in AIOMFAC equilibrium calculations were predicted by the EVAPORATION model (Compernolle et al., 2011). We use the equilibrium state at $\sim 0$ % RH (i.e., dry conditions) from the AIOMFAC equilibrium simulation to approximate organic particulate matter amounts, comparable to experimental measurements under dry condi-

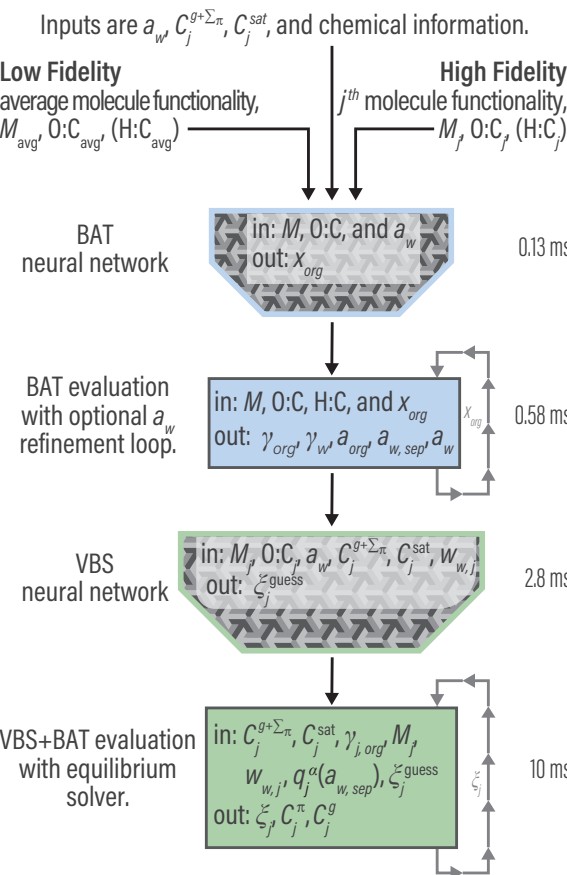

Inputs are $a_w$, $C_j^{g+\Sigma\pi}$, $C_j^{sat}$, and chemical information.

**Low Fidelity**
average molecule functionality,
$M_{avg}$, O:C$_{avg}$, (H:C$_{avg}$)

**High Fidelity**
$j^{th}$ molecule functionality,
$M_j$, O:C$_j$, (H:C$_j$)

BAT
neural network

in: $M$, O:C, and $a_w$
out: $x_{org}$

0.13 ms

BAT evaluation
with optional $a_w$
refinement loop.

in: $M$, O:C, H:C, and $x_{org}$
out: $\gamma_{org}$, $\gamma_w$, $a_{org}$, $a_{w,sep}$, $a_w$

$x_{org}$

0.58 ms

VBS
neural network

in: $M_j$, O:C$_j$, $a_w$, $C_j^{g+\Sigma\pi}$, $C_j^{sat}$, $w_{w,j}$,
out: $\xi_j^{guess}$

2.8 ms

VBS+BAT evaluation
with equilibrium
solver.

in: $C_j^{g+\Sigma\pi}$, $C_j^{sat}$, $\gamma_{j,org}$, $M_j$,
$w_{w,j}$, $q_j^\alpha(a_{w,sep})$, $\xi_j^{guess}$
out: $\xi_j$, $C_j^\pi$, $C_j^g$

$\xi_j$

10 ms

**Figure 6.** High level program outline for the VBS + BAT model, including the use of two artificial neural networks. The listed times represent time per single evaluation call, averaged over 13,000 random simulations for a system comprised of 11 species. The H : C value, used in the component density calculation, is estimated when not given (see SI Sect. 4).

tions. From the dry AIOMFAC equilibrium simulation, the effective $C_{dry}^{sat}$ for each organic species is calculated, which is used as an input in the VBS + BAT simulations. This process allows for a fair comparison between AIOMFAC and VBS + BAT equilibrium simulations, since we are starting with the same dry mass concentrations, the only difference being the treatment of non-ideality and phase equilibria as a function of RH.

## 5.1 Co-condensation of Organic Matter

Organic matter co-condensation is the first improvement the VBS + BAT model offers over the standard VBS (dry) model. Here, co-condensation refers to the RH-dependent gas–particle partitioning of different organic compounds alongside changes of aerosol water content (Topping et al., 2013). In Fig. 7a, the VBS + BAT model, using the individual organic molecule

properties (O : C$_j$, H : C$_j$, $M_j$ and effective $C^{sat}_{j,\text{dry}}$), is compared to a standard VBS (dry) prediction (inputs: $M_j$ and effective $C^{sat}_{j,\text{dry}}$) and a VBS + BAT prediction using average molecular properties for representing the organic aerosol fraction. The average inputs (O : C$_{\text{avg}}$, H : C$_{\text{avg}}$ and $M_{\text{avg}}$) used in the VBS + BAT (avg. prop.) simulation case are mass-weighted means obtained from the dry AIOMFAC equilibrium calculation output. That calculation case uses a recalculated effective $C^{sat}_{\text{avg. dry}}$.

The recalculated $C^{sat}_{\text{avg. dry}}$ is needed to force all the simulations to be equal in total organic aerosol mass concentration at 0 % RH. The VBS + BAT (avg. prop.) case mimics a situation where measurements of the volatility distribution ($C^{sat}_{\text{dry}}$) and of bulk organic properties (O : C$_{\text{avg}}$, H : C$_{\text{avg}}$ and $M_{\text{avg}}$) are available, e.g. from laboratory or field experiments. This also reflects a situation comparable to using the minimal input properties needed for a implementation of VBS + BAT in a CTM.

The percentage difference in PM organic mass of both the high fidelity and averaged VBS + BAT simulations compared to the benchmark calculation is less than 5 % over the majority of the RH range (Fig. 7b). A notable deviation occurs only at high (> 96 %) RH, where a relatively sharp transition to a water-rich phase occurs in the $\alpha$-pinene SOA system (affected by the approximation via the prescribed $q^\alpha$ function in VBS + BAT). At an RH of 99.95 % the error in VBS + BAT, VBS + BAT (avg. prop.), and VBS (dry) are respectively 43 %, −12 %, and −21 % for $\alpha$-pinene SOA. The agreement is closer for the isoprene SOA case, for which the error in VBS + BAT, VBS + BAT (avg. prop.), and VBS (dry) are respectively 0.01 %, −0.2 %, and −44 % at an RH of 99.95 %. The VBS + BAT model performs remarkably well and represents a clear improvement over the standard VBS (dry) model, which ignores relevant water uptake of the isoprene SOA system over a large range in RH and the high-RH change to a water-rich phase in the $\alpha$-pinene SOA case. The latter is particularly relevant for capturing more realistic CCN activation behavior, further discussed in Sect. 5.3.

A second AIOMFAC equilibrium calculation probes the effect of inorganic salts by adding a 50 % dry mass fraction of ammonium sulfate. The salting-out effect does not drastically affect the resulting organic particulate matter mass concentration. However, there is room for improvement of the BAT model by accounting for ion $\leftrightarrow$ organic interactions. The inorganic salts will affect which phase or phases the organics partition into, yet the present VBS + BAT model is not accounting for this. This result suggests a CTM implementation could use a ZSR approximation to combine the water content contributed by inorganic salts and organics (treated as completely phase-separated). To validate that approach and its limitations, we will need to evaluate a much broader set of organic species and salt concentrations in future work.

## 5.2 Hygroscopic Growth

Equilibrium water uptake as a function of RH and its indirect effect on the partitioning of organics is a crucial process. The VBS + BAT simulations account for this process, while traditional VBS implementations do not. Referring to the SOA systems shown in Fig. 7, the hygroscopic growth predictions by VBS + BAT and the AIOMFAC-equilibrium model are compared in Fig. 8. Panel **(a)** shows the absolute PM water mass concentrations as a function of RH, while panel (b) represents hygroscopicity in terms of predicted $\kappa_{\text{HGF}}$ parameters. According to the AIOMFAC-based equilibrium prediction, the water uptake by $\alpha$-pinene SOA is low for RH < 98 %, as expected from previous studies (Rastak et al., 2017; Zuend and Seinfeld, 2012), which the VBS + BAT model captures well. For isoprene-derived SOA, the VBS + BAT (avg. prop.) simulation underpredicts the water content by a substantial amount, while the case with individual surrogate components performs well.

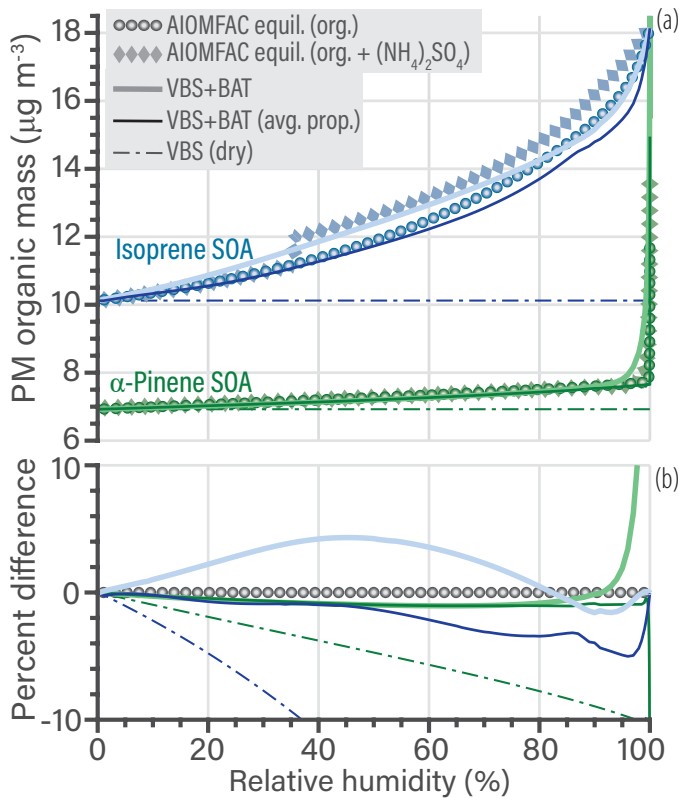

**Figure 7. (a)** Comparison of predicted PM organic mass concentrations as a function of equilibrium relative humidity for a bulk solution ($= a_w$) at 298.15 K. Simulations for isoprene SOA are shown in blue and those for $\alpha$-pinene SOA in green. The benchmark AIOMFAC equilibrium predictions are shown for the salt-free cases (circles); for comparison, an additional case (diamonds) shows SOA mixed with approximately $50\%$ ammonium sulfate (dry mass fraction). In the AIOMFAC equilibrium calculation, ammonium sulfate crystallization was suppressed for RH $> 35\%$. Note that the ammonium sulfate mass concentration in the PM is not shown, only its indirect effect on organic mass concentration. The thick curves show the VBS + BAT predictions with multiple organic surrogate components of individual molecular properties, while the thin curve shows a simulation assuming a hypothetical average molecule calculated from the dry mass, i.e., a mass-weighted mean of $O:C_j$, $H:C_j$, $M_j$, but keeping the set of individual molecule effective $C_{j,\mathrm{dry}}^{sat}$ values to mimic a distribution of volatilities. The thin dashed line shows the standard VBS simulation ignoring water uptake (dry). **(b)** The percentage difference of the three VBS simulations compared to the organics-only benchmark case. Figure S8 of the SI shows an equivalent graph but with expanded limits.

We traced this discrepancy back to the treatment of the IEPOX isoprene oxidation products in the VBS + BAT (avg. prop.) run. In the molecule-specific VBS + BAT simulation, the IEPOX products are treated using hydroxyl functional groups, and all other components are multifunctional hydroperoxides. Whereas the VBS + BAT (avg. prop.) run forces all species to be multifunctional hydroperoxides which causes the IEPOX products to be represented as less hygroscopic than they actually are.

To alleviate this side effect, one could split average organic properties into two groups: one assuming hydroxyl functionality and one assuming multifunctional hydroperoxides (e.g., $50\%$ by mass being from the hydroxyl class and $50\%$ from the hydroperoxide class). Lastly, it is interesting that at $90\%$ RH the large relative deviation in water mass ($55\%$) for isoprene SOA only translated to a $4\%$ error in predicted PM organic mass (Fig. 7a). This characteristic is mainly due to one of the surrogate species, a 2-methyl tetrol dimer (Lin et al., 2012), which is always partitioned to the PM (low vapor pressure) but the change

in the applied molecular functionality (in avg. prop. case) changes its effective hygroscopicity and thereby the water content of the simulated aerosol at high RH. See Tables S7 and S8 in the SI for details about the surrogate species of the SOA systems.

The VBS + BAT model provides simultaneous predictions of water and organic partitioning, which means that hygroscopic growth parameters can be calculated for comparison with other models and simpler hygroscopicity parameterizations. In this case, we predict the widely-used hygroscopicity parameter, $\kappa_{\mathrm{HGF}}$, related to the hygroscopic growth factor of the organic

mixture as a function of composition (and indirectly RH). The definition of $\kappa_{\mathrm{HGF}}$ used in this study is slightly different from the $\kappa$ parameter introduced by Petters and Kreidenweis (2007), since our definition accounts for the effect of organic co-condensation. Our generalized definition of $\kappa_{\mathrm{HGF}}$ was introduced by Rastak et al. (2017) (see derivation and justification in their SI). It is given by Eq. (22), where $V$ indicates volume contributions, with $V_{org}$ the cumulative contribution of organic component volumes at any RH level after gas–particle equilibration, while $V_{org,\mathrm{dry}}$ quantifies the total (organic) volume under

dry conditions (RH $\approx 0\%$):

$$\frac{1}{a_w} = 1 + \kappa_{\mathrm{HGF}} \frac{V_{org,\mathrm{dry}}}{V_w + V_{org} - V_{org,\mathrm{dry}}}. \tag{22}$$

Figure 8b shows a comparison of the predicted $\kappa_{\mathrm{HGF}}$ values. Most VBS + BAT simulations are in good agreement with the benchmark model, except for the VBS + BAT (avg. prop.) run for isoprene SOA. In the average prop. isoprene SOA case, the underpredicted water content is propagated forward causing the $\kappa_{\mathrm{HGF}}$ value to underpredict the AIOMFAC-based benchmark

value. For any given initial particle size, the Kelvin effect could be included and a $\kappa_{\mathrm{CCN}}$ predicted at an adequate level of supersaturation. Although, if the interest is in cloud droplet activation, the Köhler curve can be directly calculated from the VBS + BAT output.

## 5.3 BAT-derived CCN Properties

Our last model application focuses on $\kappa$ at the CCN activation point, denoted as $\kappa_{\mathrm{CCN}}$ of the organic aerosol. The BAT

model is used to understand composition effects on the hygroscopic growth parameter of organic species at CCN activation conditions and the related ongoing discussion within the atmospheric science community. The BAT model can predict an entire Köhler curve directly and does not rely on a $\kappa_{\mathrm{CCN}}$ prediction for applications in the context of cloud droplet formation thermodynamics. Thus, the exercise of predicting $\kappa_{\mathrm{CCN}}$ is here mainly carried out to inform on the relationship with existing

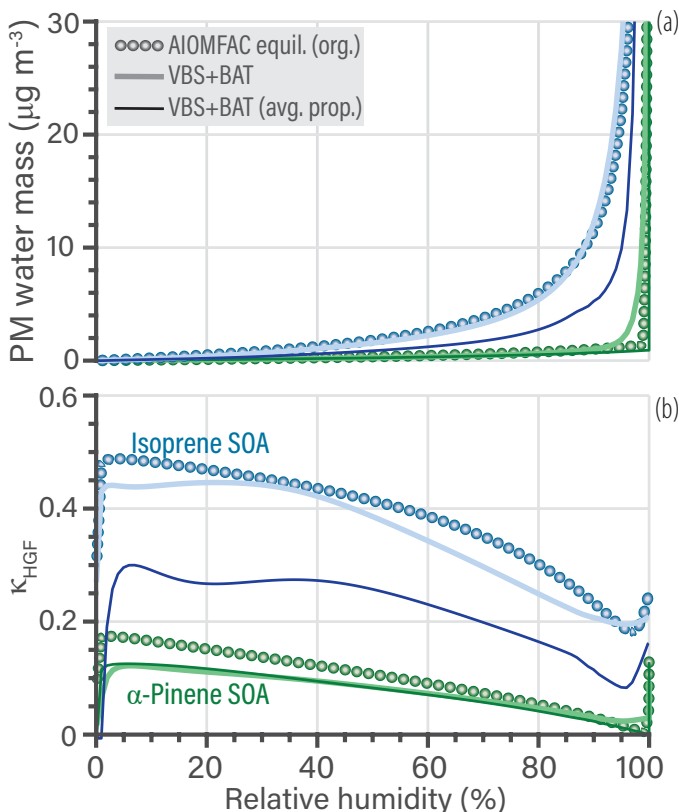

**Figure 8. (a)** Comparison of PM water mass as a function of bulk equilibrium relative humidity. The isoprene SOA simulation is shown in blue and the $\alpha$-pinene SOA simulation in green. **(b)** The calculated $\kappa_{\mathrm{HGF}}$ parameter of the organic mixture, which changes as a function of RH both due to non-ideal mixing and organic co-condensation.

approaches. The $\kappa$-Köhler framework reduces hygroscopic growth to a single parameter ($\kappa$) that can be used to compare the properties of different potential CCN particles (Petters and Kreidenweis, 2007). Over the past decade, the research community progressed by characterizing (organic) aerosol hygroscopic growth measurements by a single $\kappa$ value for ease of comparison and use for parameterizations of CCN activation in large-scale models (e.g. Petters and Kreidenweis, 2007; Rastak et al., 2017). The overarching goal was to link measured aerosol physicochemical properties to CCN activation behavior (critical supersaturation, critical dry diameter, etc.). A common approach was to fit a linear dependence of $\kappa$ to organic O : C (Jimenez et al., 2009; Chang et al., 2010; Massoli et al., 2010; Cappa et al., 2011; Duplissy et al., 2011; Frosch et al., 2011; Lambe et al., 2011; Wong et al., 2011; Rickards et al., 2013; Thalman et al., 2017). A resulting linear fit was not always consistent with observations, due to the nonlinear behavior of $\kappa$ vs. O : C, so Kuwata et al. (2013) introduced a set of water-solubility bins to account for nonlinear step changes. More recently, Wang et al. (2019) focused on relating $\kappa$ to molar mass and assumed ideal mixing of organics with water. There are at least two main factors that many of the previous approaches do not fully account

for. The first is assigning a single $\kappa$ value and assuming it to be representative at all RH levels, which has been shown to be inaccurate in multiple cases as this treatment does not account for non-ideal behavior changing with RH (or $a_w$), especially in the RH range of 90 – 100 % (see Fig. 8b). The second being the use of a linear function to describe the non-linear behavior caused by liquid–liquid phase separation. More advanced thermodynamic models, like UNIFAC and AIOMFAC, have been used to gain insight into the complex CCN activation process accounting for phase separation and non-ideal mixing (Petters et al., 2016; Ovadnevaite et al., 2017; Renbaum-Wolff et al., 2016; Rastak et al., 2017; Hodas et al., 2016). The BAT model can simulate the same processes as those more detailed thermodynamic models, but with less (or incomplete) information about the molecular structure and/or composition of the organic aerosol fraction. We acknowledge that there remain a number of challenges accompanying predictions of CCN activation potential, including accounting for composition-dependent bulk–surface partitioning of different organic and inorganic components in multicomponent aerosol and associated evolving surface tension (e.g. Ruehl et al., 2016; Malila and Prisle, 2018; Davies et al., 2019). At present, those aspects may be best understood and represented by detailed process models, though future BAT extensions may enable improvements also on a reduced-complexity level.

The reduced-complexity inputs of the BAT model and its continuous behavior as a function of $O : C$ and $M_{org}$ allow for establishing a direct link between those organic aerosol properties ($O : C$ and $M_{org}$) and the predicted CCN activation potential. For these BAT model predictions, we revert to the original definition of $\kappa_{CCN}$ by assuming no organic co-condensation in Eq. 22 (i.e., $V_{org,\mathrm{dry}} = V_{org}$). Accounting for the Kelvin effect with an assumption about the air–droplet surface tension, one can calculate the equilibrium saturation ratio $S$ of the aerosol / CCN,

$$S = a_w \exp\left( \frac{4\sigma M_w}{RT\rho_w D} \right). \tag{23}$$

Here, we assume a fixed volume of organics equal to a spherical droplet of 100 nm (dry) diameter over the full RH range ($V_{org} = V_{org,\mathrm{dry}}$). This fixed organic volume means that we are neglecting co-condensation, so that these $\kappa_{CCN}$ values are independent of the organic's volatility. Including co-condensation would tend to increase the apparent value of $\kappa_{CCN}$ when the organic volatility is sensitive to co-condensation, e.g. in the case of semi-volatile organic compounds, but not for extremely low-volatility organic compounds (ELVOC). The surface tension ($\sigma$) used here is a volume-weighted average of water ($\sim 72\,\mathrm{mN\,m^{-1}}$) and a typical organic ($\sim 30\,\mathrm{mN\,m^{-1}}$). Examples of the Köhler curves and the associated $\kappa_{HGF}$ values are shown in Fig. 9. The $\kappa_{CCN}$ value is the $\kappa_{HGF}$ value that corresponds to the maximum point on a Köhler curve. If the organic is completely miscible with water, there is just a single $\kappa_{CCN}$ value. When there is a miscibility gap, we can calculate a $\kappa_{CCN}$ for both the $\alpha$ and $\beta$ phases. Here, the $\beta$-phase $\kappa_{CCN}$ marks the global maximum on the Köhler curve, so we use it as an approximation for these organics. A non-equilibrium model would be needed to accurately resolve the full Köhler curve during dynamic particle growth; the Köhler curve may extend to higher supersaturations in the miscibility gap region since both size and surface tension evolve (which would affect the effective $\kappa_{CCN}$). When the organic particle approaches an $O : C$ of zero, the particle does not activate into a cloud droplet as it remains non-hygroscopic (though it may adsorb a water film at high supersaturations). In those cases, we assign $\kappa_{CCN}$ to be the $\kappa_{HGF}$ value as the water activity asymptotically approaches one.

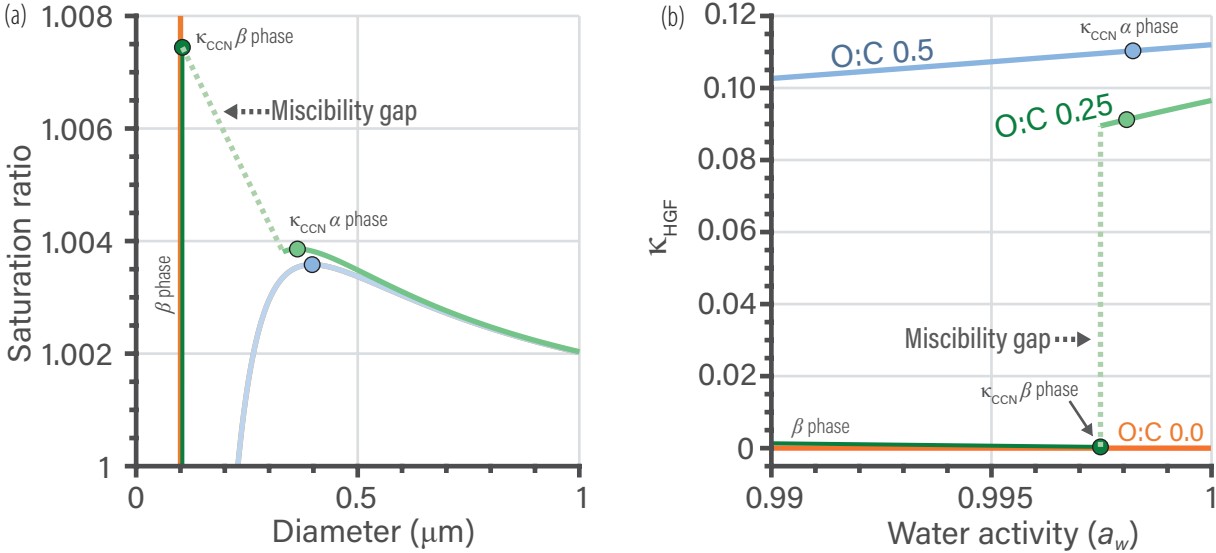

**Figure 9.** Three different organics with $O:C$ of 0.5 (blue), 0.25 (green) and 0.0 (orange) with constant $M_{org}$ of 200 g mol$^{-1}$ at $T =$ 298.15 K. **(a)** CCN growth curves for the three water–organic mixtures. (b) $\kappa_{HGF}$ versus water activity curves for the water–organic mixtures. The circles represent the extracted $\kappa_{CCN}$ values, based on the maximum in saturation ratio vs. particle diameter. The $\beta$-phase curve is shown in a darker shade for the organics with a miscibility gap (green and orange). The miscibility gap is denoted by a dotted line, and is an undefined region in a thermodynamic equilibrium context (crossing it is a transient, non-equilibrium feature). The simulations assume a 100 nm diameter-equivalent volume of organic matter at all points, while water uptake changes the overall particle diameter. The surface tension is approximated as a volume-weighted mean surface tension with pure water $\sigma$ of 72 mN m$^{-1}$ and pure organics of 30 mN m$^{-1}$.

Using the extracted $\kappa_{CCN}$ from individual Köhler curves we can show isopleths of $M_{org}$ in Fig. 10a or $O:C$ (Fig. 10b). Note, these are two-component systems of water and a single organic, which is not necessarily the same as a mixture with an equivalent mean $M_{org}$ and $O:C$ values. The relationship between the mean properties and the resulting $\kappa_{CCN}$ of an organic mixture would depend on the spread of the individual compounds that make up the mixture, which may span non-linear regions in the water uptake behavior. In Fig. 10a, the $\kappa_{CCN}$ values exhibit an $O:C$-dependence, but the magnitude of that dependence varies with molar mass. The $O:C$-dependence of $\kappa_{CCN}$ increases towards lower molar mass of the organic. Focusing on the $M_{org}$-dependence of $\kappa_{CCN}$ at a fixed $O:C$ in Fig. 10b, we notice a nonlinear dependence on molar mass. This is anticipated as the $\kappa_{CCN}$ "ideal" formula also suggests a nonlinear relationship, $\kappa_{CCN} = i\,\rho_{org}\,M_w/(\rho_w M_{org})$, with $i$ as the Van't Hoff factor, i.e., the effective degree of solute dissociation. By synthesizing the two molecular dependencies shown in Fig. 10, we can anticipate how $\kappa_{CCN}$ varies within distinct aerosol populations. As SOA particle mass loading increases, the aerosol fraction of relatively lower molar mass organics (of higher abundance in the gas phase) tends to increase too, which in turn leads to an increase in $\kappa_{CCN}$. A lower total aerosol mass concentration would typically mean that the average molar mass is larger and thus decreases $\kappa_{CCN}$ and indirectly the $O:C$-dependence. This mass loading effect may explain the remaining variability in reported $\kappa_{CCN}$ values, but will need further study.

The measured $\kappa_{\mathrm{CCN}}$ data of $\alpha$-pinene SOA shown in Fig. 10a indicate a water-rich $\alpha$-phase-like behavior. It is interesting that the measured data points start roughly at the limit of miscibility predicted by the BAT model when using the hydroxyl functionalization. That might mean only a small fraction of species need to be miscible to drive the water uptake – and/or that hydroxyl and carboxyl groups are the dominant functionalities of the molecules (both sharing the same BAT functional group translation parameters).

It is also worth comparing the $\alpha$-phase $\kappa_{\mathrm{CCN}}$ predictions for $M_{org} = 300 \text{ g mol}^{-1}$ when applying either the hydroxyl or hydroperoxide molecular functionality parameters with the BAT model. The two $\alpha$-phase curves in Fig. 10a are nearly identical, suggesting that the type of oxygen-bearing functional group is marginal in dilute systems (at the same O : C ratio). This observation explains why an ideal mixing rule can work well over a broad range of O : C (Wang et al., 2019). A limitation when applying an ideal mixing rule by default is clearly identified for system of intermediate to low average O : C, in which a $\beta$-to-$\alpha$ phase transition occurs under hydration conditions.

## 6  Discussion

We developed the BAT model from the desire to capture the thermodynamics of non-ideal water $\leftrightarrow$ organic interactions with only bulk species information, like O : C. In that reduced-complexity effort, we focus on determining representative average relationships and do not expect to model a single component's hygroscopicity and gas–particle partitioning perfectly. The latter case is better approached by group-contribution models like UNIFAC and AIOMFAC – or for high accuracy by system-specific parameterizations (e.g., using a Duhem-Margules model). The goal of the BAT model is to represent the bulk O : C and molar mass dependencies of a wide range of water–organic mixtures to a reasonable degree of accuracy. From this premise, the VBS + BAT model might fail when any one organic compound from a mixture dominates the water uptake. For example, we expect an equimolar mixture of squalane (O : C $= 0$, H : C $= 2$, $M_{org} = 422 \text{ g mol}^{-1}$) and malonic acid (O : C $= 1.33$, H : C $= 1.33$, $M_{org} = 104 \text{ g mol}^{-1}$) to have significant errors ($> 10 \%$) in predicted organic PM mass and water content, since the bulk properties of those compounds are very different. However, a mixture of squalane and 1-hexanol (O : C $= 0.16$, H : C $= 2.33$, $M_{org} = 102 \text{ g mol}^{-1}$), both having a low O : C ratio, is expected to be represented more accurately in a VBS + BAT simulation. That understanding is a prerequisite when using the VBS + BAT model for the interpretation of laboratory studies, but perhaps less critical for the modeling of tropospheric aerosol. An ambient (organic) aerosol is made up of a distribution of organic species, which is in line with the assumptions inherent in the design of the VBS + BAT model. The more species present in a mixture, the less influential any single species becomes. This effect in more complex mixtures may further support the assumption of quasi-ideal mixing among organic compounds (exception may exist). Thus, we expect the VBS + BAT model accuracy to be often better for complex organic aerosol systems than for seemingly simpler ternary systems.

We can examine how this mixture diversity concept plays out by comparing BAT model predictions to recent experimental findings by Marsh et al. (2019), none of which were used in determining the BAT parameter sets. Their experimental work used a comparative kinetic electrodynamic balance method to measure the organic mass fraction ($w_{org}$) of a mixed water–organic

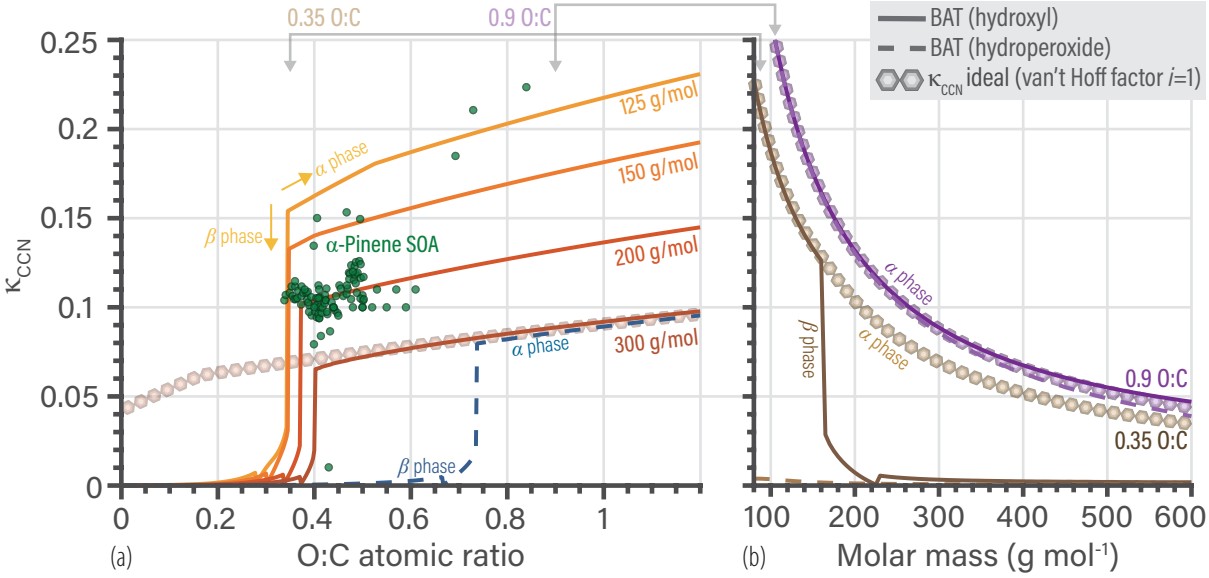

**Figure 10.** Comparison of $\kappa_{CCN}$ derived from $\alpha$-pinene SOA measurements with those from BAT model simulations of CCN activation. The simulations assume a 100 nm diameter equivalent volume of organic matter at the CCN activation point. The droplet surface tension is calculated as a volume-weighted mean; see Sect. 5.3. Shown are the predictions for two-component systems comprising water and a single organic. **(a)** BAT simulations and experimental data for $\kappa_{CCN}$ as a function of O : C. The data for $\alpha$-Pinene SOA is listed in the SI (Massoli et al., 2010; Poulain et al., 2010; Frosch et al., 2011; Kuwata et al., 2013; Rickards et al., 2013; Cain and Pandis, 2017; Wang et al., 2019). **(b)** The molar mass dependence of $\kappa_{CCN}$ for two cases at constant O : C of 0.35 or 0.9, showing ideal mixing as well as miscibility gap considerations with BAT in the O : C = 0.35 case. **(a, b)** BAT simulations based on the hydroxyl group parameter set are shown as solid curves while the dashed curves show a BAT calculation using the hydroperoxide functional group parameters. The thick curves represent single-organic + water systems with the $\kappa_{CCN}$ of the $\beta$-phase transitioning to the one based on the $\alpha$-phase at O : C ratios of 0.3 – 0.4. The hexagons show $\kappa_{CCN}$ for an organic component assuming ideal mixing with water (in panel (a) for $M_{org} = 300\,\mathrm{g\,mol^{-1}}$) and a van't Hoff factor of 1.

droplet (Rovelli et al., 2016). Marsh et al. (2019) measured $w_{org}$ over a wide range of water activities, making this a good comparison for the BAT model; shown in Fig. 11. In their experiments, they had used a few nitrogen–containing organics. In our application of the BAT model for those compounds, the nitrogen atoms were only accounted for in the organic density and molar mass input of the model. We use the organic mixture composition and measured RH from Marsh et al. (2019) for each data point to run an iterative BAT calculation to retrieve the water uptake, which then allow retrieving a $w_{org}$ value. In Fig. 11, the measured and modeled $w_{org}$ values of a variety of aqueous mixtures cluster along the 1:1 line, indicating good agreement. The majority of data points are within $\pm\,10\,\%$ model–measurement uncertainty. Mixtures of pimelic acid isomers (orange squares) will all have identical properties in the BAT model representation due to identical O : C and $M_{org}$ values of these compounds. Therefore, the pimelic acid isomer mixture has no diversity from the BAT model perspective, characterizing

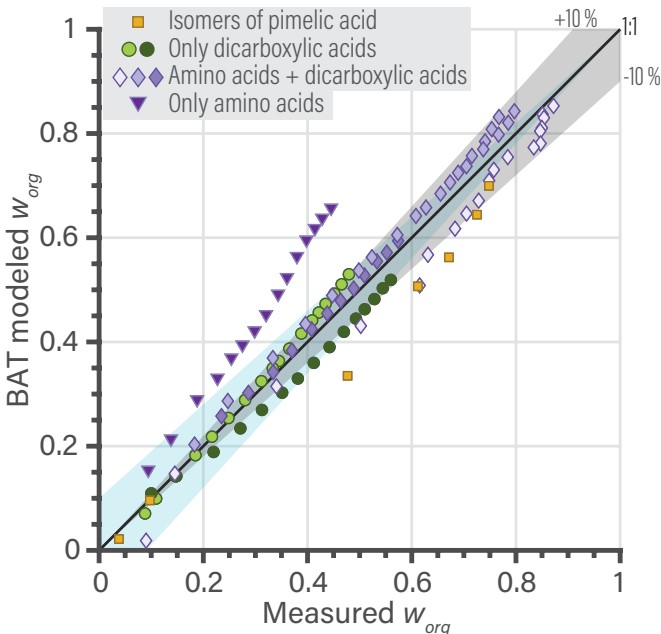

**Figure 11.** Comparison of BAT model predictions of $w_{org}$ with measurements from a comparative kinetic electrodynamic balance by Marsh et al. (2019). The amino acids are glycine, lysine, and arginine; the dicarboxylic acids are oxalic, malonic, glutaric, and methylsuccinic acid; the isomers are pimelic acid, 2,2-dimethyl glutaric acid and 3,3-dimethyl glutaric acid. Gray shading: $\pm$ 10 % uncertainty in $w_{org}$; blue shading: $\pm$ 10 % uncertainty in $w_{\mathrm{water}}$ ($= 1 - w_{org}$). The compositions of each mixture are listed in the SI.

a system for which BAT is expected to perform less accurately. The mixture consisting of amino acids only is also a case where the BAT model clearly deviates from the measurements. Since the BAT model was not trained to predict nitrogen–containing organics, this behavior is not unexpected. However, when the mixture diversity increases by adding dicarboxylic acids to the amino acid mixture, the model error in organic mass fraction, and concurrently water content, reduces to less than 10 %. The error in predicting the water uptake of mixtures of dicarboxylic acids is also on the order of $\pm$ 10 %. In conclusion, the accuracy of the BAT model tends to improve when the organic mixture becomes more diverse.

After mainly comparing to data for subsaturated conditions in Fig. 11, we now focus on predictions for the regime supersaturated with respect to water vapor. In Fig. 12, the measurement derived $\kappa_{\mathrm{CCN}}$ is compared with the corresponding BAT model prediction. The data set contains 30 supersaturated droplet activation measurements of known chemical species (e.g., oleic acid, glucose, and levoglucosan). The average error in the measurements is shown as the gray shaded area in Fig. 12, which covers the average of the $\kappa_{\mathrm{CCN}}$ range observed for each component. A subset of 18 chemicals reported a $\kappa_{\mathrm{CCN}}$ range, from which the average error was calculated to be $\pm$ 42 %. The data set we used was compiled by Petters et al. (2016) and Petters and Kreidenweis (2007), which includes measurements derived from multiple sources (Broekhuizen et al., 2004; Brooks et al., 2004; Frosch et al., 2010; Huff Hartz et al., 2006; Petters et al., 2006; Petters and Kreidenweis, 2007; Petters et al., 2009,

2016; Pradeep Kumar et al., 2003; Raymond, 2003; Suda et al., 2014; Svenningsson et al., 2006). Our comparison excludes the nitrogen-containing compounds. The BAT predictions assumed no organic co-condensation and had an evolving surface tension as described in Sect. 5.3. The BAT predictions vs. measurements had an RMSE of 0.055 and overall agreed within the reported measurement error. Substantial differences are found for the $0.35 < \mathrm{O:C} < 0.55$ range, in which the resulting $\kappa_{\mathrm{CCN}}$ is highly sensitive to a correct prediction of miscibility. For example, the miscibility is over-predicted for phthalic acid $(\mathrm{O:C} = 0.5)$ while it is under-predicted for pinic acid $(\mathrm{O:C} = 0.44)$, shown in Fig. 12. In the full data set of 30 molecules, another subset of 16 molecules were not in the training database of the BAT model, so a corresponding plot with only this validation data is shown in the section 5.1 of the SI, including predictions by both BAT and AIOMFAC. The validation data shows similar agreement to Fig. 12, with a measurement vs. BAT RMSE of 0.061 and measurement vs. AIOMFAC RMSE of 0.059. The AIOMFAC $\kappa_{\mathrm{CCN}}$ predictions are better in the miscibility transition region than the BAT model, but overall the models show similar predictive skill for this metric. We chose to focus on well-defined chemical systems for all of the direct BAT model–measurement comparisons, allowing for minimal uncertainty in the input data. Additional comparisons of BAT to complex ambient and laboratory OA systems will be carried out in the future, since additional analyses are necessary for the estimation of volatility, molecular mass, and $\mathrm{O:C}$ distributions. Such analyses will enable a fair evaluation of VBS + BAT model predictions against measurements for systems that are unresolved on the molecular composition level.

Future work will explore ways to improve the BAT model by adding extensions to include other intermolecular interactions thus far ignored. These additions will focus on organic $\leftrightarrow$ organic and ion $\leftrightarrow$ organic interactions, likely using a similar methodology. The liquid–liquid phase separation treatment via $q^\alpha$ may benefit from improvements, so the predictions of biphasic multicomponent systems become more accurate, especially for cases where the spread in $\mathrm{O:C}$ and $M_{org}$ of organic components is large. We will look at this using two methodologies, the first using AIOMFAC-derived data to fit the $\mathrm{O:C}$ dependence of $q^\alpha$. The second method involves building a model similar to BAT, except for three species, i.e., water $\leftrightarrow$ organic $\leftrightarrow$ organic (Redlich and Kister, 1948). That approach will allow calculating the $q^\alpha$ for each organic directly, but this would add additional computational costs within the VBS equilibrium solver. Throughout such improvements, added complexity needs to be balanced by considerations of computational costs and whether a significant improvement over the current methods are achieved.

## 7   Conclusions

In this study, we introduced the BAT model, which was designed to access varying levels of chemical fidelity. This flexibility means that the integrated VBS + BAT model is well suited for both comparison to experimental observations and for implementations in global and regional atmospheric chemical transport models. In both application cases, the typical lack of chemical structure information precludes the direct use of more detailed models, such as AIOMFAC-based gas–particle equilibrium calculations. The VBS + BAT integration solves this problem by allowing for non-ideal thermodynamic simulations in organic–water systems, even when molecular structure information is limited to bulk elemental composition. This flexibility also promises its utility in CTM implementations, in which the tracking of the exact chemical structures for all species is

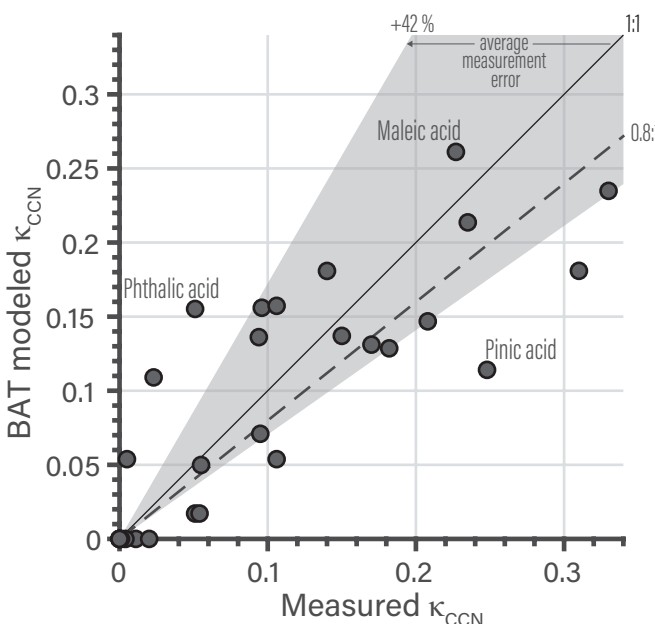

**Figure 12.** Single-component organic aerosol measurements of $\kappa_{\mathrm{CCN}}$ are compared against those predicted by corresponding BAT model simulations of CCN activation. The gray shading represents $\pm$ 42 % average uncertainty in the measured $\kappa_{\mathrm{CCN}}$. The dashed line is a linear fit with a zero intercept, $\kappa_{\mathrm{CCN,BAT}} = \kappa_{\mathrm{CCN,measured}} \times 0.799 \, [\pm 0.059]$ with a Pearson's $R^2$ of 0.66. The model–measurement RMSE was 0.055. The BAT simulations assume a 100 nm diameter equivalent volume of organic matter at the CCN activation point. The droplet surface tension is calculated as a volume-weighted mean. A list of the 30 measurement points is given in Table S6 of the SI, with the data obtained from the following studies: Broekhuizen et al. (2004); Brooks et al. (2004); Petters et al. (2006); Petters and Kreidenweis (2007); Petters et al. (2009, 2016); Frosch et al. (2010); Huff Hartz et al. (2006); Pradeep Kumar et al. (2003); Raymond (2003); Suda et al. (2014); Svenningsson et al. (2006).

impractical, yet information about the evolution of the bulk aerosol (and gas phase) properties is often available. The computational overhead for the VBS + BAT model is comparable to the standard VBS, due to the integration of deep learning neural networks, reducing the need for costly numerical iteration. In conclusion, the implementation of a more realistic organic aerosol model in CTMs is likely feasible.

5    Our comparisons of AIOMFAC-based equilibrium and VBS + BAT simulations demonstrate agreement within about 5 % error over the majority of the RH range. Due to the limited information available from the BAT model, the transition through a miscibility gap had to be prescribed via a semi-empirical transition function instead of an explicit prediction. This prescribed transition in the VBS + BAT model did introduce additional error in the equilibrium partitioning at high humidities for organic mixtures with a miscibility gap – but is beneficial in terms of computational efficiency. The VBS + BAT model can be used

10   reliably across a wide range of the composition space, but our test cases show that caution should be used in the composition range near the onset of a liquid–liquid miscibility gap.

The interplay between O : C, molar mass, and water uptake for CCN activation clearly show the complex behavior of organic $\kappa_{\mathrm{CCN}}$ values. Our distinction between $\kappa_{\mathrm{CCN}}$ and the more general $\kappa_{\mathrm{HGF}}$ helps to differentiate between the subsaturated and supersaturated behavior of organic aerosol. The use of the BAT model in $\kappa_{\mathrm{CCN}}$ prediction correctly captures the nonlinear dependence of $\kappa_{\mathrm{HGF}}$ (and $\kappa_{\mathrm{CCN}}$) on organic properties and is preferable to previous linear fits.

Finally, we present a comparison between the BAT model and comparative kinetic electrodynamic balance measurements of organic mass fractions as a means of independent verification of BAT. The comparison highlights how the BAT model may perform relatively poorly in the cases of certain individual organic species, but when modeling a mixture diverse in number of components and functional groups, the accuracy tends to improve and is typically within $\pm\ 10\ \%$ uncertainty. A diverse mixture is typically a good description of ambient organic aerosol. Therefore, the BAT model is well suited for reduced-complexity predictions involving ambient organic aerosol thermodynamics. Future work in the context of simplified aerosol thermodynamics will be necessary for the development of computationally efficient models, similar to VBS + BAT, which further account for organic $\leftrightarrow$ inorganic interactions in the presence of dissolved electrolytes.

*Code and data availability.* The data presented here, the MATLAB source code, and a standalone executable of the VBS + BAT model is freely available at https://github.com/Gorkowski/Binary_Activity_Thermodynamics_Model.

*Author contributions.* KG and AZ conceptualized the project and developed the methodology. KG wrote the software and created the visualizations. TP and AZ acquired the financial support. KG, TP, and AZ co-wrote the manuscript.

*Competing interests.* The authors declare no competing interests.

*Acknowledgements.* This project was undertaken with the financial support of the Government of Canada through the federal Department of Environment and Climate Change. We further acknowledge support by the Natural Sciences and Engineering Research Council of Canada (NSERC), grants RGPIN/04315-2014 and RGPIN/06529-2015.

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
