# Peer review of "RH-dependent organic aerosol thermodynamics via an efficient reduced-complexity model"

_Atmospheric Chemistry and Physics, 2019_

## Referee Comment (RC1) · Anonymous Referee #1 · 6 Jul 2019

Gorkowski et al. developed a very useful model BAT that can treat the non-ideal mixing of organics and water and can predict the liquid-liquid phase separation, which is very important in SOA partitioning. The BAT model uses the measurable organic aerosol properties (oxidation state, molar mass and vapor pressure) as inputs and the simulated results agree with a comprehensive thermodynamic model AIOMFAC. The BAT model is successfully coupled with the VBS model predicting the gas-particle partitioning. The topic of this study is timely and highly relevant in improvement of thermodynamic aerosol treatment in chemical transport models. I recommend this manuscript for publication after the following comments can be addressed.

Major comments: My major concern goes to the method of Functional Group Translation: P12, Line 10-11: Can the "functional group translation" also treat the nitrogen-

or sulfur-bearing functional groups? P14, Line 8-10: I suggest adding a more detailed description to explain how to do "a distinct multifunctional translation". How the functional group translation is calculated for C97OOH in Fig.4(b)? The translated O:C ratio and molar mass can be added in Tables S5 and S6 in the supplement. P14, Line 22-25: Is Fig.4a based on the carboxy-based, ketone-based, etc parameterizations? The shaded grey area and the purple line in Fig. 4a are not explained in the main text. Please help me understand Fig.4a.

Minor comments: (1) P5, Line 21: It is not proper to describe Eq. (6) as the effective volatility of "all species". It is still the effective volatility of the compound j but includes water and inorganics in the absorbing phase. (2) P9, Line 9: Could the authors explain more how you get the scaling factor in the form of [s1(1+O:C)s2]? From Section 3.2 it seems s1 and s2 are fitted by the training dataset generated by the AIOMFAC model, instead of experimental data as you wrote here on Line 9. (3) P10, Line 31: The authors wrote "the light green domain starts at $\sim$ 20% of the O:C ratio reached at the miscibility limit and covers up to the blue domain", but from Table S1, it seems the light green domains starts from O:C of 0.05 and covers up to the O:C of 10% of the miscibility line? In the excel file, the mid O:C region is "0.05 < O:C < 0.1+ miscibility line", which is different from Table S1 (0:05 < O: C< 0.1 miscibility line). (4) P11, Line 8-10: Could the authors explain in a more detailed way how the equation (S14) is derived to calculate the limit of miscibility line? How you determined the O:C range of 0.05$\sim$0.45? (5) P11, Line 29: The sentence is correct but the ($a_w > x_w$) confuses me as from Fig.3, for the higher O:C region, the predicted $a_w$ is smaller than $x_w$. (6) P12, Line 6-7: I couldn't see this result from Fig.3 and I don't quite understand the grey areas in Fig.3. Could the authors help explain it? (7) P14, Line 11: It is better to describe the Fig.4 from Fig.4(a) to (c). (8) Figure 6: Should $\xi_j$ be $\xi_{jguess}$ in the output of the VBS neural network? I also suggest add $a_{w,sep}$ in the program outline.

Technical corrections: (1) P2, Line 31: "remains" should be "remain". (2) P10, Line15: should be organic $\leftrightarrow$ organic interactions. The latter "organic" is missing. (3) P31, Line

1: There are two "the" at the beginning of the sentence.

---

## Referee Comment (RC2) · Anonymous Referee #2 · 12 Aug 2019

Gorkowski et al. present a modeling approach to predict the water content, CCN activity, liquid-liquid phase separation, and gas-particle partitioning of single component and mixed organic aerosol. The focus of the work is to produce reduced complexity models that have fast runtime while preserving the fidelity of the predictions. This is achieved by training the reduced complexity model using more computationally expensive modeling framework.

This manuscript is an ambitious attempt to contribute to the efficiency of modeling a wide range of organic aerosol processes. Constructing such comprehensive and fast models is technically demanding and the authors should be complimented for their often clever approaches. For example, conceiving and finding a suitable set of fitting coefficients that represent OA through Eq. (18) and (19) is impressive. A selected set of

validations is presented, and these validations appear to demonstrate that the reduced complexity models are adequate. However, I do have concerns about the stability and validation of the model. A detailed formal evaluation of the BAT and NN model that is independent of training data is needed. Furthermore, more systematic validation of the model predictions against experimental data is needed, especially against single-component CCN data. I anticipate that the paper will be acceptable for publication if formal, systematic, and independent validation is included.

Major comments

Both the BAT model and the NN model are trained. Figure 2a shows the training points for the BAT model. The standard approach in machine learning is to have a training set for which the model is optimized, and a validation set for compounds that the model has not been tuned to. This does not appear to have been done and one might seriously question the fidelity of the model outside the training set. Showing activity for citric acid is insufficient. I recommend that the authors test the model against 100 (or so) compounds that were not used in the optimization and show a scatter plot of AIOMFAC vs. BAT for activity coefficient at various RH, mole-fraction of the predicted miscibility gap, water activity of the miscibility gap, and predicted kappa CCN. Only through independent tests and systematic characterization of the error can one be confident that the BAT coefficients truly represent AIOMFAC. A similar case is to be made for the NN training. Systematic validation against with non-training data needs to be presented.

Related to this point: "Therefore, in an attempt to design a more general organic activity coefficient model, we made two important changes. First, we change the independent composition variable used in Eq. (15). Instead of mole fraction $x_{org}$ , we introduce a scaled volume fraction ($\varphi_{org}$) in the series expansion of $G^E/RT$. Second, we introduce a parameterization of the scalar $c_0$ n coefficients by means of multivariate functions, which are dependent on common characteristics of organic molecules."

(1) Please provide some rationale why switching to volume fraction was required. It is not clear to the referee or from the text.

(2) I don't understand why using Redlich-Kister was selected. The RK expansion can fit to arbitrary precision. While it is true that the model is thermodynamically consistent in the limit of x = 1 and x = 0, the polynomial can lead to maxima and minima in the excess function that may or may not be realistic. This is particularly concerning since the polynomial coefficients are themselves computed through a multivariate model. (It is impressive that the system converged). I am concerned that the BAT model coefficients are overfitted and not be representative of other compositions, especially with respect to a phase separation which represents a discontinuity and is highly sensitive to very small fluctuations in the excess function. This issue re-emphasizes the need to independently verify the fidelity of the BAT model. When addressing this concern, the authors should discuss why RK was selected instead of the Margules or van Laar model, which would be less sensitive to error from the use of polynomials by limiting the shape of the excess function.

There are a few comparisons to actual data. While it is clear that the model cannot be compared to every data point in the literature, the real-world validation appears not to be systematic. For example, it's not clear why single component data from Marsh et al. (2019) was selected for hygroscopic growth and various a-pinene SOA for CCN. The composition dependence of subsaturated water content on O:C for SOA (e.g. Pajunoja et al., 2015, doi:10.1002/2015GL063142) is far more revealing than the supersaturated data. Many data sets for single subsaturated water uptake of single component organic aerosol are available. Ideally a proper validation set would systematically probe O:C and functional group coverage, and would considers experimental error. A plot like Figure 11 should be made for available single component CCN data, including for cases where LLPS is known to control CCN activity. A validated dataset with comparison against UNIFAC/LLPS is available in Petters et al. (2016, GMD, 10.5194/gmd-9-111-2016).

Other comments

The tone of the hygroscopic growth and CCN section should be revised. For examples

"Over the past decade, the research community then progressed by characterizing (organic) aerosol hygroscopic growth measurements by a single $\kappa$ value, with sometimes inconsistent distinction between a $\kappa$ value at subsaturated and supersaturated humidity conditions."

"Our clear distinction between $\kappa$CCN and the more general $\kappa$HGF helps the community understand clearly the subsaturated and supersaturated behavior of organic aerosol"

While it is true that there has been a debate on $\kappa$CCN and $\kappa$HGF the authors should acknowledge that 100s of experimental and modeling papers were devoted to this subject, with many important individual contributions explaining the origin of the discrepancy and the composition dependence of $\kappa$HGF. While the BAT model may capture some of these now very well understood effects, it does not really reveal anything new. Please rephrase the text and/or provide a more nuanced perspective on the topic.

---

## Author Response (AR1)

**1 Anonymous Referee #1**

Gorkowski et al. developed a very useful model BAT that can treat the non-ideal mixing of organics and water and can predict the liquid-liquid phase separation, which is very important in SOA partitioning. The BAT model uses the measurable organic aerosol properties (oxidation state, molar mass and vapor pressure) as inputs and the simulated results agree with a comprehensive thermodynamic model AIOMFAC. The BAT model is successfully coupled with the VBS model predicting the gas-particle partitioning. The topic of this study is timely and highly relevant in improvement of thermodynamic aerosol treatment in chemical transport models. I recommend this manuscript for publication after the following comments can be addressed.

**Authors Response:** We thank the reviewer for her/his appreciation of this work and support of the manuscript. We have added clarifying text about the functional group translation approach and possible extensions to it. Below are our responses to specific comments, with reviewer comments in purple, our responses in black text, and changes to the manuscript showing removed text in red with strikethroughs and added text in blue with underlines. The page numbers listed after "Manuscript Revisions on Page" refer to the revised manuscript.

**1.1 Major comments**

My major concern goes to the method of Functional Group Translation: P12, Line 10-11: Can the "functional group translation" also treat the nitrogen or sulfur-bearing functional groups?

**Authors Response:** This version of the BAT model does not include a parameter set to account explicitly for sulfur or nitrogen moieties. We have expanded on the functional group translation discussion in the main text to point out possible extensions to it. We also changed the phrase to 'molecule functionality translation' since there could be confusion with a traditional group-contribution approach (like in UNIFAC / AIOMFAC) accounting for individual functional group effects rather than a whole molecule's effect based on a predominant functionality classification. Our method is not an individual functional group translation, though it may be possible to develop one, right now a whole molecule is assigned a fixed set of translation coefficients to translate the BAT parameterization when the predominant functionalities of a molecule (or class of molecules when averaging) is known in a system and when it is different from hydroxyl-dominated molecules.

**Manuscript Revisions on Page 14–15:** *We reiterate that the BAT model is describing the whole molecule, and so these translations are not for the individual functional groups . This method is different from the group contribution approach taken by UNIFAC and AIOMFAC, as here the whole molecule is assigned one effective functionality. For multifunctional molecules, a distinct multifunctional translation  may be derived, like we did for the SOA oxidation products (see Fig. 4b).  This can be done by using AIOMFAC to generate training data for multifunctional molecules that are representative of VOC oxidation products. The molecular translation coefficients are then fitted using the generated training database. If this fitting of the translation coefficients is not practical, then the  predominant or most representative oxygen-bearing functionality on the molecule should be chosen  for an approximate molecule functionality translation. Extensions to include organic nitrate and sulfate functionalities will be a topic of future development. In principle, additional molecular functionality translations for each combination of molecular functionalities could be developed, which would be practical if the number of permutations is small. If the number*

*of combinatorial permutations of molecular functionalities is large, then that development direction would lead to increased complexity, which is not the goal of the BAT model. We will explore different weighting and scaling methods of the translations coefficients based on* $N : C$ *and* $S : C$ *elemental ratios to retain the reduced-complexity approach. If accurate activity coefficient predictions of a known set of multifunctional molecules are desired and the molecular structures are known, then the use of AIOMFAC or a system specific model instead of BAT is recommended.*

P14, Line 8-10: I suggest adding a more detailed description to explain how to do "a distinct multifunctional translation". How the functional group translation is calculated for C97OOH in Fig.4(b)? The translated O:C ratio and molar mass can be added in Tables S5 and S6 in the supplement.

**Authors Response:** See the response to the comment above and related changes to the manuscript. As suggested, we have also added translated values to the SI tables.

P14, Line 22-25: Is Fig.4a based on the carboxy-based, ketone-based, etc parameterizations? The shaded grey area and the pink line in Fig. 4a are not explained in the main text. Please help me understand Fig.4a.

**Authors Response:** Yes, Fig. 4a shows the effect of BAT parameterizations with regard to the limit-of-miscibility lines when based on the indicated functional groups rather than hydroxyl. The dotted pink line is the result of applying the multifunctional hydroperoxide translation, which is used in Fig. 4b. This is stated in the figure caption and we now mention it in the text too. The grey area is the error in the $O : C$ prediction, this is explained in the figure caption, and we also added a clarifying note in the main text for this.

**Manuscript Revisions on Page 15:** *We use  such translations to plot the  limit-of-miscibility lines for all of the fitted functional group types considered (Fig. 4a). The dotted pink line is from the multifunctional hydroperoxide translation and the gold line is from the PEG translation, both have example translations shown in Fig. 4b and 4c respectively. The uncertainty range in the* $O : C$ *prediction of a limit of miscibility is also shown in Fig. 4a as a shaded gray region. These miscibility limit lines represent the same process (phase separation limit), but for different functional groups, so it is informative to compare their relative positions  in Fig. 4a.*

Minor comments:

(1) P5, Line 21: It is not proper to describe Eq. (6) as the effective volatility of "all species". It is still the effective volatility of the compound j but includes water and inorganics in the absorbing phase.

**Authors Response:** We meant to state that it applies to all species; phrasing amended.

**Manuscript Revisions on Page 5:** *The effective  saturation concentration of each species, including water and other inorganic constituents in liquid phase* $\pi$*, is defined by Eq. (6).*

(2) P9, Line 9: Could the authors explain more how you get the scaling factor in the form of [s1(1+O:C)s2]? From Section 3.2 it seems s1 and s2 are fitted by the training dataset generated by the AIOMFAC model, instead of experimental data as you wrote here on Line 9.

**Authors Response:** You are correct, it is fitted with the AIOMFAC training dataset.

**Manuscript Revisions on Page 9:**  *In Eq. (16), $\rho_{org}$ and $\rho_w$ are the liquid-state densities of the organic component and water, respectively, while $s_1$ and $s_2$ are two scaling parameters determined  during the model fitting to training data.*

(3) P10, Line 31: The authors wrote "the light green domain starts at $\sim 20\%$ of the O:C ratio reached at the miscibility limit and covers up to the blue domain", but from Table S1, it seems the light green domains starts from O:C of 0.05 and covers up to the O:C of 10% of the miscibility line? In the excel file, the mid O:C region is "$0.05 < O:C < 0.1+$ miscibility line", which is different from Table S1 ($0:05 < O:C < 0.1$ miscibility line).

**Authors Response:**  The excel file $O:C$ bounds are correct, the SI has been revised to match this. Table S1 range is the data used in the fitting of the coefficients but that range does not directly map to the coefficients used in a BAT model calculation. A weighted averaged of the coefficients (SI section 2.3) are used in the BAT calculation; an example graph has been added to the SI. The text has been clarified accordingly.

**Manuscript Revisions on Page 11:**  *The blue domain includes components that have no miscibility limit with water. The light green domain starts at $\sim$  $\sim 30\%$ of the $O:C$ ratio reached at the miscibility limit and covers up to the blue domain.*
*. . .*
*These domains represent the three regions where each set of optimized parameters dominates. Parameter optimization for each sets of coefficients was carried out on a wider and overlapping* $O:C$ *range than shown in Fig. 2a.*
*. . .*
*An example of the sigmoidal transition function is shown in the SI, Fig. S1.*

(4) P11, Line 8-10: Could the authors explain in a more detailed way how the equation (S14) is derived to calculate the limit of miscibility line? How you determined the O:C range of 0.05 to 0.45?

**Authors Response:**  We added text to the SI (page 3, Sect. 2.2) to explain our approach for this.

**SI Revisions on Page 3:**  *The limit of miscibility line is determined from an initial BAT model  fit involving the $O:C$ region close to where the miscibility gap vs. complete miscibility transition occurs. We started by fitting the BAT coefficients using a wide $O:C$ range (0.0 to 0.8) and then progressively narrowed it to the transition region ($O:C$ 0.05 to 0.45). We then scanned $O:C$ and $M_{org}$ to map out where the miscibility transition occurred (within BAT). The resulting $O:C$ values  were used to fit the limit of miscibility line, $\vartheta_{\mathrm{ML}}$, as a function of organic molar mass, *

(5) P11, Line 29: The sentence is correct but the (aw > xw) confuses me as from Fig.3, for the higher O:C region, the predicted aw is smaller than xw.

**Authors Response:**  Thanks for spotting this, you are correct. The sentence has been revised.

**Manuscript Revisions on Page 12:**  *Moving up towards higher $O:C$, there is a transition to rather hydrophilic behavior and the water uptake at given equilibrium RH is predicted to become higher than that of an ideal mixture ($a_w < x_w$).*

(6) P12, Line 6-7: I couldn't see this result from Fig.3 and I don't quite understand the grey areas in Fig.3. Could the authors help explain it?

**Authors Response:** The increase with $M_{org}$ can also be seen in Fig. 4a (hydroxyl curve) and from the additional isopleth plots we added to the SI with this revised version of the manuscript.

The gray areas mark LLPS regions due to either activity (organic or water) being greater than 1, e.g. a pure organic phase would have $a_{org} = 1$, which would therefore favor phase separation whenever the (forced) mixing leads to activities of one or several components exceeding 1. LLPS also occurs where an identical activity (either $a_w$ or $a_{org}$) is predicted for two different mole fractions of water. In this figure, the gray shading indicates that an initial binary mixture within the gray area would be unstable and undergo LLPS, leading to two phases of distinct compositions.

**Manuscript Revisions on Page 13:** *In a binary mixture, LLPS is also clearly indicated anywhere a component activity is (predicted) to be greater than* $1.0$ *when assuming a single liquid phase in the calculation*  *(gray areas in Fig. 3). These gray areas mark initial compositions that would be unstable and quickly lead to separation into two phases of distinct water mole fractions; in the case of Fig. 3 with the final phase compositions given by the two intersection points of a line of constant* $O : C$ *(of compound in question) and the water activity contour at the edge of the phase separation area. Additional isopleths at different organic molar masses (75 to 2000* g mol$^{-1}$*) are shown in the SI Sect. 6. Based on BAT predictions, in comparison to the case shown in Fig. 3, this phase separation region moves to higher* $O : C$ *as the molar mass of the organics increases and to lower* $O : C$ *as molar mass decreases.*

(7) P14, Line 11: It is better to describe the Fig.4 from Fig.4(a) to (c).

**Authors Response:** We went with Fig.4b & c first as they are direct examples, which follow clearly from the introduction of the translation methodology in the main text. We don't think it is necessary for the figure description to be chronological with the main text. We have considered changing the order of the Fig 4. (b & c graphs on the left then a), but we think it is more aesthetically pleasing the way it is.

(8) Figure 6: Should $\xi$j be $\xi$jguess in the output of the VBS neural network? I also suggest add aw,sep in the program outline.

**Authors Response:** We added $a_{w,sep}$ and we changed $\xi$j to $\xi$jguess.

**1.2  Technical corrections**

(1) P2, Line 31: "remains" should be "remain". **Authors Response:** Changed.

(2) P10, Line15: should be organic $\leftrightarrow$ organic interactions. The latter "organic" is missing.
**Authors Response:** Added.

(3) P31, Line1: There are two "the" at the beginning of the sentence.
**Authors Response:** fixed.

**2 Anonymous Referee #2**

Gorkowski et al. present a modeling approach to predict the water content, CCN activity, liquid-liquid phase separation, and gas-particle partitioning of single component and mixed organic aerosol. The focus of the work is to produce reduced complexity models that have fast runtime while preserving the fidelity of the predictions. This is achieved by training the reduced complexity model using more computationally expensive modeling framework. This manuscript is an ambitious attempt to contribute to the efficiency of modeling a wide range of organic aerosol processes. Constructing such comprehensive and fast models is technically demanding and the authors should be complimented for their often clever approaches. For example, conceiving and finding a suitable set of fitting coefficients that represent OA through Eq. (18) and (19) is impressive. A selected set of validations is presented, and these validations appear to demonstrate that the reduced complexity models are adequate. However, I do have concerns about the stability and validation of the model. A detailed formal evaluation of the BAT and NN model that is independent of training data is needed. Furthermore, more systematic validation of the model predictions against experimental data is needed, especially against single component CCN data. I anticipate that the paper will be acceptable for publication if formal, systematic, and independent validation is included.

**Authors Response:** We thank the reviewer for her/his positive comments and the concerns about model validation. Our description did indeed not include finer details about the splits of the database into training/validation/testing data in the context of fitting the neural networks and of BAT. Although, such procedures were followed during development; we have improved the description in the manuscript in this regard. In the revised manuscript version, we have added independent validation data for fitting the BAT coefficients and explored the stability of the BAT model through additional plots added to the SI. Additional text was added to describe the training of the neural networks using BAT-generated random data which was then separated into training (70 %), validation (15 %), and testing (15 %) data sets. Below are our responses to specific comments, with reviewer comments in purple, our responses in black text, and changes to the manuscript showing removed text in red with strikethroughs and added text in blue with underlines. The page numbers listed after "Manuscript Revisions on Page XX:" refer to the revised manuscript.

**2.1 Major comments**

Both the BAT model and the NN model are trained. Figure 2a shows the training points for the BAT model. The standard approach in machine learning is to have a training set for which the model is optimized, and a validation set for compounds that the model has not been tuned to. This does not appear to have been done and one might seriously question the fidelity of the model outside the training set. Showing activity for citric acid is insufficient. I recommend that the authors test the model against 100 (or so) compounds that were not used in the optimization and show a scatter plot of AIOMFAC vs. BAT for activity coefficient at various RH, mole-fraction of the predicted miscibility gap, water activity of the miscibility gap, and predicted kappa CCN. Only through independent tests and systematic characterization of the error can one be confident that the BAT coefficients truly represent AIOMFAC. A similar case is to be made for the NN training. Systematic validation against with non-training data needs to be presented.

**Authors Response:**
The reliability and validation of both BAT and NN models were assessed as outlined more specifically below. We note here

that there is a distinction between the training of an unknown multivariate function, as in the case of a NN, requiring a machine learning approach to determine the functional form (i.e. number of neurons and layers) and fit associated activation function parameters – and that of the BAT model, for which a thermodynamic model was prescribed with fixed coefficient functions (after initial tests), which then becomes a classical parameter optimization problem.

- **Neural Networks:** We have added a more detailed description of the training of the neural networks, as we did use the standard practice of training, validation, and test data sets. Also note that in the case of the NN, these are random data sets generated by the function the NN is trying to invert and not measurements or AIOMFAC-generated data. Hence, a large number of data points ($\sim$ 10 million) were generated for the NN training and validation.

  **Manuscript Revisions on Page 20:** *To fit the neuron activation functions, we generate a random data set of* $\mathrm{O}:\mathrm{C}_j$, $M_{org,j}$, $x_{org,j}$, *and* $a_w$ *using the BAT model. The data corresponding to systems with a miscibility gap are parsed into two separate categories to train a separate NN.* *We generated a database of* $9.8 \times 10^6$ *data points for miscible organics and* $4.6 \times 10^5$ *data points for phase separated systems. Each database was then split into training data (70 %), validation data (15 %), and test data (15 %), which was used to train the BAT-NN.* *Our NN inputs are* $\mathrm{O}:\mathrm{C}_j$, $M_{org,j}$, *and* $a_w$ *with* $x_{org,j}$ *as the target output.*

  And: *We tested different NN input combinations and settled on using* $C_j^{g+\Sigma_\pi}$, $\mathrm{O}:\mathrm{C}_j$, $M_{org,j}$, *BAT-derived water mass fraction* ($w_{w,j}$) *and* $a_w$ *associated with organic component* $j$. *Using the VBS + BAT equilibrium solver, we generated a random database of 13,000 data points split into training data (70 %), validation data (15 %), and test data (15 %). This generated database was then used for the training of the NN.* *The NN output target is the vector of partitioning coefficients, which is subsequently used as the initial guess for solving the coupled VBS + BAT system of non-linear equations.*

- **BAT model:** Given that the BAT model is a multivariate function, a validation data set for it is also a good suggestion. We have added the following clarifications.

**Manuscript Revisions on Page 10-11:** *We generated a database of 37 known organic chemical structures and 123 artificial, yet possible chemical structures.* *There were an additional 16 organic chemicals used for a validation database (SI Table S6), and therefore not included in the fitting of the model.*
. . .
*For each structure there are an additional 40 data points at varying mole fractions, which means the training database has 6400 points and the validation database has 640 points.*
. . . *Page 11* . . .
*Generally, the BAT model showed good agreement to the training database with a root mean squared error (RMSE) in* $a_w$ *of 0.058 (5.8 % RH) and in organic activity* ($a_{org}$) *of 0.090. The validation database showed a similar agreement with a RMSE in* $a_w$ *of 0.066 and in* $a_{org}$ *of 0.096 (details in SI Sect. 5). The BAT model is valid for organic molecules within the following domain:* $0 \leq \mathrm{O}:\mathrm{C} \leq 2$ *and* $75 \leq M_{org} \leq 500\,\mathrm{g\,mol^{-1}}$ *with realistic behavior up to* $750\,\mathrm{g\,mol^{-1}}$. *Additional error analysis for the BAT model is shown in SI Sect 5.* *In panels (b) and (c) of Fig. 2, we show two examples of the BAT predictions, after domain-specific optimization, compared to the AIOMFAC-generated data.*

**SI Revisions:** SI Sect. 5, titled "BAT Model Validation and Error Analysis", Pages 8 – 12: *copied section begins*

**5 BAT Model Validation and Error Analysis**

Given that the BAT model is a multivariate function, a validation data set is used to assess the possibility of overfitting of the model depending on the training data set. The species used in the training and validation (Table S6) data sets are listed in the attached MS Excel file, the summary of the error analyses are shown in Table S5. Figure S3 compares the calculated water and organic activities at the same organic mole fraction, which is clearer than directly comparing activity coefficients from each model. For $O : C$ values lower than 0.2, the deviation from the 1:1 line is more substantial than the deviation for higher $O : C$ compounds. This is expected as such compounds show a miscibility gap over a wide range of composition space and associated high activities when computed for the initial, well-mixed single-phase case. For a quantitative assessment we calculated the root mean squared error (RMSE) of the activities predicted by the two models (AIOMFAC being the benchmark). If there was substantial overfitting, there would be a large difference in the RMSE between the training data and the validation data. For the RMSE calculation, we excluded the points where the activity was greater than one, as those represent unstable physical states and large deviations there can overwhelm the RMSE. Model–model deviations for those unstable cases are largely irrelevant in practise, because what matters is the comparison of the predictive skill for the composition of the stable phases (in LLPS or single-phase case). Table S5 lists the compiled error assessments for the training data and the validation data. The similar RMSE values between the hydroxyl training and validation data suggest the model is not overfitting and has general applicability within the training domain of the parameter space ($O : C$ and molar mass ranges). This agreement suggest that model behavior is realistic and our excess Gibbs function is smooth with no discontinuities. The smooth excess Gibbs function then leads to smooth activity curves and activity coefficients. Discontinuities like liquid–liquid phase separation are only derived from analysis of the excess Gibbs function (via post-processing) and are not directly built into the coefficients of the BAT model.

We did not generate additional validation data sets for the translation coefficients for each molecular functionality type for two reasons. First, our translation has only four coefficients and will be well constrained by 100+ data points used in the fit. Second, our translation function constitutes a smooth map; thus, no artifacts due to potential overfitting are expected. In addition to the thermodynamic activities, we can also compare how well we detect and predict the $a_{w,\text{sep}}$ point. For the organic compounds in the binary aquous systems that underwent phase separation, the RMSE of BAT vs. AIOMFAC $a_{w,\text{sep}}$ predictions are listed in Table S5. Overall the BAT $a_{w,\text{sep}}$ prediction was $< \pm 0.01$, the $a_w$ prediction was $< \pm 0.09$ (9 % RH), and the $a_{org}$ prediction was $< \pm 0.15$ compared to AIOMFAC.

[Figure]

**Figure 1.** :**SI Figure S5** Comparisons of the BAT predictions with the AIOMFAC model predictions for the validation data set (Table S6). An activity value above one represents an unstable or metastable mixing state, and in practice the mixture would phase separate readily when given that initial mixture composition. The color bar represents the $O:C$ of the compound, and each organic–water system is shown by 40 comparison points spanning the composition range from dilute to concentrated. Water activity $(a_w = \gamma_w(1 - x_{org}))$ is shown in (a) and organic activity $(a_{org} = \gamma_{org}x_{org})$ in (b). Both models are compared at the same organic mole fraction, $x_{org}$.

**Table 1.** :**SI Table S5** BAT model data point numbers for model fit and validation as well as root mean squared errors (RMSE) for the training and validation databases, which were generated by the AIOFMAC model.

| | Hydroxyl (training) | Hydroxyl (validation) | Carboxyl | Hydro-peroxide SOA | Hydro-peroxide | PEG | Ketone | Ether | Ester |
|---|---|---|---|---|---|---|---|---|---|
| Points for activity comparison ($a < 1$) | 5511 | 607 | 451 | 573 | 910 | 120 | 421 | 557 | 488 |
| RMSE of $a_w$ | 0.0580 | 0.0667 | 0.0408 | 0.0690 | 0.0711 | 0.0335 | 0.0845 | 0.0730 | 0.0820 |
| RMSE of $a_{org}$ | 0.0901 | 0.0964 | 0.0771 | 0.0950 | 0.0982 | 0.0520 | 0.1320 | 0.0970 | 0.1450 |
| Points for LLPS comparison | 52 | 4 | 5 | 9 | 5 | none | 10 | 9 | 21 |
| RMSE of $a_{w,\text{sep}}$ | 0.0066 | 0.0127 | 0.0031 | 0.0039 | 0.0061 | none | 0.0075 | 0.0032 | 0.0024 |

5.1 CCN Hygroscopicity Parameter Validation

We compare, in Fig. S4, measurement-derived $\kappa_{CCN}$ data against the BAT and AIOMFAC model predictions of $\kappa_{CCN}$. The validation dataset contained 16 supersaturated growth measurements on known chemical species, listed in Table S6 (Petters et al., 2009; Broekhuizen et al., 2004; Brooks et al., 2004; Frosch et al., 2010; Huff Hartz et al., 2006; Petters et al., 2016, 2006; P . The average error in the measurements is shown as the shaded gray region and is the average of the $\kappa_{CCN}$ range observed. The validation data shows similar agreement between the two models with a measurement vs. BAT RMSE of 0.061 and measurement vs. AIOMFAC RMSE of 0.059. The AIOMFAC $\kappa_{CCN}$ predictions are better in the miscibility transition region than those from the BAT model, but overall both models show similar predictions.

[Figure]

**Figure 2. :SI Figure S6** Comparison of single-component organics $\kappa_{CCN}$ measurements against those predicted by BAT (black circles) and AIOMFAC (white circles) model simulations of CCN activation. The blue, dashed lines connect the BAT and AIOMFAC predictions for the same species when there is a large difference. Gray shading represents $\pm\,42\,\%$ average uncertainty in the measured $\kappa_{CCN}$. The black dashed line is the BAT model linear fit with a zero intercept, $\kappa_{CCN,BAT} = \kappa_{CCN,measured} \times 0.78\,[\pm\,0.078]$ with a Pearson's $R^2 = 0.48$. The black dotted line is the AIOMFAC model linear fit with a zero intercept, $\kappa_{CCN,AIOMFAC} = \kappa_{CCN,measured} \times 0.75\,[\pm\,0.066]$ with a $R^2 = 0.57$. The RMSE between the measurements and predictions were 0.061 for BAT and 0.059 for AIOMFAC. The simulations assumed a 100 nm diameter equivalent volume of organic matter at the CCN activation point and the droplet surface tension was calculated as a volume-weighted mean. A list of the 16 validation points is given in Table S6

*copied section ends*

**Note, the table and citation errors of text overflow only show up in the tracked changes text, and not in the clean/revised manuscript and SI text.**

Related to this point: "Therefore, in an attempt to design a more general organic activity coefficient model, we made two important changes. First, we change the independent composition variable used in Eq. (15). Instead of mole fraction xorg, we introduce a scaled volume fraction ('org) in the series expansion of GE/RT. Second, we introduce a parameterization of the scalar c0 n coefficients by means of multivariate functions, which are dependent on common characteristics of organic molecules."

(1) Please provide some rationale why switching to volume fraction was required. It is not clear to the referee or from the text.

**Authors Response:** As described near the bottom of page 9: "The scaled volume fraction acknowledges that neither mole fraction nor volume fraction (nor mass fraction) perfectly accounts for the composition-dependence of activity coefficients when describing various binary systems." The switching to volume fraction is not required *per se*, but some form of scaling of the mole fraction composition scale is advantageous when the same binary Gibbs excess function is used for more than a single system and in particular when targeting a wide range in molecular masses of the organic component and therefore large shape and size differences compared to water molecules. Using volume fraction scaling is a better natural choice as it includes accounting for differences in molecular sizes, while the scaling further helps to achieve optimal model performance, which was also confirmed by preliminary tests we run before settling for the present functional form of the BAT model. We add the following text to explain this step.

**Manuscript Revisions on Page 9–10:** *The introduced change of composition scale improves the flexibility of this model when optimized for a wide range of binary systems characterized by the same set of model parameters ($s_1, s_2; a_{n,1}, a_{n,2}, a_{n,3}$, etc., with $n = 1, 2 \ldots$). The mole fraction scale works well for binary systems involving two components of similar molecular size and shape. However, this is rarely the case in aqueous organic mixtures with organic compounds of substantially higher molar mass than water. The volume fraction scale implicitly accounts to some extent for the size difference between organic and water molecules, which means that the coefficient functions $c_n$ do not need to correct for the molecular size- and composition-dependence as much as when mole fraction were used. It is for a similar reason that local composition models like UNIFAC describe organic molecules as a combination of similar-sized segments (subgroups) occupying a regular lattice, which contributes to the so-called combinatorial activity in those models. The scaled volume fraction acknowledges that neither mole fraction nor volume fraction (nor mass fraction) perfectly accounts for the composition-dependence of activity coefficients when describing various binary systems. Alternatively, a scaled mole fraction composition scale could have been used, but we chose to scale volume fractions as the scaling coefficient values constitute a smaller adjustment when used with this composition scale, meaning that a simpler scaling function was sufficient. Importantly, Eq. (19) remains consistent with all thermodynamic relations, including that $G^E$ becomes zero at both limits: $\phi_{org} = 0$ (when $x_{org} = 0$), $\phi_{org} = 1$ (when $x_{org} = 1$).*

(2) I don't understand why using Redlich-Kister was selected. The RK expansion can fit to arbitrary precision. While it is true that the model is thermodynamically consistent in the limit of x = 1 and x = 0, the polynomial can lead to maxima and minima in the excess [Gibbs] function that may or may not be realistic. This is particularly concerning since the polynomial coefficients are themselves computed through a multivariate model. (It is impressive that the system converged). I am concerned that the BAT model coefficients are overfitted and not be representative of other compositions, especially with respect to a phase separation which represents a discontinuity and is highly sensitive to very small fluctuations in the excess [Gibbs] function. This issue re-emphasizes the need to independently verify the fidelity of the BAT model. When addressing this concern, the authors should discuss why RK was selected instead of the Margules or van Laar model, which would be less sensitive to error from the use of polynomials by limiting the shape of the excess [Gibbs] function. There are a few comparisons to actual data. While it is clear that the model cannot be compared to every data point in the literature, the real-world validation appears not to be systematic. For example, it's not clear why single component data from Marsh et al. (2019) was selected for hygroscopic growth and various a-pinene SOA for CCN. The composition dependence of subsaturated water content on O:C for SOA (e.g. Pajunoja et al., 2015, doi:10.1002/2015GL063142) is far more revealing than the supersaturated data. Many data sets for single subsaturated water uptake of single component organic aerosol are available. Ideally a proper validation set

would systematically probe O:C and functional group coverage, and would considers experimental error. A plot like Figure 11 should be made for available single component CCN data, including for cases where LLPS is known to control CCN activity. A validated dataset with comparison against UNIFAC/LLPS is available in Petters et al. (2016, GMD, 10.5194/gmd-9-111-2016).

**Authors Response:**

**BAT function**: We chose the Redlich–Kister functional form specifically because it could be fitted to arbitrary precision and account for extrema in activity coefficients, if necessary; however, more than two coefficient terms ($c_1$, $c_2$) seemed to add little value to the fits (see description in Section 3.2). Use of two coefficient terms (in BAT as parameterized functions) also means that the shape of the Gibbs excess function is constrained towards realistic behavior, similar to a two-parameter Margules model. We also wanted the excess Gibbs function and activity coefficients to be capable of expressing maxima and minima as that behavior is important for models that allow for liquid–liquid phase separation, thus we did not use the van Laar model. Since we went with two polynomial terms and had 1000+ data points covering the range from very low to very high concentrations of aqueous organic systems to fit the model, the behaviour of BAT is well constrained in the O : C and molar mass space considered. Additional isopleth of Figure 3 for lower and higher molar masses were added to explore functional irregularities in the BAT Model Validation and Error Analysis section of the SI.

**Manuscript Revisions on Page 8:** *In addition $G^E$ must be capable of expressing maxima and minima within the mixed composition space ($0 < x_{org} < 1$) to correctly capture possible phase separation behavior.*

**Validations**: The concerns about validation/overfitting to AIOMFAC have been addressed in the response to the first comment; validation to measurements are discussed below.

We used the comparison to measurements by Marsh et al. (2019) mainly since the chemical species in those experiments are known. This is also why we did not use any measured OA data sets for subsaturated conditions, for which chemical composition is not well known. More comparisons and analyses of ambient and laboratory data sets (e.g. Pajunoja et al., 2015, doi:10.1002/2015GL063142) have been split off to future work as additional analysis is required (estimations/assumptions of volatility, molecular weight, and O : C distributions), which would distract from the the main point of introducing the VBS + BAT model. We have added a comparison plot showing modelled vs. measured $\kappa_{CCN}$ data as suggested for single-component (aqueous) organic aerosol.

**Manuscript Revisions on Page 30–32:** *After mainly comparing to data for subsaturated conditions in Fig. 11, we now focus on predictions for the regime supersaturated with respect to water vapor. In Fig. 12, the measurement derived $\kappa_{CCN}$ is compared with the corresponding BAT model prediction. The data set contains 30 supersaturated droplet activation measurements of known chemical species (e.g., oleic acid, glucose, and levoglucosan). The average error in the measurements is shown as the gray shaded area in Fig. 12, which covers the average of the $\kappa_{CCN}$ range observed for each component. A subset of 18 chemicals reported a $\kappa_{CCN}$ range, from which the average error was calculated to be $\pm 42\%$. The data set we used was compiled by Petters et al. (2016) and Petters and Kreidenweis (2007), which includes measurements derived from multiple sources (Broekhuizen et al., 2004; Brooks et al., 2004; Frosch et al., 2010; Huff Hartz et al., 2006; Petters et al., 2006; Petters and Kreide. Our comparison excludes the nitrogen-containing compounds. The BAT predictions assumed no organic co-condensation and had an evolving surface tension as described in Sect. 5.3. The BAT predictions vs. measurements had an RMSE of 0.055 and overall agreed within the reported measurement error. Substantial differences are found for the $0.35 < O : C < 0.55$ range, in which the resulting $\kappa_{CCN}$ is highly sensitive to a correct prediction of miscibility. For example, the miscibility is over-predicted*

*for phthalic acid (O : C = 0.5) while it is under-predicted for pinic acid (O : C = 0.44), shown in Fig. 12. In the full data set of 30 molecules, another subset of 16 molecules were not in the training database of the BAT model, so a corresponding plot with only this validation data is shown in the section 5.1 of the SI, including predictions by both BAT and AIOMFAC. The validation data shows similar agreement to Fig. 12, with a measurement vs. BAT RMSE of 0.061 and measurement vs. AIOMFAC RMSE*

5  *of 0.059. The AIOMFAC $\kappa_{CCN}$ predictions are better in the miscibility transition region than the BAT model, but overall the models show similar predictive skill for this metric. We chose to focus on well-defined chemical systems for all of the direct BAT model–measurement comparisons, allowing for minimal uncertainty in the input data. Additional comparisons of BAT to complex ambient and laboratory OA systems will be carried out in the future, since additional analyses are necessary for the estimation of volatility, molecular mass, and O : C distributions. Such analyses will enable a fair evaluation of VBS + BAT*

10  *model predictions against measurements for systems that are unresolved on the molecular composition level.*

[Figure]

**Figure 3. :Main text Figure 12:** Single-component organic aerosol measurements of $\kappa_{CCN}$ are compared against those predicted by corresponding BAT model simulations of CCN activation. The gray shading represents $\pm 42\%$ average uncertainty in the measured $\kappa_{CCN}$. The dashed line is a linear fit with a zero intercept, $\kappa_{CCN,BAT} = \kappa_{CCN,measured} \times 0.799\ [\pm 0.059]$ with a Pearson's $R^2$ of 0.66. The model–measurement RMSE was 0.055. The BAT simulations assume a 100 nm diameter equivalent volume of organic matter at the CCN activation point. The droplet surface tension is calculated as a volume-weighted mean. A list of the 30 measurement points is given in Table S6 of the SI, with the data obtained from the following studies: Broekhuizen et al. (2004); Brooks et al. (2004); Petters et al. (2006); Petters and Kreidenweis (2007); Petters et al. (2009, 2016); Frosch et al. (2010); Hu

⁚

Other comments

The tone of the hygroscopic growth and CCN section should be revised. For examples

"Over the past decade, the research community then progressed by characterizing (organic) aerosol hygroscopic growth measurements by a single $\kappa$ value, with sometimes inconsistent distinction between a $\kappa$ value at subsaturated and supersaturated humidity conditions."

"Our clear distinction between $\kappa$ CCN and the more general $\kappa$HGF helps the community understand clearly the subsaturated and supersaturated behavior of organic aerosol"

While it is true that there has been a debate on $\kappa$CCN and $\kappa$HGF the authors should acknowledge that 100s of experimental and modeling papers were devoted to this subject, with many important individual contributions explaining the origin of the discrepancy and the composition dependence of $\kappa$HGF. While the BAT model may capture some of these now very well understood effects, it does not really reveal anything new. Please rephrase the text and/or provide a more nuanced perspective on the topic.

**Authors Response:** Right, the BAT model does not reveal any new processes. Nevertheless, it may provide a different way to visualize the sub- vs. supersaturated hygroscopicity signatures. Everything we showed could also be done – and most has been done – with models like UNIFAC or AIOMFAC. We have added that point and revised this section's statements to address this reviewer's concerns. An advantage of the BAT model, with its intrinsic and continuous dependence on $M_{org}$ and $O:C$, is its ability to compute the isolines shown in Figure 10, which would have to be discretized by a set of molecular formulas in a similar figure when using UNIFAC/AIOMFAC.

**Manuscript Revisions on Page 25–27:**

*copied section begins*

[revised manuscript text omitted]

**1 Overview**

The supplemental information covers the BAT model equations and the approaches for the parameterizations of different functional group classes and phase separation treatments. These approaches include the $O:C$ blending method developed for the transition regions between the three BAT model parameterization regions, the functional group translations approach to convert input parameters to OH-group equivalents, finding the $a_{w,\text{sep}}$ point for the liquid–liquid transition from a organic-rich to a water-rich phase, and the density estimation method for organic compounds. The attached supplemental Microsoft$^{\circledR}$ Excel workbook file contains all the coefficient values, the SOA model system's input properties, validation systems, and all the data shown in the figures of the main text.

**2 BAT model**

**2.1 BAT Equations**

The explicit equations for our BAT model are listed below in Eqs. (S1) to (S11). To improve the clarity, we define $O:C \equiv \vartheta$, where $O:C$ refers to the $O:C$ of an organic component ("$org$") or the average $O:C$ of a mixture of organics. The determined coefficients are listed in Tables S1 & S2.

$$c_1 = a_{1,1} \exp(a_{1,2}\,\vartheta) + a_{1,3} \exp\left(a_{1,4}\,\frac{M_w}{M_{org}}\right) \tag{S1}$$

$$c_2 = a_{2,1} \exp(a_{2,2}\,\vartheta) + a_{2,3} \exp\left(a_{2,4}\,\frac{M_w}{M_{org}}\right) \tag{S2}$$

$$\phi_{org} = x_{org}\left(x_{org} + (1-x_{org})\frac{\rho_{org}}{\rho_w}\frac{M_w}{M_{org}}\left[s_1(1+\vartheta)^{s_2}\right]\right)^{-1} \tag{S3}$$

$$G^E/RT = \phi_{org}(1-\phi_{org})\left[c_1 + c_2(1-2\phi_{org})\right] \tag{S4}$$

$$\quad \frac{d(G^E/RT)}{dx_{org}} = \frac{d(G^E/RT)}{d\phi_{org}}\frac{d\phi_{org}}{dx_{org}} \tag{S5}$$

$$\frac{d\phi_{org}}{dx_{org}} = \left(\frac{\rho_{org}}{\rho_w}\frac{M_w}{M_{org}}\left[s_1(1+\vartheta)^{s_2}\right]\right)\left(\frac{\phi_{org}}{x_{org}}\right)^2 \tag{S6}$$

$$\frac{d(G^E/RT)}{dx_{org}} = \left\{(1-2\phi_{org})\left[c_1 + c_2(1-2\phi_{org})\right] - 2c_2\phi_{org}(1-\phi_{org})\right\}\frac{d\phi_{org}}{dx_{org}} \tag{S7}$$

$$\ln(\gamma_{org}) = (G^E/RT) + (1-x_{org})\frac{d(G^E/RT)}{dx_{org}} \tag{S8}$$

$$a_{org} = \gamma_{org} x_{org} \tag{S9}$$

$$\quad \ln(\gamma_w) = (G^E/RT) - x_{org}\frac{d(G^E/RT)}{dx_{org}} \tag{S10}$$

$$a_w = \gamma_w(1-x_{org}) \tag{S11}$$

Here, the activity coefficients of organic and water, $\gamma_{org}$ and $\gamma_w$, respectively, as well as the corresponding activities ($a_{org}$, $a_w$) are defined on mole fraction basis (i.e. $\gamma_{org} = \gamma_{org}^{(x)}$), each with the pure component as reference and standard states (where activity coefficients become unity). The output from the BAT calculation can also be used to calculate the Gibbs energy of

15 mixing ($\Delta_{\mathrm{mix}}G$), since the non-ideal interactions are parameterized (i.e., the excess Gibbs energy of mixing: $G^E$). Note, for simplicity, we do not include standard state chemical potentials of water and the organic, which would add an additional linear component to the curve. This is deemed justified given the approximate nature of the miscibility gap treatment. We present this calculation below with $\Delta_{\mathrm{mix}}G$ being normalized by $R$, $T$, and the total sum of moles $n_t = n_w + n_{org}$ in the binary system.

$$\frac{\Delta_{\mathrm{mix}}G^{\mathrm{ideal}}}{RTn_t} = (1-x_{org})\ln(1-x_{org}) + x_{org}\ln(x_{org}) \tag{S12}$$

$$\quad \frac{\Delta_{\mathrm{mix}}G}{RTn_t} = \frac{\Delta_{\mathrm{mix}}G^{\mathrm{ideal}}}{RTn_t} + \frac{G^E}{RTn_t} \tag{S13}$$

**Table S1.** Scaled volume coefficients of the fitted BAT model.

| Region | O : C bounds | Training data points | $s_2$ | $s_1$ |
|---|---|---|---|---|
| low O : C | O : C $< 0.15$ | 1000 | -5.988895 | 6.940689 |
| mid. O : C |  $0.05 <$ O : C $< \vartheta_{\mathrm{ML}} + 0.1$ | 2680 | -1.219164 | 4.742729 |
| high O : C | $\vartheta_{\mathrm{ML}} <$ O : C | 3600 | -0.078682 | 3.650860 |
| misciblity line | $0.05 <$ O : C $< 0.45$ | 2360 | -1.237227 | 4.069905 |

**Table S2.** The eight power series coefficients ($a_{n,1-4}$; $n = 1, 2$) used in the hydroxyl-group-parameterized BAT model.

| Region | $a_{1,1}$ | $a_{2,1}$ | $a_{1,2}$ | $a_{2,2}$ | $a_{1,3}$ | $a_{2,3}$ | $a_{1,4}$ | $a_{2,4}$ |
|---|---|---|---|---|---|---|---|---|
| low O : C | 7.089476 | -0.622678 | -7.711860 | -100.0 | -38.859410 | 3.08E-09 | -100.0 | 61.888120 |
| mid. O : C | 5.872214 | -0.974049 | -4.535007 | -100.0 | -5.129327 | 2.109751 | -28.092320 | -23.676830 |
| high O : C | 5.921550 | -100.0 | -2.528295 | -100.0 | -3.883017 | 1.353916 | -7.898128 | -11.601450 |
| misciblity line | 5.885109 | -0.984901 | -4.731250 | -6.227207 | -5.201652 | 2.320286 | -30.822970 | -25.840370 |

**2.2 Limit of Miscibility Line**

The limit of miscibility line is determined from an initial BAT model  fit involving the O : C region close to where the miscibility gap *vs.* complete miscibility transition occurs. We started by fitting the BAT coefficients using a wide O : C range (0.0 to 0.8) and then progressively narrowed it to the transition region (O : C 0.05 to 0.45). We then scanned O : C and $M_{org}$ to map out where the miscibility transition occurred (within BAT). The resulting O : C values  were used to fit the limit of miscibility line, $\vartheta_{\mathrm{ML}}$, as a function of organic molar mass,

$$\vartheta_{\mathrm{ML}} = \frac{0.205}{1 + \exp\left(26.6\left(\frac{M_w}{M_{org}} - 0.12\right)\right)^{0.843}} + 0.23. \tag{S14}$$

**2.3 O : C Transition Region Blending**

We used three different sets of fitted coefficients for the base BAT model representing hydroxyl functionality molecules. The split was based on the limit of complete miscibility of organics with water and further separated by O : C. A sigmoidal function was introduced to provide a smooth transition when traversing from one of the domains to the next in the 2-D parameter space (e.g., when O : C is increased gradually at a constant molar mass coordinate) – otherwise, spurious discontinuities would occur. The sigmoidal function provides a weighted map between the parameters from one domain to the next (over a short range in the boundary region). In effect, we are blending the different regions in the hydroxyl BAT model. Low to medium O : C region blending is listed first (Eqs. S15 to S22), where $\vartheta_{\mathrm{ML}}$ is the $\vartheta$ value at the limit of miscibility line and $b_1$, $b_2$, and $b_{\mathrm{ML}}$ are the

blending coefficients (Table S3). These are followed by and example of the blending weights as a function of $O : C$, Fig. S1.

$$\vartheta_b = \vartheta - \vartheta_{\mathrm{ML}} b_{\mathrm{ML}} \tag{S15}$$

$$\varpi_b = \frac{1}{1 + \exp[-b_1(\vartheta_b - b_2)]} \tag{S16}$$

$$\vartheta_{b,\,norm} = \vartheta - 0.75\,\vartheta_{\mathrm{ML}}\,b_{\mathrm{ML}} \tag{S17}$$

$$\varpi_{norm} = \frac{1}{1 + \exp(-b_1(\vartheta_{b,\,norm} - b_2))} \tag{S18}$$

$$\varpi_{mid} = \varpi_b / \varpi_{norm} \tag{S19}$$

$$\varpi_{low} = 1 - \varpi_{mid} \tag{S20}$$

$$G^E/RT\bigg|_{blended} = \varpi_{low}\, G^E/RT\bigg|_{low} + \varpi_{mid}\, G^E/RT\bigg|_{mid} \tag{S21}$$

$$\frac{d(G^E/RT)}{dx_{org}}\bigg|_{blended} = \varpi_{low}\, \frac{d(G^E/RT)}{dx_{org}}\bigg|_{low} + \varpi_{mid}\, \frac{d(G^E/RT)}{dx_{org}}\bigg|_{mid} \tag{S22}$$

Medium to high $O : C$ region blending (Eqs. S23 to S27):

$$\vartheta_b = \vartheta - \vartheta_{\mathrm{ML}} \tag{S23}$$

$$\varpi_{high} = \frac{1}{1 + \exp(-b_1(\vartheta_b - b_2))} \tag{S24}$$

$$\varpi_{mid} = 1 - \varpi_{high} \tag{S25}$$

$$G^E/RT\bigg|_{blended} = \varpi_{high} G^E/RT\bigg|_{high} + \varpi_{mid} G^E/RT\bigg|_{mid} \tag{S26}$$

$$\frac{d(G^E/RT)}{dx_{org}}\bigg|_{blended} = \varpi_{high} \frac{d(G^E/RT)}{dx_{org}}\bigg|_{high} + \varpi_{mid} \frac{d(G^E/RT)}{dx_{org}}\bigg|_{mid}. \tag{S27}$$

**Table S3.** Coefficients used in the blending of the different BAT coefficient regions for a molecule with hydroxyl functionality.

| Region Transition | $b_1$ | $b_2$ | $b_{\mathrm{ML}}$ |
|---|---|---|---|
| low to mid. $O : C$ | 79.2606902 | 6.04293E-02 | 0.1899745 |
| mid. to high $O : C$ | 75.0159268 | 9.47111E-04 | - |

**2.4**

[Figure]

**Figure S1.** Example of the blending weights used to merge the three regions in the BAT model. The $O:C$ is scanned with a fixed $M_{org}$ of $200\,\mathrm{g\,mol^{-1}}$ to show how each region becomes dominant.

**2.4    BAT Molecular Functionality Translation**

The translation approach concerns the conversion from different  molecular functionalities to hydroxyl-equivalent input parameters for use with the default, hydroxyl-group-based BAT model. These translations are for the whole molecule, and not the individual functional groups. Thus, for multifunctional molecules, a distinct multifunctional translation must be derived, as we did for the SOA oxidation products. If that is not possible, then the most dominant and representative functionality should be chosen. The $O:C$ conversion is described by Eq. S28 and the molar mass translation is described by Eq. S29. The corresponding coefficients for different oxygen-bearing functionalities of the whole molecule are listed in Table S4.

$$\vartheta_{eqv.\,\mathrm{OH}} = \frac{\vartheta}{1 + t_3 \exp(-t_1\,\vartheta)} \tag{S28}$$

$$M_{eqv.\,\mathrm{OH}} = \frac{M}{1 + t_4 \exp(-t_2\,M)} \tag{S29}$$

**Table S4.** Functional group translation coefficients to convert a whole molecule to a hydroxyl-equivalent molecule for BAT model inputs.

| $t_n$ | Hydroxyl | Carboxyl | Hydroperoxide | Hydroperoxide SOA | PEG | Ketone | Ether | Ester |
|---|---|---|---|---|---|---|---|---|
| $t_1$ | none | none | 8.1716E-06 | 1.4902E-04 | 5.4477E-03 | 4.5343E-03 | 2.4434E-05 | -1.293246 |
| $t_2$ | none | none | 4.5318E-07 | 4.7363E-03 | 3.864336 | 6.4845E-04 | 1.5832E-04 | 1.0813E-03 |
| $t_3$ | none | none | 0.966090 | 0.869058 | -0.267168 | 0.138144 | 0.284974 | 1.240514 |
| $t_4$ | none | none | 0.459433 | 0.564783 | 0.255487 | 0.352454 | 0.229339 | 0.405354 |

**3 Water Activity Separation Point**

In the case of a liquid–liquid equilibrium, the relative phase preferences are described by $q_j^\alpha$, the fractional liquid–liquid partitioning of a component to phase $\alpha$ ($q_j^\alpha \leq 1.0$ in the two-liquid-phases case). Liquid–liquid phase separation (LLPS) in a

5   binary water–organic system at RH $< 100\%$ is reduced to a point and manifests itself by a jump discontinuity. The liquid phase is either a water-poor ($\beta$) or water-rich ($\alpha$) phase, with a sharp transition between these two possible states at a specific water activity ($q_j^\alpha = 1$ or 0). To approximate the location and $a_w$-width over which the liquid–liquid phase separation is prescribed to occur, we first determine a designated reference point, the so-called water activity separation point ($a_{w,\text{sep}}$). Liquid–liquid phase separation connects two points on the Gibbs energy of mixing curve that have identical slopes and a tie-line that does not

10   cross the Gibbs energy curve (Fig. S2a). This tie-line represents the connection between the two stable phase compositions at equilibrium. Prior to phase separation occurring, a mixture can enter the composition space past these two points, which will result in a metastable state and eventually an unstable state, which will lead to spontaneous, spinodal decomposition (if phase separation did not occur within the metastable region). The binary mixture can enter and remain in the metastable region, but the energy barrier for liquid–liquid phase separation is typically low at room temperature, such that phase separation is expected

15   to occur when the water content is increased. In most cases we will be interested in a case of increasing or decreasing water mole fraction at approximately constant temperature, so our $a_{w,\text{sep}}$ point in Fig. S2a will be $p_2$, which has a corresponding point $p_5$ near/within the metastable composition range. If we solved for the tie-lines at high precision and included the standard state chemical potentials of water and the organic, then points $p_1$ and $p_2$ would have identical activities. That however is not the case, but we still want to ensure identical water activities at $a_{w,\text{sep}}$. We achieve this by finding $p_2$'s corresponding point

20   ($p_5$) which has the same water activity as the $a_{w,\text{sep}}$ point, this ensures a realistic water-poor ($\beta$) to water-rich ($\alpha$) transition.

Here, we explain how to identify (to good approximation) the two stable composition points in liquid–liquid equilibrium by only using the BAT-predicted activity curves (Fig. S2b). In a binary system, both component activities must be less than one and have monotonic behavior. Any regions that show non-monotonic behavior result in a phase separation range and are denoted by the dashed lines in Fig. S2b. By connecting the mole fraction extent of the organic and water activity-based

25   (minimum) phase separation regions identified, we can construct the tie-line that connects the two stable phases over the full extent of phase separation. This tie-line is then used in our above description to find the $a_{w,\text{sep}}$ point. We note that due to omitting a computationally costly Gibbs energy minimization (with further including standard chemical potentials), the identified miscibility gap is a (typically good) approximation of the true extent of phase separation.

[Figure]

**Figure S2.** BAT simulation used to describe the identification of the $a_{w,\text{sep}}$ point. The simulation uses an organic compound with hydroxyl functionalities, $M_{org}$ of $100\,\text{g}\,\text{mol}^{-1}$ and O : C of 0.225. The identified $a_{w,\text{sep}}$ value is here 0.9741 (black star). (a) The normalized $\Delta_{\text{mix}}G$ curve (black) with the tie-line in dashed red. The approximate stable phase-separation tie-line points and compositions are marked by $p_1$ and $p_2$, with the extent of the corresponding metastable regions denoted by $p_3$ and $p_4$. The end point in the metastable region at the same water activity as $p_2$ is marked by $p_5$. (b) The organic (green) and water (blue) mole-fraction-based activities for this binary system. The apparent minimum regions of phase separation required by each component are indicated by dashed lines. The approximate mole fraction extent of the actual phase separation region is identified by the extremes in composition, i.e. end points $p_1$ and $p_2$. The $a_{w,\text{sep}}$ point is the water activity corresponding to the composition at $p_2$, indicated by a black star.

**4 Organic Density Estimation**

Organic density model from **?**, Eqs. (S34 to S36). If $H : C$ is not known then we use $H : C = 2 - \vartheta$.

$$M_C = 12.010 \text{ g mol}^{-1} \tag{S30}$$

$$M_N = 14.006 \text{ g mol}^{-1} \tag{S31}$$

$$M_O = 16.0 \text{ g mol}^{-1} \tag{S32}$$

$$M_H = 1.008 \text{ g mol}^{-1} \tag{S33}$$

$$n_c = \frac{M_{org}}{M_C + M_H \, H : C + M_O \, \vartheta + M_H \, N : C} \tag{S34}$$

$$\rho^* = \frac{M_{org}}{5 n_c (2 + H : C + 2 \, \vartheta + 2 \, N : C)} \tag{S35}$$

$$\rho_{est.} = \rho^* (1 + \min(0.1 \, n_c \, \vartheta + 0.1 \, n_c \, N : C, \, 0.3)) \tag{S36}$$

**5 BAT Model Validation and Error Analysis**

Given that the BAT model is a multivariate function, a validation data set is used to assess the possibility of overfitting of the model depending on the training data set. The species used in the training and validation (Table S6) data sets are listed in the attached MS Excel file, the summary of the error analyses are shown in Table S5. Figure S3 compares the calculated water and organic activities at the same organic mole fraction, which is clearer than directly comparing activity coefficients from each model. For $O : C$ values lower than 0.2, the deviation from the 1:1 line is more substantial than the deviation for higher $O : C$ compounds. This is expected as such compounds show a miscibility gap over a wide range of composition space and associated high activities when computed for the initial, well-mixed single-phase case. For a quantitative assessment we calculated the root mean squared error (RMSE) of the activities predicted by the two models (AIOMFAC being the benchmark). If there was substantial overfitting, there would be a large difference in the RMSE between the training data and the validation data. For the RMSE calculation, we excluded the points where the activity was greater than one, as those represent unstable physical states and large deviations there can overwhelm the RMSE. Model–model deviations for those unstable cases are largely irrelevant in practise, because what matters is the comparison of the predictive skill for the composition of the stable phases (in LLPS or single-phase case). Table S5 lists the compiled error assessments for the training data and the validation data. The similar RMSE values between the hydroxyl training and validation data suggest the model is not overfitting and has general applicability within the training domain of the parameter space ($O : C$ and molar mass ranges). This agreement suggest that model behavior is realistic and our excess Gibbs function is smooth with no discontinuities. The smooth excess Gibbs function then leads to smooth activity curves and activity coefficients. Discontinuities like liquid–liquid phase separation are only derived from analysis of the excess Gibbs function (via post-processing) and are not directly built into the coefficients of the BAT model.

[Figure]

**Figure S3.** Comparisons of the BAT predictions with the AIOMFAC model predictions for the validation data set (Table S6). An activity value above one represents an unstable or metastable mixing state, and in practice the mixture would phase separate readily when given that initial mixture composition. The color bar represents the $O : C$ of the compound, and each organic–water system is shown by 40 comparison points spanning the composition range from dilute to concentrated. Water activity ($a_w = \gamma_w (1 - x_{org})$) is shown in (a) and organic activity ($a_{org} = \gamma_{org} x_{org}$) in (b). Both models are compared at the same organic mole fraction, $x_{org}$.

We did not generate additional validation data sets for the translation coefficients for each molecular functionality type for two reasons. First, our translation has only four coefficients and will be well constrained by 100+ data points used in the fit. Second, our translation function constitutes a smooth map; thus, no artifacts due to potential overfitting are expected.

5 In addition to the thermodynamic activities, we can also compare how well we detect and predict the $a_{w,\text{sep}}$ point. For the organic compounds in the binary aqueous systems that underwent phase separation, the RMSE of BAT vs. AIOMFAC $a_{w,\text{sep}}$ predictions are listed in Table S5. Overall the BAT $a_{w,\text{sep}}$ prediction was $< \pm 0.01$, the $a_w$ prediction was $< \pm 0.09$ (9 % RH), and the $a_{org}$ prediction was $< \pm 0.15$ compared to AIOMFAC.

**5.1 CCN Hygroscopicity Parameter Validation**

10 We compare, in Fig. S4, measurement-derived $\kappa_{\text{CCN}}$ data against the BAT and AIOMFAC model predictions of $\kappa_{\text{CCN}}$. The validation dataset contained 16 supersaturated growth measurements on known chemical species, listed in Table S6 (**???????????**). The average error in the measurements is shown as the shaded gray region and is the average of the $\kappa_{\text{CCN}}$ range observed. The validation data shows similar agreement between the two models with a measurement vs. BAT RMSE

**Table S5.** BAT model data point numbers for model fit and validation as well as root mean squared errors (RMSE) for the training and validation databases, which were generated by the AIOFMAC model.

| | Hydroxyl (training) | Hydroxyl (validation) | Carboxyl | Hydro-peroxide | Hydro-peroxide SOA | PEG | Ketone | Ether | Ester |
|---|---|---|---|---|---|---|---|---|---|
| Points for activity comparison $(a < 1)$ | 5511 | 607 | 451 | 573 | 910 | 120 | 421 | 557 | 488 |
| RMSE of $a_w$ | 0.0580 | 0.0667 | 0.0408 | 0.0690 | 0.0711 | 0.0335 | 0.0845 | 0.0730 | 0.0820 |
| RMSE of $a_{org}$ | 0.0901 | 0.0964 | 0.0771 | 0.0950 | 0.0982 | 0.0520 | 0.1320 | 0.0970 | 0.1450 |
| Points for LLPS comparison | 52 | 4 | 5 | 9 | 5 | none | 10 | 9 | 21 |
| RMSE of $a_{w,\mathrm{sep}}$ | 0.0066 | 0.0127 | 0.0031 | 0.0039 | 0.0061 | none | 0.0075 | 0.0032 | 0.0024 |

of 0.061 and measurement vs. AIOMFAC RMSE of 0.059. The AIOMFAC $\kappa_{\mathrm{CCN}}$ predictions are better in the miscibility transition region than those from the BAT model, but overall both models show similar predictions.

[Figure]

**Figure S4.** Comparison of single-component organics $\kappa_{CCN}$ measurements against those predicted by BAT (black circles) and AIOMFAC (white circles) model simulations of CCN activation. The blue, dashed lines connect the BAT and AIOMFAC predictions for the same species when there is a large difference. Gray shading represents $\pm\,42\,\%$ average uncertainty in the measured $\kappa_{CCN}$. The black dashed line is the BAT model linear fit with a zero intercept, $\kappa_{CCN,BAT} = \kappa_{CCN,measured} \times 0.78\,[\pm\,0.078]$ with a Pearson's $R^2 = 0.48$. The black dotted line is the AIOMFAC model linear fit with a zero intercept, $\kappa_{CCN,AIOMFAC} = \kappa_{CCN,measured} \times\,0.75\,[\pm\,0.066]$ with a $R^2 = 0.57$. The RMSE between the measurements and predictions were 0.061 for BAT and 0.059 for AIOMFAC. The simulations assumed a 100 nm diameter equivalent volume of organic matter at the CCN activation point and the droplet surface tension was calculated as a volume-weighted mean. A list of the 16 validation points is given in Table S6

Table S6: Chemical species used in the $\kappa_{CCN}$ measurement comparison, which contains the subset of 16 species used for BAT model validation.

| Start of Table S6 | | | | | | | | | |
|---|---|---|---|---|---|---|---|---|---|
| Chemical | Validation Data | BAT functionality | O : C | H : C | $M_{org}$ $(\mathrm{g\,mol^{-1}})$ | BAT $\kappa_{CCN}$ | AIOMFAC $\kappa_{CCN}$ | Measured $\kappa_{CCN}$ | Measurement Reference |
| Cetyl alcohol | yes | hydroxyl | 0.06 | 2.00 | 242.50 | 1.93E-06 | 0.053 | 2.00E-05 | Petter (2016) |
| Oleic acid | no | carboxyl | 0.11 | 1.78 | 282.47 | 4.03E-06 | 0.128 | 1.00E-05 | Petters (2009) |

| Chemical | Validation Data | BAT functionality | O : C | H : C | $M_{org}$ (g mol$^{-1}$) | BAT $\kappa_{CCN}$ | AIOMFAC $\kappa_{CCN}$ | Measured $\kappa_{CCN}$ | Measurement Reference |
|---|---|---|---|---|---|---|---|---|---|
| | | | | | | | | | Continuation of Table S6 |
| Stearic acid | no | carboxyl | 0.11 | 2.00 | 284.48 | 3.97E-06 | | 1.00E-05 | Petters (2009) |
| Palmitic acid | no | carboxyl | 0.13 | 2.00 | 256.43 | 4.66E-06 | | 1.00E-05 | Petters (2009) |
| Myristic acid | yes | carboxyl | 0.14 | 2.00 | 228.37 | 5.37E-06 | 0.053 | 1.00E-05 | Petters (2009) |
| Peroxide-ether | no | hydroper-oxide | 0.21 | 2.14 | 246.40 | 3.09E-06 | | 3.70E-03 | Suda (2014) |
| Peroxide-ether with aldehyde | no | hydroper-oxide | 0.29 | 2.00 | 260.00 | 4.10E-06 | | 9.20E-04 | Suda (2014) |
| Cis-Pinonic acid | yes | carboxyl | 0.30 | 1.60 | 184.24 | 0.054 | 0.106 | 0.005 | Petters (2016) |
| Pinonic acid | yes | carboxyl | 0.30 | 1.60 | 184.24 | 0.054 | 0.106 | 0.106 | Raymond (2003) and Petters (2007) |
| Peroxide-ether with acid | no | hydroper-oxide | 0.36 | 2.00 | 276.40 | 0.000 | | 0.020 | Suda (2014) |
| Diperoxide-diether | no | hydroper-oxide | 0.43 | 2.14 | 294.40 | 0.000 | | 0.011 | Suda (2014) |
| Azelaic acid | yes | carboxyl | 0.44 | 1.78 | 188.22 | 0.109 | 0.031 | 0.023 | Petters (2009) |
| Homophthalic acid | yes | carboxyl | 0.44 | 0.89 | 180.16 | 0.136 | 0.050 | 0.094 | Huff Hartz (2006) and Petters (2007) |
| Pinic acid | no | carboxyl | 0.44 | 1.56 | 187.21 | 0.114 | | 0.248 | Raymond (2003) and Petters (2007) |
| Norpinic acid | no | carboxyl | 0.50 | 1.50 | 172.18 | 0.129 | 0.179 | 0.182 | Raymond (2003) and Petters (2007) |

| Chemical | Validation Data | BAT functionality | O : C | H : C | $M_{org}$ $(\mathrm{g\,mol^{-1}})$ | BAT $\kappa_{CCN}$ | AIOMFAC $\kappa_{CCN}$ | Measured $\kappa_{CCN}$ | Measurement Reference |
|---|---|---|---|---|---|---|---|---|---|
| | | | | | | | | | |
| Phthalic acid | yes | carboxyl | 0.50 | 0.75 | 166.14 | 0.155 | 0.051 | 0.051 | Huff Hartz (2006) and Petters (2007) |
| Pimelic acid | yes | carboxyl | 0.57 | 1.71 | 160.17 | 0.137 | 0.133 | 0.150 | Frosch (2010) |
| Adipic acid | no | carboxyl | 0.67 | 1.67 | 146.14 | 0.156 | | 0.096 | Broekhuizen (2004) and Petters 2007 |
| Polyacrylic acid | no | carboxyl | 0.67 | 1.33 | 2000.00 | 0.017 | | 0.054 | Brooks (2004), and Petters (2009) |
| Polyacrylic acid | no | carboxyl | 0.67 | 1.33 | 2000.00 | 0.017 | | 0.051 | Petters (2006, 2007) |
| Glutaric acid | no | carboxyl | 0.80 | 1.80 | 147.13 | 0.157 | 0.133 | 0.106 | Petters (2009) |
| Levoglucosan | yes | hydroxyl | 0.83 | 1.67 | 162.14 | 0.147 | 0.140 | 0.208 | Svenningsson (2006) and Petters (2007) |
| Maltotriose hydrate | yes | hydroxyl | 0.89 | 1.78 | 504.44 | 0.050 | 0.028 | 0.055 | Petters (2009) |
| Sucrose | yes | hydroxyl | 0.92 | 1.83 | 342.30 | 0.071 | 0.061 | 0.095 | Petters (2009) |
| alpha-Ketoglutaric acid | yes | carboxyl | 1.00 | 1.20 | 146.11 | 0.181 | 0.179 | 0.310 | Petters (2016) |
| Erythritol | yes | hydroxyl | 1.00 | 2.50 | 122.12 | 0.181 | 0.180 | 0.140 | Petters (2009) |
| Glucose | yes | hydroxyl | 1.00 | 2.00 | 180.16 | 0.131 | 0.128 | 0.170 | Petters (2009) |
| Maleic acid | yes | carboxyl | 1.00 | 1.00 | 116.10 | 0.235 | 0.234 | 0.330 | Petters (2016) |
| Succinic acid | yes | carboxyl | 1.00 | 1.50 | 118.09 | 0.214 | 0.212 | 0.235 | Petters (2009) |

Continuation of Table S6

| Continuation of Table S6 | | | | | | | | | |
|---|---|---|---|---|---|---|---|---|---|
| Chemical | Validation Data | BAT functionality | O : C | H : C | $M_{org}$ $(\mathrm{g\,mol^{-1}})$ | BAT $\kappa_{\mathrm{CCN}}$ | AIOMFAC $\kappa_{\mathrm{CCN}}$ | Measured $\kappa_{\mathrm{CCN}}$ | Measurement Reference |
| Malonic acid | no | carboxyl | 1.33 | 1.33 | 104.06 | 0.261 | 0.234 | 0.227 | Pradeep Kumar (2003) and Petters (2007) |
| End of Table | | | | | | | | | |

**6  Additional Water Activity Isopleths**

The non-ideal behavior of water–organic mixtures is here explored at different molecular masses of the organic, analogous to Fig. 2 of the main text. This is used to probe for any functional irregularities and was used to place bounds on realistic BAT model behavior. In Fig. S5 the isopleths for 75, 100, 150, and 200 $\mathrm{g\,mol^{-1}}$ of organic molar mass are shown. The black region in Figs. S5 – S7 represent regions of phase separation due to water activity ($a_w > 1$) and light gray those due to organic activities ($a_{org} > 1$). In the 75 $\mathrm{g\,mol^{-1}}$ case (Fig. S5a), one can start to see irregular behavior in the black phase separation region as it has a bump at $\mathrm{O:C}$=0.25. The lower limit for reasonable behavior is then approximately 75 $\mathrm{g\,mol^{-1}}$ due to that irregularity – at least for $\mathrm{O:C} < 0.3$, while physically reasonable behavior is shown for higher $\mathrm{O:C}$ ratios. LLPS is clearly larger than the dark gray shaded areas as the 0.9 $a_w$ contour has identical activities for two different mole fractions of water, which is indicative LLPS. Figure S6 shows the $a_w$-isopleths as molecular mass increases: for 300, 500, 800, and 1000 $\mathrm{g\,mol^{-1}}$. Above 500 $\mathrm{g\,mol^{-1}}$ the model is unconstrained by training data and it is at these higher molecular masses that the contours indicate artifacts due to transition effects among the distinct $\mathrm{O:C}$ ranges of the three BAT model domains. The dips in the $a_w$-contours at an $\mathrm{O:C}$ of about 0.1 and 0.4 in Fig.S6c & d are non-physical. Such non-physical domain transition effects are further enhanced for high molar mass compounds when the x-axis shows the mole fraction of water. To get a clearer picture of this behavior at high molecular masses, we generated isopleths graphs for 500, 800, 1000, and 2000 $\mathrm{g\,mol^{-1}}$ (Fig. S7). We changed the x-axis to a mass fraction scale to better visualize the water uptake by these large molecules. In Fig.S7b, we can start to see irregular phase separation behavior indicated by an apparent region of miscibility at $0.1 < \mathrm{O:C} < 0.15$, with phase separation at slightly higher and lower $\mathrm{O:C}$. It is likely a non-physical artifact with a miscible region sandwiched between the black regions; it should very likely be one contiguous phase separation region. This irregular behavior then continues to expand as the molecular weight increases in Fig.S7c & d. However, we emphasize here that the gray areas only show the minimum extent of an LLPS region, while a liquid–liquid equilibrium computation (as done with VBS + BAT) needs to be done to determine the thermodynamically favoured parameter space exhibiting LLPS. If one is interested in phase separation predictions and BAT calculations for organics of $\mathrm{O:C} < 0.45$, then the BAT model is limited to the molar mass range below 750 $\mathrm{g\,mol^{-1}}$. If one is only interested in the $\mathrm{O:C}$ region above 0.5, then the BAT model should be applicable, with reasonable behavior exhibited up to at least 2000 $\mathrm{g\,mol^{-1}}$.

[Figure]

**Figure S5.** Predicted water activity contours generated by the BAT model for binary aqueous mixtures of generic organic compounds of constant molar mass yet variable $O:C$ at $T = 298.15$ K. The contours link water mole fraction and the organic $O:C$ to the resulting water activity in a binary water–organic mixture. The combined shaded regions in dark ($a_w > 1$) and light gray ($a_{org} > 1$) represent the minimum extent of liquid–liquid phase separation for a certain $O:C$. The bumps in the contours at $O:C$ of 0.1 and 0.3 stem from the transitions between the BAT model's low-, medium-, and high-$O:C$ parameterization domains. The $M_{org}$ used is as follows: (a) 75 g mol$^{-1}$, (b) 100 g mol$^{-1}$, (c) 150 g mol$^{-1}$, and (d) 200 g mol$^{-1}$.

[Figure]

**Figure S6.** Predicted water activity contours generated by the BAT model for binary aqueous mixtures of generic organic compounds of constant molar mass yet variable $O:C$ at $T = 298.15$ K. The contours link water mole fraction and the organic $O:C$ to the resulting water activity in a binary water–organic mixture. The combined shaded regions in dark ($a_w > 1$) and light gray ($a_{org} > 1$) represent the minimum extent of liquid–liquid phase separation for a certain $O:C$. The bumps in the contours at $O:C$ of 0.1 and 0.45 stem from the transitions between the BAT model's low-, medium-, and high-$O:C$ parameterization domains. The $M_{org}$ used is as follows: (a) 300 $g\,mol^{-1}$, (b) 500 $g\,mol^{-1}$, (c) 800 $g\,mol^{-1}$, and (d) 1000 $g\,mol^{-1}$

[Figure]

**Figure S7.** Predicted water activity contours generated by the BAT model for binary aqueous mixtures of generic organic compounds of constant molar mass yet variable $O:C$ at $T = 298.15$ K. **Note the change to a mass fraction scale.** The contours link water mass fraction and the organic $O:C$ to the resulting water activity in a binary water–organic mixture. The combined shaded regions in dark ($a_w > 1$) and light gray ($a_{org} > 1$) represent the minimum extent of liquid–liquid phase separation for a certain $O:C$. The bumps in the contours at $O:C$ of 0.1 and 0.45 stem from the transitions between the BAT model's low-, medium-, and high-$O:C$ parameterization domains. The $M_{org}$ used is as follows: (a) 500 g mol$^{-1}$, (b) 800 g mol$^{-1}$, (c) 1000 g mol$^{-1}$, and (d) 2000 g mol$^{-1}$

**7  SOA Mixtures**

The model comparison focuses on the predictions of bulk liquid aerosol mass concentration, and we used the AIOMFAC-based equilibrium gas–particle partitioning predictions as a benchmark. The AIOMFAC-equil. calculations include consideration of liquid–liquid phase separation and consider relatively high-fidelity input, as the AIOMFAC model uses functional group information and accounts for non-ideal interactions among all species. In contrast, the VBS + BAT approach only includes non-ideal water $\leftrightarrow$ organic interactions (implicitly assuming ideal organic $\leftrightarrow$ organic mixing) and rather limited molecular structure information (O : C and $M_{org}$). The full extent of the percentage difference in organic aerosol mass between the VBS + BAT approach and AIOMFAC-equil. is shown in Fig. S8.

For our simulated aerosol systems, we use surrogate systems representing $\alpha$-pinene SOA (Table S7) and isoprene SOA (Table S8) products based on predictions from the Master Chemical Mechanism, as was detailed in **?** and **?**, respectively. The $\alpha$-pinene SOA system used here contains 10 organic species as surrogates of the SOA, and the isoprene SOA system is comprised of 21 organic surrogate species. The input O : C and $M_{org}$ used for BAT are listed in Tables S7 & S8 and the molecular functionality translations to OH-equivalents (done internally in the model) are listed in square brackets.

[Figure]

**Figure S8.** Percent difference in organic aerosol mass between the VBS + BAT approach and AIOMFAC-equil. as a function of equilibrium relative humidity for a bulk solution ($= a_w$) at 298.15 K. Simulations for isoprene SOA are shown in blue and those for $\alpha$-pinene SOA in green. The benchmark AIOMFAC equilibrium predictions are shown for the salt-free cases (circles). The thick curves show the VBS + BAT prediction with different organic components, while the thin curve shows a simulation assuming an average molecule calculated from the dry mass, i.e., average O : C, H : C, $M_{org}$, and we kept the individual molecule's effective $C_{dry}^{sat}$. The thin dashed line shows the percent difference in the standard VBS simulation with no water uptake (dry).

Table S7: Properties of the $\alpha$-pinene SOA organic mixture used. The brackets denote the BAT model's internal molecular functionality translation.

| | | | | | | | |
|---|---|---|---|---|---|---|---|
| | | | | Start of Table S7 | | | |
| MCM Name | SMILES | BAT functionality | O : C [OH eqv.] | H : C | $M_{org}$ (g mol$^{-1}$) [OH eqv.] | $C^{g+\Sigma_\pi}$ ($\mu$g m$^{-3}$) | eff. $C_{dry}^{sat}$ ($\mu$g m$^{-3}$) |
| C107OOH | O=CCC1CC(OO)( C(=O)C)C1(C)C | hydroper- oxideSOA | 0.40 [0.22] | 1.60 | 200.17 [164.22] | 8.7918E+00 | 5.7429E+03 |
| C97OOH | OCC1CC(OO)( C(=O)C)C1(C)C | hydroper- oxideSOA | 0.44 [0.24] | 1.78 | 188.17 [152.78] | 3.9840E+00 | 3.2741E+02 |

| MCM Name | SMILES | BAT functionality | O : C [OH eqv.] | H : C | $M_{org}$ (g mol$^{-1}$) [OH eqv.] | $C^{g+\Sigma_\pi}$ (μg m$^{-3}$) | eff. $C^{sat}_{dry}$ (μg m$^{-3}$) |
|---|---|---|---|---|---|---|---|
| C108OOH | O=CCC(CC(=O)C(=O)C)C(C)(C)OO | hydroper-oxideSOA | 0.50 [0.27] | 1.60 | 216.13 [216.13] | 1.1344E+00 | 1.6671E+02 |
| PINIC | OC(=O)CC1CC(C(=O)C)C1(C)C | carboxyl | 0.44 [0.44] | 1.56 | 186.17 [186.17] | 6.2815E-01 | 1.4953E+01 |
| C921OOH | OCC(=O)C1(OO)CC(CO)C1(C)C | hydroper-oxideSOA | 0.56 [0.30] | 1.78 | 204.18 [168.09] | 9.1858E-01 | 2.1280E+00 |
| C812OOH | OCC1CC(OO)(C(=O)O)C1(C)C | hydroper-oxideSOA | 0.86 [0.46] | 1.75 | 195.17 [159.44] | 7.6636E-01 | 7.1911E-01 |
| C811OH | OCC1CC(C(=O)O)C1(C)C | hydroper-oxideSOA | 0.38 [0.20] | 1.75 | 158.17 [124.84] | 3.9949E-01 | 1.1569E+03 |
| C813OOH | OCC(CC(=O)C(=O)O)C(C)(C)OO | hydroper-oxideSOA | 0.75 [0.40] | 1.75 | 206.14 [169.98] | 3.1319E-01 | 3.0180E-02 |
| ALDOL-dimer | CC(=O)C(=O)CC(C(C=O)=CCC1CC(C(O)=O)C1(C)C)C(C)(C)OO | hydroper-oxideSOA | 0.37 [0.20] | 1.47 | 368.30 [335.21] | 4.0696E+00 | 2.7866E-06 |
| ESTER-dimer | CC1(C)C(CC1C(O)=O)CC(=O)OCC(=O)C2CC(CC(O)=O)C2(C)C | ester | 0.37 [0.12] | 1.56 | 368.31 [289.50] | 1.0174E+00 | 3.6370E-06 |

End of Table

Table S8: Properties of the isoprene SOA organic mixture used. The brackets denote the BAT model's internal molecular functionality translation.

| MCM Name | SMILES | BAT functionality | O : C [OH eqv.] | H : C | $M_{org}$ (g mol$^{-1}$) [OH eqv.] | $C^{g+\Sigma_\pi}$ (μg m$^{-3}$) | eff. $C^{sat}_{dry}$ (μg m$^{-3}$) |
|---|---|---|---|---|---|---|---|

| MCM Name | SMILES | BAT functionality | O : C [OH eqv.] | H : C | $M_{org}$ (g mol$^{-1}$) [OH eqv.] | $C^{g+\Sigma_\pi}$ (µg m$^{-3}$) | eff. $C^{sat}_{dry}$ (µg m$^{-3}$) |
|---|---|---|---|---|---|---|---|
| IEB1OOH | OCC(O)C(C)(OO)C=O | hydroper-oxideSOA | 1.00 [0.54] | 2.00 | 150.11 [117.51] | 3.2124E+00 | 5.0688E+01 |
| IEB2OOH | OOC(C=O)C(C)(O)CO | hydroper-oxideSOA | 1.00 [0.54] | 2.40 | 150.11 [117.51] | 2.4919E-01 | 2.3180E+02 |
| C59OOH | OCC(=O)C(C)(CO)OO | hydroper-oxideSOA | 1.00 [0.54] | 2.00 | 150.09 [117.50] | 4.2176E+00 | 2.2954E+01 |
| IEC1OOH | OCC(=O)C(C)(CO)OO | hydroper-oxideSOA | 1.00 [0.54] | 2.00 | 150.09 [117.50] | 1.4709E+00 | 2.2954E+01 |
| C58OOH | O=CC(O)C(C)(CO)OO | hydroper-oxideSOA | 1.00 [0.54] | 2.00 | 150.11 [117.51] | 3.3475E-01 | 5.0688E+01 |
| IEPOXA | CC(O)(CO)C1CO1 | hydroxyl | 0.60 | 2.00 | 118.13 | 8.6354E-11 | 3.5120E+13 |
| C57OOH | OCC(O)C(C)(OO)C=O | hydroper-oxideSOA | 1.00 [0.54] | 2.00 | 150.11 [117.51] | 2.7170E-01 | 5.0688E+01 |
| IEPOXC | CC1(CO1)C(O)CO | hydroxyl | 0.60 [0.60] | 2.00 | 118.13 [118.13] | 2.7879E-09 | 5.2036E+04 |
| HIEB1OOH | OCC(O)C(CO)(OO)C=O | hydroper-oxideSOA | 1.20 [.64] | 2.00 | 166.11 [132.13] | 2.8903E-01 | 1.0370E-01 |
| INDOOH | OCC(ON(=O)=O)C(C)(CO)OO | hydroper-oxideSOA | 1.40 [0.75] | 2.20 | 197.14 [161.32] | 2.5037E-01 | 4.5117E-01 |
| IEACO3H | CC(O)(C1CO1)C(=O)OO | hydroper-oxideSOA | 1.00 [0.54] | 1.60 | 148.10 [115.69] | 5.3463E-08 | 5.6321E+04 |
| C525OOH | OCC(=O)C(CO)(CO)OO | hydroper-oxideSOA | 1.20 [0.64] | 2.00 | 166.09 [132.12] | 2.1592E-01 | 3.9838E-02 |
| HIEB2OOH | OOC(C=O)C(O)(CO)CO | hydroper-oxideSOA | 1.20 [0.64] | 2.00 | 166.11 [132.13] | 1.4203E-01 | 7.0484E-01 |
| IEC2OOH | OCC(=O)C(C)(OO)C=O | hydroper-oxideSOA | 1.00 [0.54] | 1.60 | 148.06 [115.66] | 2.0876E-06 | 4.2944E+03 |
| INAOOH | OCC(C)(OO)C(O)CON(=O)=O | hydroper-oxideSOA | 1.40 [0.75] | 2.20 | 197.14 [161.32] | 1.3898E-01 | 1.7351E+00 |

| | | | Continuation of Table S8 | | | | | |
|---|---|---|---|---|---|---|---|---|
| MCM Name | SMILES | BAT func-tionality | O : C [OH eqv.] | H : C | $M_{org}$ $(\mathrm{g\,mol^{-1}})$ [OH eqv.] | $C^{g+\Sigma_\pi}$ $(\mathrm{\mu g\,m^{-3}})$ | eff. $C^{sat}_{\mathrm{dry}}$ $(\mathrm{\mu g\,m^{-3}})$ | |
| C510OOH | O=CC(O)C(C)(OO)CON(=O)=O | hydroper-oxideSOA | 1.40 [0.75] | 1.8 | 195.10 [159.38] | 4.1752E-03 | 2.6990E+02 | |
| INB1OOH | OCC(OO)C(C)(CO)ON(=O)=O | hydroper-oxideSOA | 1.40 [0.75] | 2.20 | 197.14 [161.32] | 7.1561E-02 | 4.2126E-01 | |
| IECCO3H | CC1(CO1)C(O)C(=O)OO | hydroper-oxideSOA | 1.00 [0.54] | 1.60 | 148.11 [115.71] | 7.5983E-07 | 1.8033E+04 | |
| INCOOH | OCC(OO)C(C)(O)CON(=O)=O | hydroper-oxideSOA | 1.40 [0.75] | 2.20 | 197.14 [161.32] | 3.0754E-02 | 7.3141E+00 | |
| INB2OOH | OOCC(O)C(C)(CO)ON(=O)=O | hydroper-oxideSOA | 1.40 [0.75] | 2.20 | 197.14 [161.32] | 3.4893E-02 | 1.4651E+00 | |
| 2-Methyltetrol-dimer | CC(O)(CO)C(O)COC(C)(CO)C(O)CO | hydroxyl | 0.70 [0.70] | 2.30 | 254.28 [254.28] | 7.2215E+00 | 2.5788E-06 | |
| | | | End of Table | | | | | |